# Data Provenance for Image Auto-Regressive Generation

**Bihe Zhao, Louis Kerner, Michel Meintz, Tameem Bakr, Franziska Boenisch, Adam Dziedzic***
CISPA Helmholtz Center for Information Security

## Abstract

Image autoregressive models (IARs) have recently demonstrated remarkable capabilities in visual content generation, achieving photorealistic quality and rapid synthesis through the next-token prediction paradigm adapted from large language models. As these models become widely accessible, robust data provenance is required to reliably trace IAR-generated images to the source model that synthesized them. This is critical to prevent the spread of misinformation, detect fraud, and attribute harmful content. We find that although IAR-generated images often appear visually identical to real images, their generation process introduces characteristic patterns in their outputs, which serves as a reliable provenance signal for the generated images. Leveraging this, we present a post-hoc framework that enables the robust detection of such patterns for provenance tracing. Notably, our framework does not require modifications of the generative process or outputs. Thereby, it is applicable in contexts where prior watermarking methods cannot be used, such as for generated content that is already published without additional marks and for models that do not integrate watermarking. We demonstrate the effectiveness of our approach across a wide range of IARs, highlighting its high potential for robust data provenance tracing in autoregressive image generation.

## 1 Introduction

Recent progress in image autoregressive models (IARs) has led to significant advancements in image generation. Driven by advances in language modeling, these models produce high-quality images at rapid pace using the next-token prediction paradigm (Tian et al., 2024; Han et al., 2025). As IAR-generated images become visually indistinguishable from natural content, several challenges, including the spread of misinformation, fraud, and harmful content dissemination, arise. Additionally, as generated data "pollutes" the data ecosystem, it is increasingly used to train new generative models, which degrade model performance (called *model collapse* (Alemohammad et al., 2024; Shumailov et al., 2024)) and amplify existing biases (Wyllie et al., 2024). Therefore, data provenance, *i.e.,* identifying and attributing generated images to their generators, is highly important.

Several provenance methods have been developed for generative vision models, including both watermarking (Fernandez et al., 2023; Liu et al., 2023; Wen et al., 2023; Zhao et al., 2023; Gunn et al., 2024; Kerner et al., 2025; Jovanović et al., 2024; Tong et al., 2025; Wang et al., 2025b) and fingerprinting (Kim et al., 2024; Yu et al., 2021; Nie et al., 2023). Yet, these methods require the integration of additional signals into the models or into the images either during or after generation. This introduces perceptible or statistical changes, is not applicable to trace provenance for content that has already been published without marks, and often results in a trade-off between robustness, imperceptibility, and applicability.

In this work, we propose the first post-hoc provenance framework for IAR-generated images that does not require *any* modification of the generation process, is model-agnostic, and applicable to previously published, and unmarked content. Our proposed framework builds on our intriguing observation that because IARs encode images as sequences of discrete tokens from a fixed codebook, *i.e.,* their "vocabulary", they introduce a quantization step that leaves model-specific artifacts in the generated images as shown in Figure 1. Specifically, token representations of generated images

---

*Correspondence to adam.dziedzic@cispa.de.

are consistently closer to the codebook entries than those of natural images. We refer to this as *QuantLoss* and show that it can be used to trace IAR-generated images back to their generators.

We further enhance the reliability of our framework by amplifying existing signals and integrating additional, carefully designed ones. First, we train a model to approximate the *inverse of the IAR's decoder* and use it to encode images whose provenance we want to test. Since the inverted decoder leads to a higher fidelity mapping from images to the tokens that the image was potentially generated from, it strengthens our provenance signals based on QuantLoss. Additionally, we introduce a novel token-search algorithm for the next-scale prediction paradigm, enabling more accurate tracking of generated images' initial tokens and enabling a more meaningful comparison to the codebook tokens. Beyond QuantLoss, we also identify a complementary signal, *EncLoss*, which captures the deviation observed when encoding an image to the latent space with the inverse decoder and then decoding it back to the image space with the original decoder. This process yields low loss for generated images due to feature consistency, but typically greater loss for natural images. All these signals can eventually be combined to robustly trace the provenance of IAR-generated images.

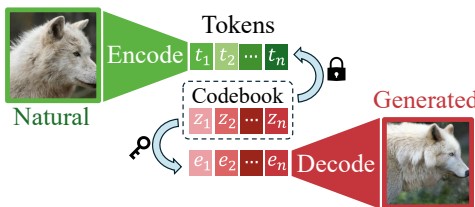

Figure 1: **Data Provenance: Token Space.** Since the generated tokens of a given IAR are sampled from the codebook entries, the codebook acts as a key to distinguish the token representations of generated images from those of real images.

We evaluate our method on state-of-the-art IARs, including VAR (Tian et al., 2024), RAR (Yu et al., 2024b), LlamaGen (Sun et al., 2024), Taming (Esser et al., 2021), and Infinity (Han et al., 2025), as well as a vector-quantized diffusion model (VQ-Diffusion) (Gu et al., 2022). We are able to detect the images generated from these models with almost 100% success rate, which contributes to a reliable provenance tracing of generated content in IARs. We note that the post-hoc finetuning of the encoder has relatively small overhead compared to IAR training, especially for the newly developed IARs with increasing model scale and training data. We also analyze the robustness of our method to conventional image post-processing techniques and show that our method can still detect most of the generated content, significantly outperforming the existing methods.

In summary, we make the following contributions:

1. We introduce the first post-hoc data provenance method for IARs that leverages generation-specific artifacts to reliably determine whether, and by which model, an image was generated. Our framework does not require any modifications to the model's training or generation process, is model-agnostic, and applicable to already published generated content.

2. We show that combining carefully designed provenance signals derived from the generation-specific artifacts enables near-perfect detection of generated images and accurate attribution to their source model, consistently achieving nearly 100% TPR@1%FPR across a diverse set of IARs and outperforming all baselines.

3. We provide a thorough empirical evaluation of our framework on diverse models and with diverse datasets, assessing its robustness to image perturbations. Our results highlight that our framework can provide effective provenance tracing under real-world scenarios with non-perfect data, highlighting its practical applicability.

## 2 BACKGROUND AND RELATED WORK

**Image Autoregressive Models (IARs).** IARs have recently gained traction as a new architecture for image generation, following the success of generative adversarial networks (Karras et al., 2020; Choi et al., 2020; Karras et al., 2021) and diffusion models (Rombach et al., 2022; Saharia et al., 2022; Podell et al., 2023). IARs inherit the *next-token prediction* paradigm from large language models (LLMs) by treating images (or their patches) as sequences of discrete tokens which enables them to generate images both quickly and with high quality. Building on the advances from LLMs also allows IARs to follow their clear power-law scaling (Tian et al., 2024). In the last years, the progress in autoregressive image generation has moved from early pixel-space raster-scan autore-

gression (Chen et al., 2020; Van den Oord et al., 2016) to pioneering efforts with the next scale or resolution prediction (VAR) (Tian et al., 2024; Han et al., 2025). Recent proposals opt for next-token prediction of randomized inputs permuted into different factorization orders with annealing probability (RAR) (Yu et al., 2024b), and to even vanilla autoregressive models that apply the exactly same *next-token prediction* as LLMs and feature an image tokenizer with high-quality reconstruction and high-utilization of the codebook (LlamaGen) (Sun et al., 2024), along with many other recent contributions advancing the state-of-the-art in autoregressive image generation (Ren et al., 2025; Li et al., 2024; Team, 2024; Yu et al., 2024a; Shao et al., 2025; Deng et al., 2025; Tang et al., 2025). Most contemporary IARs (Sun et al., 2024; Tian et al., 2024; Yu et al., 2024b; Han et al., 2025) are composed of an autoencoder and an autoregressive model. The autoencoders, also known as image tokenizers, function as a mapping between the pixel and the token space, which are usually built on the VQGAN architecture introduced in Taming (Esser et al., 2021). In our work, we leverage the pixel-token mapping to reliably trace generated images.

**Image Provenance.** The goal of image provenance is to attribute the entity (*e.g.,* model or content creator) that generated a given image. Existing methods can be grouped into watermark-based (Kerner et al., 2025; Jovanović et al., 2025; Tong et al., 2025), fingerprint-based (Kim et al., 2024; Yu et al., 2021; Nie et al., 2023), and reconstruction-based approaches (Wang et al., 2023; 2024; 2025a). **Watermark-based** and **fingerprint-based methods** embed model-specific information into the generation process or training pipeline to enable source identification (Zhao et al., 2025; Mahara & Rishe, 2025). Since these approaches require interventions either in the training or inference stage, they degrade the quality of the generated image and cannot be performed retroactively after the models or their non-marked generations are released. On the contrary, **reconstruction-based methods** leverage the innate features of the generative models for image attribution, which do not introduce any perturbations to the generated images. For example, RONAN (Wang et al., 2023) is proposed to attribute the image generated by variational autoencoders (VAE), GANs, and diffusion models by reverse-engineering the generative process back to its input space. RONAN is not directly applicable to IARs as it is only effective for deterministic generation, while IARs rely on a random sampling process during each of their next-token predictions. LatentTracer (Wang et al., 2024) is proposed specifically for diffusion models to trace generated images by optimizing in the latent space of a decoder. Although LatentTracer can be applied to IARs, it demonstrates suboptimal performance and proves computationally expensive for many images. Recently, Wang et al. (2025a) proposed to calibrate the reconstruction loss by double reconstruction to improve attribution performance for diffusion models. However, we show that this method has insufficient performance for IARs. In contrast, our method provides effective and reliable provenance of IAR-generated images.

**Membership Inference Attack.** While our work focuses on data provenance, it is important to distinguish it from membership inference attacks (MIAs), which address a fundamentally different attribution problem. MIA aims to determine whether a given data point was part of a model's training set (Shokri et al., 2017; Salem et al., 2019). MIAs are commonly used for auditing privacy leakage (Steinke et al., 2023; Rossi et al., 2026; Marek et al., 2026). In contrast, data provenance seeks to identify whether a given image was *generated* by a model, which is critical for tracing synthetic content and preventing model collapse caused by training on generated data (Alemohammad et al., 2024; Shumailov et al., 2024). Importantly, existing MIA methods for IARs require access to class labels or text prompts in addition to images (Kowalczuk et al., 2025; Yu et al., 2025), which are generally unavailable for generated images found in the wild. Our post-hoc provenance framework operates solely on images themselves, making it applicable to real-world scenarios where auxiliary information is absent and where content has already been published without metadata.

## 3 METHOD

We begin by outlining the necessary preliminaries and notation for vector-quantized representations. Next, we formalize the problem of provenance tracing in IARs. Finally, we introduce our data provenance framework, presenting both the QuantLoss-based and EncLoss-based signals that we develop for effective post-hoc provenance detection.

### 3.1 PRELIMINARIES ON VECTOR-QUANTIZED REPRESENTATIONS

IARs tokenizers consist of three main components:

1. **Encoder** $E$: A convolutional neural network (CNN)-based feature extractor with down-sampling ratio $p$ that projects input pixels $x \in \mathbb{R}^{H \times W \times 3}$ to a latent feature map $f \in \mathbb{R}^{\frac{H}{p} \times \frac{W}{p} \times C}$, where $H \times W$ are spatial dimensions and $C$ denotes the channels. We denote the encoding as $x \xrightarrow{E} f$.

2. **Quantizer** $Q$: The main part of the quantizer is a codebook $Z \in \mathbb{R}^{N \times C}$ containing $N$ learnable prototype vectors, each with the channel dimension $C$. The index of each prototype vector serves as a discrete *token* for quantization. Every spatial feature $f^{(i,j)}$ is mapped to its nearest entry $z_n$ ($n \in [N]$) in codebook $Z$ to obtain the integer indices $t_Z^{(i,j)}$. We denote the quantization as $f \xrightarrow{Q} t_Z$ and the dequantization (mapping from the tokens $t_Z$ to the quantized feature map $f_Z$) as $t_Z \xrightarrow{Q^{-1}} f_Z$, and define them as follows:

$$Q : t_Z^{(i,j)} = \arg\min_{n \in [N]} \|f^{(i,j)} - z_n\|_2, \quad Q^{-1} : f_Z^{(i,j)} = Z[t_Z^{(i,j)}]. \tag{1}$$

3. **Decoder** $D$: A CNN symmetric to the encoder that decodes the quantized feature map $f_Z$ to the image $x_Z$. We denote the decoding as $f_Z \xrightarrow{D} x_Z$.

Together, the stages in the above framework can be expressed as:

$$x \xrightarrow{E} f \xrightarrow{Q} t_Z \xrightarrow{Q^{-1}} f_Z \xrightarrow{D} x_Z. \tag{2}$$

The training of a IAR model is performed in two stages: (1) the encoder and decoder pair, including the quantizer with its codebook, are pre-trained, followed by the (2) training of the *autoregressive transformer* (AR) to predict the next tokens during generation. The encoder and quantizer are only used for mapping images from pixels to tokens during training, while the decoder is used for mapping AR-generated tokens to image pixels during both training and inference (generation).

## 3.2 Problem Formulation

Given a suspect image $x$ and a IAR model $M$, our goal is to develop a framework that attributes the image $x$ to the given IAR, or identifies $x$'s provenance as not generated by $M$. $x$ can be *any* image, including a natural one or an image generated by a generative model. For $M$, we assume white-box access to its encoder $E$, decoder $D$, and quantizer $Q$, which represents a realistic setup as many of the state-of-the-art IARs are open-source. Most importantly, we only assume post-hoc provenance, *i.e.,* $M$ and $x$ are already given and it is not possible to modify the training or generation process.

## 3.3 A Framework for Data Provenance in IARs

We introduce two complementary signals specifically designed for IAR provenance detection: QuantLoss (Section 3.3.1) and EncLoss (Section 3.3.2). Finally, we describe how we combine these two into a joint provenance signal for our framework presented in Figure 2.

### 3.3.1 QuantLoss: Provenance Signal based on Codebook Distance

We design our QuantLoss provenance signals based on our observation that the token representations differ significantly between natural and IAR-generated images. Intuitively, the representations of generated images are consistently closer to the codebook entries than those of natural images (see Figure 1). We first formalize this **observation**, then describe how we leverage it as a provenance feature, and finally introduce the feature's two core building blocks, namely the **decoder inversion** and the **quantization**.

**Formalizing our Observation on Proximity to Codebook Tokens.** The *generation* of an image $x_Z$ by a IAR with codebook $Z$ is formalized in Equation (3). The generated image $x_Z$ is initially sampled as discrete tokens $t_Z$ from the token sample space $T_Z$ by the IAR. This sample space for generated image consists of the possible combinations of tokens from the codebook $Z$. In contrast, natural images are drawn from natural (real-world) data distributions, and therefore have a much larger and more diverse space. In addition, different IARs have distinct codebooks corresponding to distinct sample spaces. Thus, the codebooks of different IARs naturally serve as a provenance signal for the synthetic images. In essence, IAR-generated images leave a "fingerprint" in token

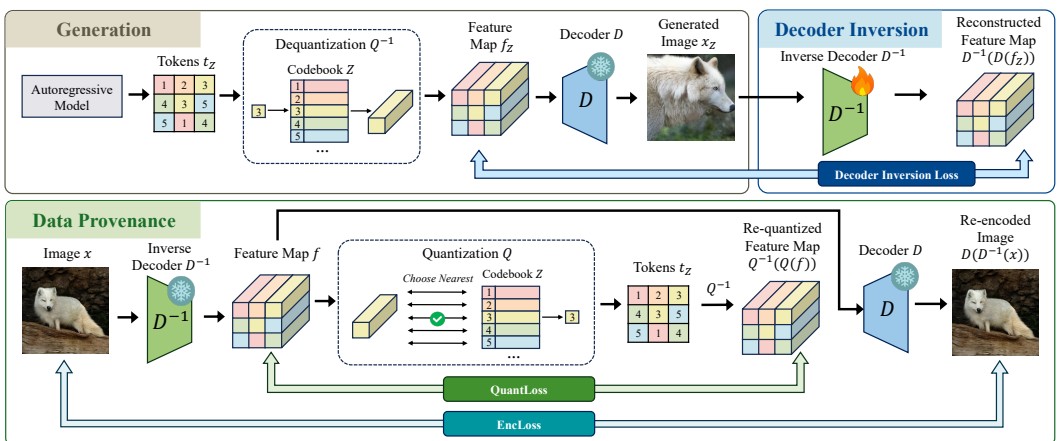

Figure 2: **Overview of our data provenance framework for IARs.** (1) During **Generation**, the tokens are generated by the autoregressive model, dequantized to a feature map, and decoded to a generated image. (2) Our **Decoder Inversion** aims at creating an inverse decoder that recovers the generated feature map from the generated image. (3) We propose two signals for our **Data Provenance**: *QuantLoss* between the feature map recovered by the inverse decoder and its re-quantized version, and *EncLoss* between the image and its reencoded version.

space because they were originally constructed from specific codebook entries. This observation leads to our key insight: generated images, when *inverted* back through the decoder and correctly quantized, will have feature representations that align closely with codebook entries, while natural images or images generated by other IARs will exhibit larger quantization errors.

$$
\begin{aligned}
\text{Generation:} \quad & T_Z \overset{AR}{\sim} t_Z \xrightarrow{Q^{-1}} f_Z \xrightarrow{D} x_Z \\
\text{Inversion:} \quad & t_Z \xleftarrow{Q} f_Z \xleftarrow{D^{-1}} x_Z
\end{aligned}
\tag{3}
$$

**Designing a Provenance Signal Based on our Observation.** From this insight, we design a signal to detect if an image was sampled from a codebook $Z$. We transform a given image $x$ to its continuous feature map $f$ and then to its codebook-based quantized feature map $f_Z$. The process can be expressed as follows:

$$
x \xrightarrow{D^{-1}} f \xrightarrow{Q} t \xrightarrow{Q^{-1}} f_Z.
\tag{4}
$$

Specifically, we first transform the image $x$ to the latent space with an inverted decoder $D^{-1}$. If $x$ was generated by the target IAR from $T_Z$, then with an ideal inverse decoder $D^{-1}$, the recovered feature map $f$ should already be quantized (i.e., each feature vector should exactly match a codebook entry). Therefore, the quantization step $f \xrightarrow{Q} t \xrightarrow{Q^{-1}} f_Z$ would introduce minimal error, making $f \approx f_Z$. Conversely, if $x$ is a natural image or from a different IAR, $f$ will not align with the codebook entries, resulting in significant quantization error and $f \neq f_Z$. We compute the QuantLoss $\mathcal{L}_{\text{Quant}}$ between the feature map $f$ and its quantized version $f_Z$ as follows:

$$
\mathcal{L}_{\text{Quant}}(x) = \|f - f_Z\|_2 = \|f - Q^{-1}(Q(f))\|_2,
\tag{5}
$$

**Decoder Inversion: Obtaining $D^{-1}$.** We aim at inverting an image $x$ to the quantized feature map $f_Z$. Intuitively, if $x$ was generated by the given IAR, the feature map $f_Z$ is close to the codebook entries of $Z$. This requires first inverting the decoder $D$. A naïve solution would be to apply the IARs original encoder $E$. However, we observe that $E$ is not a close inversion of $D$ for generated images for most IARs (what we show in Table 4). We attribute this behavior to the fact that $E$ is trained on natural images. To obtain a closer approximation of the inversion $D^{-1}$ of the decoder, we instead train an inversion model. Concretely, we initialize this model's weights with the original encoder weights and finetune this inverse decoder on images generated by the given IAR (see Equation (2)). During finetuning, the codebook $Z$ and the decoder $D$ are frozen, and we use the following loss to optimize $D^{-1}$:

$$
\mathcal{L}_{\text{inv}} = \|f_Z - D^{-1}(D(f_Z))\|_2.
\tag{6}
$$

Notably, this step is performed post-hoc after release of the IAR and the data point $x$. It does not interfere with the models training or the generation of data points. Given that the finetuning exclusively relies on images produced by the target IAR, it does not require the costly curation of additional training data. Finally, we can improve robustness to data augmentations by applying them to the finetuning data and training $D^{-1}$ to generate consistent quantized feature maps for both the original and augmented images. As shown in Section 4.3, this approach makes our framework less sensitive to common data perturbations and significantly enhances our method's reliability.

**Quantization $Q$.** While inverting the decoder traces the image to the latent space, we need to further invert the feature map to the token space, as the token space is where generated images are initially sampled from. To this end, we perform quantization for both single-scale and multi-scale IARs to invert a feature map $f_Z$ to tokens $t_Z$.

*Single-scale* IARs, such as RAR (Yu et al., 2024b). which generate an image through next-token prediction, tokenize images as a single feature map, where each feature corresponds to only one token and one entry in the codebook. During generation, each token is mapped to one of the spatial features in the feature map by querying the codebook ($f_Z = Q^{-1}(t_Z) = Z[t_Z]$). This process can simply be inverted by the corresponding quantization $Q$, defined in Equation (1).

*Multi-scale* IARs, such as VAR (Tian et al., 2024), redefine the autoregressive image generation as next-scale prediction. They generate an image starting from tokens responsible for low-level features in an image, and then generate the tokens for high-level details based on tokens in the previous scales. After generating tokens in all scales, the tokens are mapped to codebook entries, upsampled, and summed up to obtain the feature map. This process can be formalized as:

$$\{t_Z^{(k,i,j)}\}_{k=1}^K \xrightarrow{\text{Codebook } Z} \{f_Z^{(k,i,j)}\}_{k=1}^K \xrightarrow{\text{Upsample and Sum}} f_Z^{(i,j)}, \tag{7}$$

where $K$ is the number of scales. When quantizing the feature map to scalewise tokens, multi-scale IARs apply a scalewise greedy search: for each scale, the nearest codebook entry of a given feature is selected as the token. The detailed algorithm for the original quantization of VAR is presented in Algorithm 2. However, as tokens from *all* scales contribute to each spatial feature, inverting the feature map to the tokens with this greedy search quantization cannot invert the feature map to the original tokens.

We define the problem of searching for the token sequence in multi-scale IAR as an optimization problem. Given a target feature map $f \in \mathbb{R}^{H_K \times W_K \times C}$, we seek the optimal multi-scale token combination $\{t_k\}_{k=1}^K$ that minimizes the reconstruction error:

$$\min_{\{t_k\}_{k=1}^K} \left\| f - \hat{f}(\{t_k\}_{k=1}^K) \right\|_2^2, \tag{8}$$

where $\hat{f}(\{t_k\}_{k=1}^K)$ denotes the reconstructed feature map obtained by dequantizing and aggregating tokens across all scales as defined in Equation (7). To solve this problem, we propose an **optimized quantization** algorithm to search for a token combination across all scales that can best represent a given feature map. For each element in the token map, we initialize $N$ logits corresponding to $N$ entries in the codebook. An estimated feature map is then calculated according to the logits. Then we employ the gradient descent algorithm to minimize the distance between the estimated and target feature map. The intuition to detect images generated by VAR is the following: for a feature map generated by VAR, our algorithm enables the originally generated tokens to gradually have higher logit values with more iterations, and finally reduces the QuantLoss significantly. Any feature map not generated by VAR cannot be easily represented by tokens from the codebook, so the QuantLoss remains high even after optimization. We present the detailed algorithm in Appendix A.

Overall, once $D^{-1}$ and optimized $Q$ are obtained, the resulting QuantLoss serves as a powerful distinguishing signal: images generated by a IAR $M$ consistently exhibit significantly lower loss than those not originating from the model, enabling highly reliable provenance attribution.

### 3.3.2 ENCLOSS: PROVENANCE SIGNAL BASED ON DECODER INVERSION LOSS

As a second complementary provenance signal, we propose a feature we call *EncLoss*, which can be combined with the QuantLoss to provide more reliable data provenance. This feature is based on our observation that the decoding during generation ($f_Z \xrightarrow{D} x_Z$) maps $f_Z$ which lies in a low-

dimensional latent space to $x_Z$ which lies in a high-dimensional pixel space. Therefore, if we compress a generated image $x_Z$ back to $f_Z$ with an ideal inverse decoder $D^{-1}$, there is no information loss in this process. In contrast, if a natural image or an image generated by another model is projected from pixel space to latent space with $D^{-1}$, there is a non-negligible loss due to information compression. Using this observation, we apply inverse decoding and decoding to a given image to capture this signal, and quantify the EncLoss as $\mathcal{L}_{\text{Enc}}$ as

$$\mathcal{L}_{\text{Enc}} = \|Rec\,(x) - x\|_2, \tag{9}$$

where $Rec\,(x) := D\big(D^{-1}(x)\big)$. However, we note that this loss is not only related to the data source, but also to the *complexity* of the image. Specifically, a natural image with low complexity contains low information density, and thus also has low EncLoss when encoded into the latent space. To address the potential false positive cases caused by the low-complexity images, we calculate a calibration factor for the EncLoss inspired by AEDR (Wang et al., 2025a). Concretely, we invert the image twice, where the second EncLoss serves as an estimation of the inherent image complexity. The calibrated EncLoss can be formalized as follows:

$$\mathcal{L}_{\text{Enc}}^{\text{Cal}} = \frac{\|Rec\,(x) - x\|_2}{\|Rec\,(Rec\,(x)) - Rec\,(x)\|_2}. \tag{10}$$

**Our Final Combined Provenance Signals.** Finally, we combine the QuantLoss $\mathcal{L}_{\text{Quant}}$ and the calibrated EncLoss $\mathcal{L}_{\text{Enc}}^{\text{Cal}}$ to obtain a stronger signal for provenance. Since $\mathcal{L}_{\text{Enc}}^{\text{Cal}}$ is a ratio of errors, we design the combined loss $\mathcal{L}_{\text{Comb}}$ as a product of $\mathcal{L}_{\text{Quant}}$ and $\mathcal{L}_{\text{Enc}}^{\text{Cal}}$:

$$\mathcal{L}_{\text{Comb}} = \mathcal{L}_{\text{Quant}} \times \mathcal{L}_{\text{Enc}}^{\text{Cal}}. \tag{11}$$

## 4 EMPIRICAL EVALUATION

### 4.1 EXPERIMENTAL SETUP

**Models.** We evaluate our method on a diverse set of state-of-the-art IAR models for image generation. This includes **next-token prediction models**, such as LlamaGen (Sun et al., 2024) and Taming (Esser et al., 2021); a **random-order prediction model**, RAR (Yu et al., 2024b); and **next-scale prediction models**, such as VAR (Tian et al., 2024) and Infinity (Han et al., 2025), the latter of which is bit-wise and supports high-resolution generation. To further demonstrate the generality of our approach beyond autoregressive models, we also report results on the **vector-quantized diffusion model** VQ-Diffusion (Tang et al., 2023).

**Datasets.** We construct several evaluation datasets, which consist of real or generated images. Real images are obtained from the validation sets of standard benchmarks (1,000 images each), namely ImageNet (Deng et al., 2009), LAION (Schuhmann et al., 2022), and MS-COCO (Lin et al., 2014). The generated images are obtained by generating 1,000 images using each of the previously mentioned models. For finetuning, we use a distinct dataset generated for each tested model. For more details on finetuning, please refer to Appendix C.

**Metrics.** We report the true positive rate at 1% false positive rate (TPR@1%FPR) as our primary evaluation metric. We aim to minimize false accusations, avoiding wrongly attributing images to a model that did not generate them, while still measuring detection performance effectively.

**Baselines.** We compare against several baselines: a **naïve reconstruction-loss baseline** where images with lower autoencoder reconstruction losses are detected as belonging images, **Latent-Tracer** (Wang et al., 2024), and **AEDR** (Wang et al., 2025a). For more details on our baselines setup, please refer to Appendix D.

### 4.2 EFFECTIVENESS OF OUR DATA PROVENANCE FOR IARS

Table 1 and the extended versions Table A4 and Table A5 summarize the effectiveness of our method over all models and datasets. For a given target model denoted in the first column, each column in the table represents a different task, where 1,000 images generated by the target model (belonging set) are distinguished from 1,000 images from a single non-belonging source. Evaluated non-belonging sources cover both three natural image datasets and five image datasets generated by other models.

Table 1: **TPR@1%FPR (%) of our method and the baselines**. The first column indicates the original model that has generated the belonging images, the heading of the other columns specifies the natural datasets or generators from which the non-belonging images are obtained. Our method is instantiated with the best-performing set of signals from Section 3.3 for each original model.

| Model | Method | Natural | | | Generated | | | | | |
|---|---|---|---|---|---|---|---|---|---|---|
| | | ImageNet | LAION | MS-COCO | LlamaGen | RAR | Taming | VAR | Infinity | VQDiff |
| LlamaGen | Reconstruction | 33.6 | 34.0 | 44.3 | - | 39.7 | 4.3 | 45.7 | 70.0 | 63.0 |
| | LatentTracer | 93.5 | 89.2 | 97.9 | - | 96.3 | 80.7 | 96.9 | 99.0 | 98.7 |
| | AEDR | 50.9 | 55.3 | 50.5 | - | 59.5 | 57.7 | 67.0 | 70.7 | 68.1 |
| | Ours | **100.0** | **100.0** | **100.0** | - | **100.0** | **100.0** | **100.0** | **100.0** | **100.0** |
| RAR | Reconstruction | 3.8 | 4.1 | 7.4 | 0.8 | - | 0.1 | 5.7 | 18.1 | 18.8 |
| | LatentTracer | 6.0 | 6.1 | 15.2 | 0.4 | - | 0.0 | 9.3 | 24.6 | 26.9 |
| | AEDR | 29.5 | 16.6 | 36.6 | 10.6 | - | 2.3 | 35.9 | 49.9 | 27.6 |
| | Ours | **100.0** | **100.0** | **100.0** | **99.9** | - | **99.9** | **100.0** | **100.0** | **100.0** |
| Taming | Reconstruction | 27.5 | 21.5 | 27.6 | 10.1 | 18.9 | - | 27.7 | 39.0 | 46.1 |
| | LatentTracer | 73.0 | 61.0 | 75.9 | 36.4 | 66.8 | - | 76.0 | 85.4 | 87.4 |
| | AEDR | 80.4 | 82.5 | 81.9 | 70.7 | 80.7 | - | 78.1 | 91.9 | 87.5 |
| | Ours | **100.0** | **100.0** | **100.0** | **100.0** | **100.0** | - | **100.0** | **100.0** | **100.0** |
| VAR | Reconstruction | 1.4 | 1.4 | 3.6 | 0.1 | 1.6 | 0.0 | - | 5.9 | 5.9 |
| | LatentTracer | 3.9 | 1.3 | 12.0 | 0.2 | 5.6 | 0.1 | - | 15.4 | 15.3 |
| | AEDR | 29.1 | 15.7 | 50.6 | 14.7 | 28.3 | 14.0 | - | 37.5 | 50.8 |
| | Ours | **100.0** | **99.2** | **100.0** | **99.2** | **100.0** | **100.0** | - | **100.0** | **100.0** |
| Infinity | Reconstruction | 0.0 | 0.2 | 0.8 | 0.0 | 0.0 | 0.0 | 0.2 | - | 0.3 |
| | LatentTracer | 0.0 | 0.0 | 10.9 | 31.7 | 0.2 | 0.0 | 5.8 | - | 5.3 |
| | AEDR | 1.6 | 18.9 | 56.2 | 1.4 | 3.0 | 1.5 | 12.8 | - | 8.4 |
| | Ours | **99.4** | **85.6** | **99.4** | **99.2** | **99.5** | **99.1** | **99.4** | - | **99.4** |
| VQDiff | Reconstruction | 17.2 | 8.8 | 24.3 | 6.3 | 21.8 | 1.6 | 21.2 | 43.0 | - |
| | LatentTracer | 97.7 | 93.8 | 98.4 | 97.3 | 97.9 | 93.6 | 98.5 | 98.6 | - |
| | AEDR | 89.7 | 51.4 | 90.0 | 79.8 | 93.6 | 77.2 | 87.5 | 83.6 | - |
| | Ours | **100.0** | **99.4** | **100.0** | **99.9** | **100.0** | **99.9** | **100.0** | **100.0** | - |

Table 2: **Robustness against common image post-processing methods on RAR.** Non-belonging images are from the ImageNet dataset, and we use QuantLoss to instantiate our method.

| Method | Attacks | | | | | | |
|---|---|---|---|---|---|---|---|
| | Noise (0.05) | Kernel (9) | JPEG (60) | Brightness (1.6) | Contrast (2.0) | Saturation (2.0) | Resize (0.5) |
| LatentTracer | 3.4 | 4.7 | 4.8 | 2.3 | 3.0 | 3.6 | 2.2 |
| Reconstruction | 2.3 | 3.0 | 3.6 | 1.4 | 1.6 | 3.1 | 1.0 |
| AEDR | 7.3 | 11.4 | 8.9 | 1.9 | 1.4 | 9.5 | 0.2 |
| Ours (w/o Aug) | 60.4 | 74.9 | 91.7 | 67.9 | 45.7 | 97.4 | 88.5 |
| Ours (w/ Aug) | **87.8** | **80.5** | **96.1** | **92.3** | **91.1** | **99.2** | **98.4** |

We first observe that the naïve reconstruction baseline is not effective in detecting which model a given image was generated by. Although LatentTracer can obtain relatively good performance on LlamaGen and VQ Diffusion when compared with other baseline methods, it fails for RAR, VAR, and Infinity. While AEDR yields slightly better results than LatentTracer on those models, its overall performance over the diverse set of models falls short. In contrast, our method yields perfect or near-perfect results, *i.e.,* around 100% TPR over all models and datasets, highlighting its strength for practical data provenance tracing.

Additionally, our method only requires one-time finetuning to obtain the inverse decoder, which can be used to evaluate provenance on an unlimited number of images. Notably, our QuantLoss operates in the latent space of the autoencoder and does not require decoding into the full image, which allows for efficient provenance nearly $2 \times$ faster than the reconstruction baseline and nearly $4 \times$ faster than the AEDR. We show in Table A3, that for most IARs, our method achieves the fastest data provenance, with a running time of less than 10 milliseconds.

## 4.3 ROBUSTNESS EVALUATION AGAINST IMAGE POST-PROCESSING

In practical provenance tracing applications, the original images might be modified through JPEG compression, resizing, or other post-processing operations, which can reduce provenance signals and make reliable tracing more challenging. We analyze the robustness of our proposed framework

Table 3: **Contribution of the components in our method**. We present TPR@1%FPR of different signals on different models. We denote the optimized quantization as *QuantLoss Opt*. The best instantiations of our framework for each model are highlighted in green, which corresponds to the results shown for our method in Table 1.

| Model | Method | Natural | | | Generated | | | | | |
|---|---|---|---|---|---|---|---|---|---|---|
| | | ImageNet | LAION | MS-COCO | LlamaGen | RAR | Taming | VAR | Infinity | VQDiff |
| LlamaGen | Ours (*QuantLoss*) | 100.0 | 99.8 | 100.0 | - | 100.0 | 100.0 | 100.0 | 100.0 | 100.0 |
| | Ours (*EncLoss*) | 100.0 | 100.0 | 100.0 | - | 100.0 | 100.0 | 100.0 | 100.0 | 100.0 |
| | Ours (*QuantLoss × EncLoss*) | **100.0** | **100.0** | **100.0** | - | **100.0** | **100.0** | **100.0** | **100.0** | **100.0** |
| RAR | Ours (*QuantLoss*) | 99.9 | 99.8 | 99.9 | 99.8 | - | 99.2 | 100.0 | 100.0 | 99.8 |
| | Ours (*EncLoss*) | 98.2 | 98.0 | 98.9 | 93.5 | - | 91.9 | 96.6 | 99.5 | 99.7 |
| | Ours (*QuantLoss × EncLoss*) | **100.0** | **100.0** | **100.0** | **99.9** | - | **99.9** | **100.0** | **100.0** | **100.0** |
| Taming | Ours (*QuantLoss*) | 99.6 | 88.8 | 99.6 | 96.2 | 99.6 | - | 99.5 | 99.8 | 99.5 |
| | Ours (*EncLoss*) | 100.0 | 100.0 | 100.0 | 100.0 | 100.0 | - | 100.0 | 100.0 | 100.0 |
| | Ours (*QuantLoss × EncLoss*) | **100.0** | **100.0** | **100.0** | **100.0** | **100.0** | - | **100.0** | **100.0** | **100.0** |
| VAR | Ours (*QuantLoss*) | 0.4 | 0.0 | 10.0 | 0.0 | 4.5 | 0.0 | - | 13.4 | 1.6 |
| | Ours (*QuantLoss Opt*) | 95.0 | 92.9 | 94.4 | 89.8 | 94.5 | 88.4 | - | 95.7 | 95.2 |
| | Ours (*EncLoss*) | 100.0 | 96.8 | 100.0 | 98.1 | 100.0 | 99.7 | - | 100.0 | 100.0 |
| | Ours (*QuantLoss Opt × EncLoss*) | **100.0** | **99.2** | **100.0** | **99.2** | **100.0** | **100.0** | - | **100.0** | **100.0** |
| Infinity | Ours (*QuantLoss*) | **99.4** | **85.6** | **99.4** | **99.2** | **99.5** | **99.1** | **99.4** | - | **99.4** |
| | Ours (*EncLoss*) | 0.0 | 94.9 | 98.9 | 1.4 | 0.6 | 0.4 | 11.8 | - | 35.1 |
| | Ours (*QuantLoss × EncLoss*) | 0.0 | 98.2 | 100.0 | 9.1 | 3.4 | 1.1 | 57.3 | - | 76.6 |
| VQDiff | Ours (*QuantLoss*) | 92.1 | 43.3 | 99.1 | 96.8 | 97.6 | 85.8 | 95.7 | 99.1 | - |
| | Ours (*EncLoss*) | 100.0 | **100.0** | 100.0 | 99.7 | 100.0 | **100.0** | 100.0 | 100.0 | - |
| | Ours (*QuantLoss × EncLoss*) | **100.0** | 99.4 | **100.0** | **99.9** | **100.0** | 99.9 | **100.0** | **100.0** | - |

Table 4: **Effectiveness of decoder inversion with or without finetuning the encoder**. We use the the best instantiation of our framework following Table 1. We show TPR@1%FPR across different datasets and models.

| Model | Method | Natural | | | Generated | | | | | |
|---|---|---|---|---|---|---|---|---|---|---|
| | | ImageNet | LAION | MS-COCO | LlamaGen | RAR | Taming | VAR | Infinity | VQDiff |
| LlamaGen | Ours (*Original Encoder*) | 99.9 | 99.6 | 99.9 | - | 99.9 | 99.6 | 99.9 | 99.9 | 100.0 |
| | Ours (*Inverse Decoder*) | 100.0 | 100.0 | 100.0 | - | 100.0 | 100.0 | 100.0 | 100.0 | 100.0 |
| RAR | Ours (*Original Encoder*) | 6.2 | 9.1 | 10.0 | 1.7 | - | 0.5 | 3.1 | 13.4 | 21.8 |
| | Ours (*Inverse Decoder*) | 100.0 | 100.0 | 100.0 | 99.9 | - | 99.9 | 100.0 | 100.0 | 100.0 |
| Taming | Ours (*Original Encoder*) | 15.3 | 15.7 | 15.3 | 7.8 | 8.8 | - | 12.5 | 13.7 | 19.2 |
| | Ours (*Inverse Decoder*) | 100.0 | 100.0 | 100.0 | 100.0 | 100.0 | - | 100.0 | 100.0 | 100.0 |
| VAR | Ours (*Original Encoder*) | 2.7 | 3.5 | 5.4 | 8.4 | 3.2 | 6.3 | - | 8.4 | 7.8 |
| | Ours (*Inverse Decoder*) | 100.0 | 99.2 | 100.0 | 99.2 | 100.0 | 100.0 | - | 100.0 | 100.0 |
| Infinity | Ours (*Original Encoder*) | 0.0 | 0.0 | 16.6 | 0.0 | 1.3 | 0.0 | 2.8 | - | 0.0 |
| | Ours (*Inverse Decoder*) | 99.4 | 85.6 | 99.4 | 99.2 | 99.5 | 99.1 | 99.4 | - | 99.4 |
| VQDiff | Ours (*Original Encoder*) | 86.1 | 33.2 | 82.9 | 78.8 | 95.7 | 65.8 | 83.3 | 86.1 | - |
| | Ours (*Inverse Decoder*) | 100.0 | 99.4 | 100.0 | 99.9 | 100.0 | 99.9 | 100.0 | 100.0 | - |

against common image post-processing methods. We analyze the robustness of RAR in Table 2 and provide a more extensive evaluation for robustness on additional models in Appendix H. We detail the analyzed attacks in Appendix C and provide the respective strengths in brackets in Table 2. We find that the baselines quickly break against common image post-processing transformations, while our QuantLoss allows for reliable attribution. Our framework also enables finetuning the inversion $D^{-1}$ with augmentations to further improve the robustness of attribution against the image post-processing operations. We note that for the setting where common image processing exists, the best instantiation of our method is QuantLoss. In Appendix H, we show the reason why our QuantLoss allows for more robust provenance than EncLoss, specifically when trained with augmentations.

## 4.4 ABLATION STUDIES

**Effectiveness of the Framework Components.** While, for the results in Table 1, we instantiate our framework with the best per-model combination of signals from Section 3.3, in Table 3, we ablate the impact of the individual signals. Concretely, we study the following three combinations: *QuantLoss* only uses the QuantLoss from Equation (5), *EncLoss* relies on the EncLoss from Equation (10), and *QuantLoss × EncLoss* uses the combined loss from Equation (11). For VAR, we additionally integrate the optimization step from Algorithm 3. Our results show that, for most model and dataset pairs, combining both signals yields perfect or near-perfect results, *i.e.,* 100% TPR at practical

Table 5: **Effectiveness of EncLoss calibration**. We show TPR@1%FPR for attributing belonging images v.s. non-belonging images from different real datasets or generative models.

| Model | Method | Natural | | | Generated | | | | | |
|---|---|---|---|---|---|---|---|---|---|---|
| | | ImageNet | LAION | MS-COCO | LlamaGen | RAR | Taming | VAR | Infinity | VQDiff |
| LlamaGen | EncLoss *(w/o Calibration)* | 19.0 | 23.8 | 34.3 | - | 26.4 | 2.8 | 32.8 | 63.4 | 54.9 |
| | EncLoss *(w/ Calibration)* | 100.0 | 100.0 | 100.0 | - | 100.0 | 100.0 | 100.0 | 100.0 | 100.0 |
| RAR | EncLoss *(w/o Calibration)* | 22.6 | 21.2 | 27.3 | 5.1 | - | 2.5 | 26.0 | 47.9 | 44.0 |
| | EncLoss *(w/ Calibration)* | 98.2 | 98.0 | 98.9 | 93.5 | - | 91.9 | 96.6 | 99.5 | 99.7 |
| Taming | EncLoss *(w/o Calibration)* | 53.7 | 39.1 | 49.8 | 29.5 | 43.9 | - | 52.2 | 65.2 | 70.9 |
| | EncLoss *(w/ Calibration)* | 100.0 | 100.0 | 100.0 | 100.0 | 100.0 | - | 100.0 | 100.0 | 100.0 |
| VAR | EncLoss *(w/o Calibration)* | 17.0 | 15.8 | 31.8 | 6.1 | 21.7 | 1.4 | - | 41.4 | 41.5 |
| | EncLoss *(w/ Calibration)* | 100.0 | 96.8 | 100.0 | 98.1 | 100.0 | 99.7 | - | 100.0 | 100.0 |
| Infinity | EncLoss *(w/o Calibration)* | 0.3 | 2.5 | 4.5 | 0.1 | 0.2 | 0.0 | 0.8 | - | 1.3 |
| | EncLoss *(w/ Calibration)* | 0.0 | 94.9 | 98.9 | 1.4 | 0.6 | 0.4 | 11.8 | - | 35.1 |
| VQDiff | EncLoss *(w/o Calibration)* | 15.7 | 5.4 | 24.6 | 15.5 | 14.8 | 3.5 | 21.9 | 34.3 | - |
| | EncLoss *(w/ Calibration)* | 100.0 | 100.0 | 100.0 | 99.7 | 100.0 | 100.0 | 100.0 | 100.0 | - |

Table 6: **Hyperparameter analysis for optimized token search (Algorithm 3).** We evaluate on VAR model, using VAR-generated images as the belonging image, and ImageNet as the non-belonging image. The evaluated metric is TPR@1%FPR (%). *Init w/ Orig. Quant.* denotes whether our algorithm is initialized with the original quantization in VAR. Default parameters in Bold.

| Baseline | Number of Iterations | | | | | | Learning Rate | | | | | Init w/ Orig. Quant. | |
|---|---|---|---|---|---|---|---|---|---|---|---|---|---|
| | 100 | 400 | 1000 | **1200** | 1400 | 1600 | 0.01 | 0.05 | **0.1** | 0.2 | 0.5 | No | **Yes** |
| 0.4 | 87.5 | 91.0 | 95.4 | 95.0 | 93.8 | 92.2 | 43.0 | 95.2 | 95.0 | 94.2 | 92.8 | 94.3 | 95.0 |

detection thresholds. We observe that for VAR, including the additional optimization step boosts the combined signal by, on average, roughly 10% and achieves perfect detection.

**Decoder Inversion.** We also ablate the role of relying on the inverted decoder instead of the IARs' original encoders for provenance tracing in Table 4. Our results highlight the importance of decoder inversion: while *e.g.,* for RAR, the combined loss can initially only partly attribute belonging images, after finetuning we achieve close to 100% TPR@1%FPR.

**EncLoss Calibration.** We evaluate the impact of our calibration strategy for the EncLoss signal in Table 5, comparing the uncalibrated reconstruction loss (Equation (9)) against our calibrated version (Equation (10)). Without calibration, performance is moderate because low-complexity natural images exhibit reconstruction loss similar to generated images. The calibration normalizes by image complexity, achieving near-perfect detection for most models.

**Hyperparameter Analysis for Optimized Quantization.** Table 6 analyzes hyperparameters for our optimized token search on VAR. Optimal performance (95.0-95.4% TPR@1%FPR) occurs with 1,000-1,400 iterations and learning rate 0.1. Notably, our method still achieves 87.5%TPR @1% FPR with only 100 iterations, which can reduce the runtime from 8.24s/image to 0.57s/image. In addition, initializing with VAR's original quantization provides a modest boost (95.0% vs. 94.3%).

## 5 CONCLUSIONS

We introduced the first model-agnostic, post-hoc framework for robust provenance tracing of IAR-generated images. Our approach exploits the unique quantization artifacts left by the tokenization process in IARs, distinguishing generated content even in the absence of visible differences or explicitly added watermarks. To strengthen the evidence that a given image was generated by a particular IAR, we design additional provenance features that increase the signals from quantization-artifacts and leverage additional signals from the encoding process. We show that our framework achieves near-perfect detection across a wide range of state-of-the-art IARs. Notably, it operates without requiring any architectural changes or access to the generation process, making it broadly applicable to existing, previously published, and unmarked content. Our results provide a practical and scalable solution for responsible deployment and post-hoc auditing of autoregressive generative models in real-world scenarios.

## ACKNOWLEDGMENT

We would like to acknowledge our sponsors, who support our research with financial and in-kind contributions: OpenAI and G-Research. We also thank members of the SprintML group for their feedback.

## ETHICS STATEMENT

This work addresses critical societal challenges posed by increasingly realistic AI-generated imagery, including misinformation, fraud, and harmful content dissemination. Our post-hoc provenance method serves as a defensive technology that enhances transparency and accountability in AI-generated content without compromising the quality or utility of generative models. Since our method achieves nearly 100%TPR at only 1%FPR, it has a very low risk of making false accusations. We believe the benefits of enabling reliable source attribution for combating synthetic media misuse outweigh the potential risks.

## REPRODUCIBILITY STATEMENT

We provide comprehensive implementation details together with open-sourced code to ensure reproducibility of our results. All experimental configurations, including hyperparameters for finetuning the inverse decoder across six different open-source models (LlamaGen, RAR, Taming, VAR, Infinity, VQ-Diffusion), are detailed in Appendix C with specific learning rates, batch sizes, and training schedules. Our evaluation also includes three well-known image datasets (ImageNet, LAION, MS-COCO validation sets) that are also open-source. The optimized quantization algorithm for multi-scale models is provided in detail in Algorithm 3, and robustness evaluation protocols with specific attack parameters are also documented in Appendix C. All experiments were conducted on standard hardware (NVIDIA A40 GPUs) with specified software versions.

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

## A    Detailed Algorithm of Optimized Token Search

We present the original quantization algorithm for VAR in Algorithm 2, the original dequantization algorithm for VAR in Algorithm 1, and the detailed algorithm of the optimized token search for VAR in Algorithm 3. The part introducing errors due to scalewise structure for VAR quantization in Algorithm 2 is marked in red. The common procedures for original VAR dequantization (Algorithm 1) and our approach (Algorithm 3) are marked in blue.

We observe that in the dequantization process in VAR (Algorithm 1), the representations in all scales are upscaled and added to the final feature map (row 5-6). However, during the quantization process (Algorithm 2), the feature map is considered as a whole during the codebook lookup (row 6). Therefore, if we quantize a feature map of a generated image on scale $k$, all the token representations from scales $> k$ are also part of the feature map during this lookup process, which leads to an error of the current-scale quantization.

As shown by Algorithm 3, the goal of our algorithm is to search for a scalewise token combination from the codebook that has a minimal distance to a given feature map by backpropagation. For each element in the token map, we initialize $N$ logits corresponding to $N$ entries in the codebook (row 2). An estimated feature map is then calculated according to the logits (row 3-11). Then we employ the gradient descent algorithm to minimize the distance between the estimated and target feature map (row 12-13). The intuition to detect images generated by VAR is the following: For a feature map generated by VAR, our algorithm enables the originally generated tokens to gradually have higher logit values with more iterations, and finally reduces the QuantLoss largely. Any feature map not generated by VAR cannot be easily represented by tokens from the codebook, so the QuantLoss remains high even after this optimization.

---

**Algorithm 1** Original Dequantization for VAR

**Inputs:**  multi-scale token maps $t$, codebook $Z$
**Hyperparameters:**  number of scales $K$, resolutions $\{(h_k, w_k)\}_{k=1}^{K}$

1: $\hat{f} \leftarrow 0$               ▷ Initialize reconstructed feature map
2: **for** $k \leftarrow 1$ to $K$ **do**           ▷ Iterate through all scales
3:    $t_k \leftarrow \text{QUEUE\_POP}(t)$       ▷ Obtain tokens from a given scale
4:    $z_k \leftarrow \text{LOOKUP}(Z, t_k)$     ▷ Look up codebook vectors for tokens
5:    $z_k \leftarrow \text{INTERPOLATE}(z_k, h_K, w_K)$     ▷ Upscale to full resolution
6:    $\hat{f} \leftarrow \hat{f} + \phi_k(z_k)$    ▷ Add processed features to reconstruction
7: **return** $\hat{f}$             ▷ Return reconstructed image

---

**Algorithm 2** Original Quantization for VAR

**Inputs:**  image $x$, encoder $E$, quantizer $Q$, codebook $Z$
**Hyperparameters:**  number of scales $K$, resolutions $\{(h_k, w_k)\}_{k=1}^{K}$

1: $f \leftarrow E(x)$           ▷ Encode image to get the feature map
2: $t \leftarrow [\,]$         ▷ Initialize empty queue for multi-scale tokens
3: **for** $k \leftarrow 1$ to $K$ **do**          ▷ Iterate through all scales
4:    $r_k \leftarrow Q(\text{INTERPOLATE}(f, h_k, w_k))$   ▷ Quantize interpolated features to current scale
5:    $t \leftarrow \text{QUEUE\_PUSH}(t, r_k)$     ▷ Add tokens to the token map
6:    $z_k \leftarrow \text{LOOKUP}(Z, r_k)$     ▷ Look up codebook vectors for tokens
7:    $z_k \leftarrow \text{INTERPOLATE}(z_k, h_K, w_K)$     ▷ Upscale to full resolution
8:    $f \leftarrow f - \phi_k(z_k)$    ▷ Subtract processed features from residual
9: **return** $t$           ▷ Return multi-scale tokens

---

## B    Additional Related Work

**LlamaGen** (Sun et al., 2024) demonstrated that vanilla autoregressive models, without inductive biases on visual signals, can achieve state-of-the-art image generation performance if scaling properly. There are three keys to its success. (1) A well-designed image compressor, which balances

---

**Algorithm 3** Optimized Quantization for VAR

---

**Inputs:** image $x$, inverse decoder $D^{-1}$, codebook $Z = \{z_1, ..., z_N\}$ with a size of $N$, gradient descent algorithm GD($\cdot$)

**Hyperparameters:** number of scales $K$, resolutions $\{(h_k, w_k)\}_{k=1}^K$, number of iterations $N_{iters}$

1: $f \leftarrow D^{-1}(x)$        $\triangleright$ Encode image with the inverse decoder to get the feature map
2: $L \leftarrow \{l_k\}_{k=1}^K$        $\triangleright$ Initialize the token maps logits. $l_k$ has a shape of $(h_k, w_k, N)$
3: **for** $n_{iters} \leftarrow 1$ to $N_{iters}$ **do**        $\triangleright$ Optimization iterations
4:      $\hat{f} \leftarrow 0$        $\triangleright$ Initialize the estimated feature map
5:      **for** $k \leftarrow 1$ to $K$ **do**        $\triangleright$ Iterate through all scales to calculate features on each scale
6:          **for** $i \leftarrow 1$ to $h_k$ **do**        $\triangleright$ Iterate through features in the current scale
7:              **for** $j \leftarrow 1$ to $w_k$ **do**
8:                 $p \leftarrow$ SOFTMAX$(l[k][i][j])$    $\triangleright$ Calculate the probabilities of all codebook entries
9:                 $z[k][i][j] \leftarrow \sum_{t=1}^N p_t \cdot z_t$    $\triangleright$ Calculate the feature averaged on the probabilities
10:          $z[k] \leftarrow$ INTERPOLATE$(z[k], h_K, w_K)$      $\triangleright$ Upscale the feature map to full resolution
11:          $\hat{f} \leftarrow \hat{f} + \phi_k(z[k])$     $\triangleright$ Process with convolution and add to the estimated feature map
12:      $E = \frac{1}{h_K w_K} \sum_{i=1}^{h_K} \sum_{j=1}^{w_K} (\hat{f}[i][j] - f[i][j])^2$
13:      $L \leftarrow$ GD$(L, E)$        $\triangleright$ Perform gradient descent on the logits $L$ to minimize $E$
14: $t \leftarrow \{\{\{\arg\max_n l[k][i][j]\}_{j=1}^{w_k}\}_{i=1}^{h_k}\}_{k=1}^K$     $\triangleright$ Calculate the final tokens by taking highest logits
15: **return** $t$

---

the trade-off between image quality and codebook utilization by opting for the down-sample ratio of $p = 16$. (2) A scalable image generation model developed based on the Llama architecture (Touvron et al., 2023a;b) used for LLMs, and (3) high-quality training data, especially with the finetuning on 10 M high aesthetics quality images.

**Token Mismatch.** The first papers on watermarking already observed a mismatch between the generated tokens for a given image and the image's re-encoded tokens (Meintz et al., 2025; Kerner et al., 2025; Jovanović et al., 2025; Tong et al., 2025). The problem stems primary from the decoder-encoder pairs which are not trained to optimize for the token match (see the training optimization with the compound loss, for example, in VAR by Tian et al. (2024), Equation 5). The standard training only ensures small loss between the original and generated images as well as between the latent representations after encoding and before the decoding. An additional term is the reconstructed image quality, which is measured, for example, with the LPIPS (Zhang et al., 2018) or StyleGAN's discriminator loss (Karras et al., 2019). Despite the mismatch, the token-based watermarks from Meintz et al. (2025) and bit-wise watermark proposed by Kerner et al. (2025), were able to still provide a highly-robust detection of the generated content. The other line of work by Jovanović et al. (2025) and Tong et al. (2025) further propose to finetune the encoder-decoder or encoder-only, respectively, to compensate for the token-index reconstruction errors. We leverage the inherent property of IARs with their discrete codebook and the encoding errors by showing that the significantly higher errors for the natural images allow us to distinguish them from the generated images.

**Vector Quantization in IARs.** The token-based image generation in IARs has the underlying principle inherited from LLMs, where each next predicted token is represented as an index of one of the entries in the codebook. The codebook stores a collection of relatively small dimensional representation vectors which constitute building blocks of an image. The generated tokens are decoded to a full-dimensional image. Any image can be encoded to the token latent space. The encoder performs feature extraction through a multi-layer convolutional layers with down-sampling to the latent space, which results in a collection of encoded small dimensional representation vectors. These vectors are compared with the entries in the codebook to obtain the integer indices of the tokens.

## C IMPLEMENTATION DETAILS

**Finetuning.** Our finetuning of $D^{-1}$ follows the pipeline in Figure 2, where we first generate tokens with the corresponding AR model, embed them to the original feature map $f_Z$ and use the frozen

decoder $D$ to generate images $x_Z$. We detail the finetuning hyperparameters, such as the number of images, the batch size and learning rate for every model in Table A1. All experiments were conducted on a single NVIDIA A40 GPU with 48GB of memory.

Table A1: **Finetuning details for different models.**

| Method | Number of Finetuning Data | Epoch | Batch Size | Optimizer | Learning Rate | Scheduler | Scheduler Configuration |
|---|---|---|---|---|---|---|---|
| LlamaGen | 50000 | 25 | 8 | Adam | $1 \times 10^{-5}$ | StepLR | Gamma=0.9, Step=2 |
| Taming | 50000 | 50 | 8 | Adam | $5 \times 10^{-4}$ | StepLR | Gamma=0.9, Step=2 |
| RAR | 50000 | 50 | 8 | Adam | $5 \times 10^{-4}$ | StepLR | Gamma=0.9, Step=2 |
| VAR | 50000 | 10 | 16 | Adam | $5 \times 10^{-5}$ | StepLR | Gamma=0.9, Step=2 |
| Infinity | 10000 | 10 | 2 | Adam | $1 \times 10^{-5}$ | StepLR | Gamma=0.9, Step=2 |
| VQ Diffusion | 10000 | 50 | 16 | Adam | $5 \times 10^{-5}$ | StepLR | Gamma=0.9, Step=2 |

**Augmentations.** For the robustness evaluation in Table 2, we apply augmentations during the finetuning of RAR and Taming to improve the robustness against image post-processing methods. We progressively apply weak to strong augmentations during 50 epochs of finetuning, where a more detailed recipe can be found in Table A2.

Table A2: **Augmentation hyperparameters during finetuning.**

| Strength | Epochs | JPEG-Compression (Final Quality) | Gaussian Blur (Kernel Size) | Gaussian Noise (Standard Deviation) | Brightness (Factor) | Saturation (Factor) | Resize (Ratio) | Contrast (Factor) |
|---|---|---|---|---|---|---|---|---|
| None | 1-5 | - | - | - | - | - | - | - |
| Weak | 6-10 | [90, 85, 80] | [1, 3] | [0.005, 0.01, 0.02] | [1.0, 1.1, 1.2] | [1.0, 1.2, 1.5] | [0.9, 0.85, 0.8] | [1.0, 1.2, 1.5] |
| Medium | 11-30 | [80, 75, 70, 65]) | [3, 5] | [0.02, 0.03, 0.04] | [1.3, 1.4, 1.5] | [1.5, 1.7, 2.0] | [0.8, 0.75, 0.7] | [1.5, 1.7, 2.0] |
| Strong | 31-50 | [60, 55, 50] | [5, 7, 9] | [0.03, 0.04, 0.05] | [1.5, 1.7, 2.0] | [2.0, 2.2, 2.5] | [0.7, 0.6, 0.5] | [2.0, 2.2, 2.4] |

**Hyperparameters for Optimized Quantization.** For the optimized quantization of VAR, we use 1200 iterations with a learning rate of 0.1, batch size of 8, and the Adam optimizer. We use the original quantization in VAR (Algorithm 2) as initialization. We perform an analysis of the hyperparameters in Table 6.

**Robustness.** In Table 2 we evaluate the following methods: 1) **Noise:** Adds Gaussian noise with a std of 0.05 to the image, 2) **Kernel:** Application of a Gaussian Blur with kernel size of 9, 3) **JPEG:** 60% JPEG compression, 4) **Brightness:** Increasing the brightness to 1.6, 5) **Contrast:** Changing the contrast to 2.0, 6) **Saturation:** Increasing the Saturation to 2.0, 7) **Resizing:** Decreasing the resolution of the image to 50% of its original resolution. An extended analysis of the impact of the strength of each attack can be found in Appendix H.

# D    IMPLEMENTATION DETAILS FOR BASELINE METHODS

**Reconstruction.** For this naïve baseline, we compute the loss $\mathcal{L}_{\text{rec}} = \|x - x_1\|_2$ between the original image $x$ and its first reconstruction $x_1 = D(Q(Q^{-1}(D^{-1}(x))))$ and use it to decide wether the image was generated by the model or not.

**LatentTracer.** (Wang et al., 2024) We optimize for 100 iterations with the Adam optimizer. The learning rate is 0.01, which decays by 50% after 50 iterations. The feature map is initialized as the quantized feature map encoded by the encoder.

**AEDR.** We follow the method proposed by Wang et al. (2025a) and calculate the calibrated loss $\mathcal{L}_{\text{cal}} = \frac{\mathcal{L}_{\text{rec}_1}}{\mathcal{L}_{\text{rec}_2}}$ between a first image reconstruction $x_1 = D(Q(Q^{-1}(D^{-1}(x))))$ with the first reconstruction loss $\mathcal{L}_{\text{rec}_1} = \|x - x_1\|_2$ and the second image reconstruction $x_2 = D(Q(Q^{-1}(D^{-1}(x_1))))$ with the second reconstruction loss $\mathcal{L}_{\text{rec}_2} = \|x_1 - x_2\|_2$.

# E    EXPERIMENTAL ENVIRONMENT

**Hardware.** Our experiments are performed on Ubuntu 22.04, with Intel(R) Xeon(R) Gold 6330 CPU and NVIDIA A40 Graphics Card with 48 GB of memory.

**Software.** To run our experiments we used CUDA Version 12.5 and Python 3.12.4 with PyTorch 2.7.0.

## F  THE DISTRIBUTIONS OF DIFFERENT METHODS

We analyze the distributions of the best-performing signals from Table 1 in Figure A1. We compute the loss for **all** non-belonging datasets, *i.e.,* the generated and natural datasets and compare it to the loss of the belonging dataset. The different distributions are calculated for both the original encoder and our finetuned Inverse Decoder $D^{-1}$. Figure A1 clearly shows, that the Inverse Decoder is necessary to reduce the overlap between the belonging and non-belonging loss distributions. This results in our method achieving near 100% TPR@1%FPR for data provenance.

The Combined Loss distributions for most models show an increase of the non-belonging data loss, while it decreases slightly for the belonging data. This behavior is related to the EncLoss, which is based on the ratio between the first and second reconstruction, as we formulate in Equation (10). The ratio converges to 1 for belonging images, as the difference between the first reconstruction loss and second reconstruction loss decreases. Similar for non-belonging images the second reconstruction loss decreases. However the first reconstruction loss stays consistent, as the image does not originate from the models codebook. This leads to an overall higher loss ratio and a higher Combined Loss.

## G  RUNNING TIME COMPARISON

**Running Time.** We compare the running time to determine our QuantLoss with the given baselines. As shown in table A3, after finetuning, our method is by far the fastest, followed by Reconstruction, then AEDR and finally LatentTracer, which with a running time of multiple seconds is the slowest method. We also estimate the pre-training time of different models. Notably, our inverse decoder finetuning is a relatively small overhead compared to the model pre-training stage. For example, the finetuning time is less than 0.05% compared to the pre-training time for LlamaGen.

Table A3: **Running time comparison.** We instantiate our method as only using QuantLoss for LlamaGen, RAR, Taming, VQ-Diffusion and Infinity, while using QuantLoss Opt for VAR. We also include an estimation of model pre-training time for each IAR.

| Model | Stage | LatentTracer | Reconstruction | AEDR | Ours (*QuantLoss*) |
|---|---|---|---|---|---|
| LlamaGen | Model Pre-training (hours) | >18000 | >18000 | >18000 | >18000 |
| | $D^{-1}$ Finetuning (hours) | - | - | - | 8.6 |
| | Attribution (second/sample) | 5.305 | 0.015 | 0.030 | 0.009 |
| RAR | Model Pre-training (hours) | >20000 | >20000 | >20000 | >20000 |
| | $D^{-1}$ Finetuning (hours) | - | - | - | 31.9 |
| | Attribution (second/sample) | 2.359 | 0.014 | 0.028 | 0.009 |
| Taming | Model Pre-training (hours) | >20000 | >20000 | >20000 | >20000 |
| | Pre-training (hours) | - | - | - | 42.1 |
| | Attribution (second/sample) | 3.674 | 0.013 | 0.024 | 0.006 |
| VQ-Diffusion | Model Pre-training (hours) | >10000 | >10000 | >10000 | >10000 |
| | $D^{-1}$ Finetuning (hours) | - | - | - | 4.9 |
| | Attribution (second/sample) | 3.112 | 0.022 | 0.043 | 0.008 |
| Infinity | Model Pre-training (hours) | >50000 | >50000 | >50000 | >50000 |
| | $D^{-1}$ Finetuning (hours) | - | - | - | 14.9 |
| | Attribution (second/sample) | 84.897 | 0.202 | 0.776 | 0.197 |
| VAR | Model Pre-training (hours) | >20000 | >20000 | >20000 | >20000 |
| | $D^{-1}$ Finetuning (hours) | - | - | - | 10.8 |
| | Attribution (second/sample) | 4.653 | 0.016 | 0.031 | 8.249 |

## H  EXTENDED ROBUSTNESS EVALUATION

We further analyze the robustness of our framework against different attack strengths in Figure A2 (RAR) and Figure A3 (Taming). The results show that our proposed attribution with QuantLoss

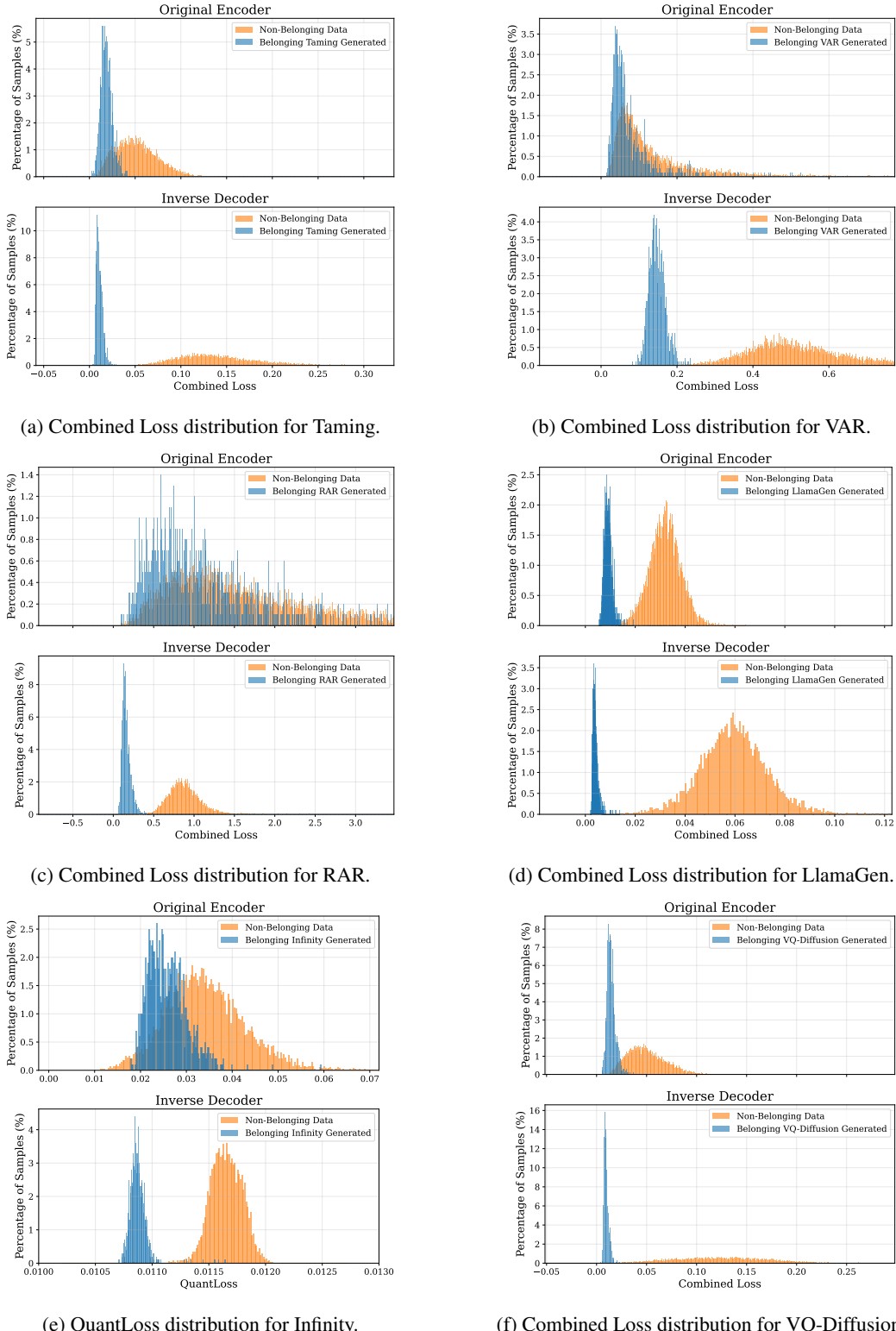

Figure A1: Distribution for Combined Loss for different models for the original encoder and the finetuned Inverse Decoder.

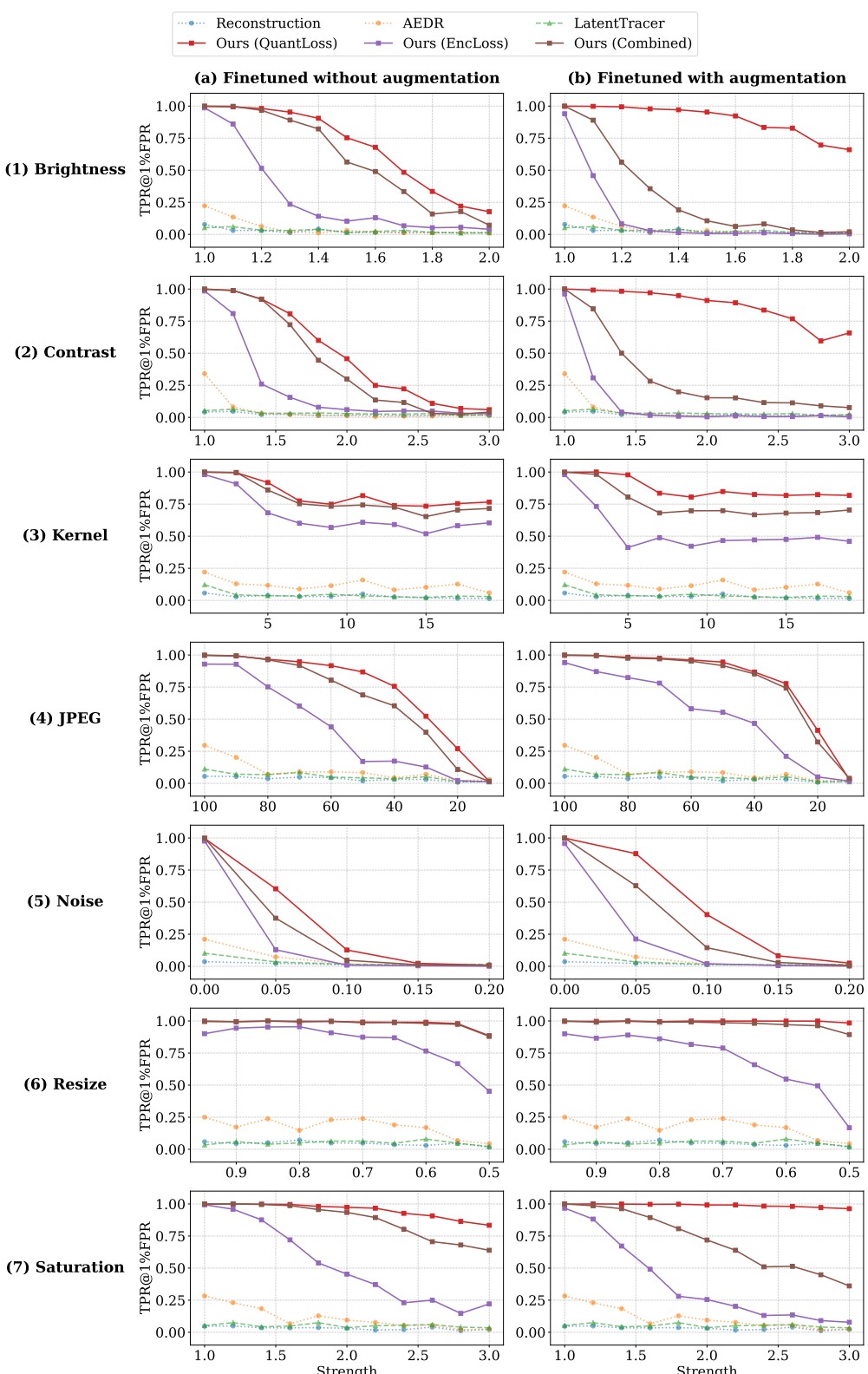

Figure A2: **Robustness Test for RAR on 7 common image post-processing techniques.**

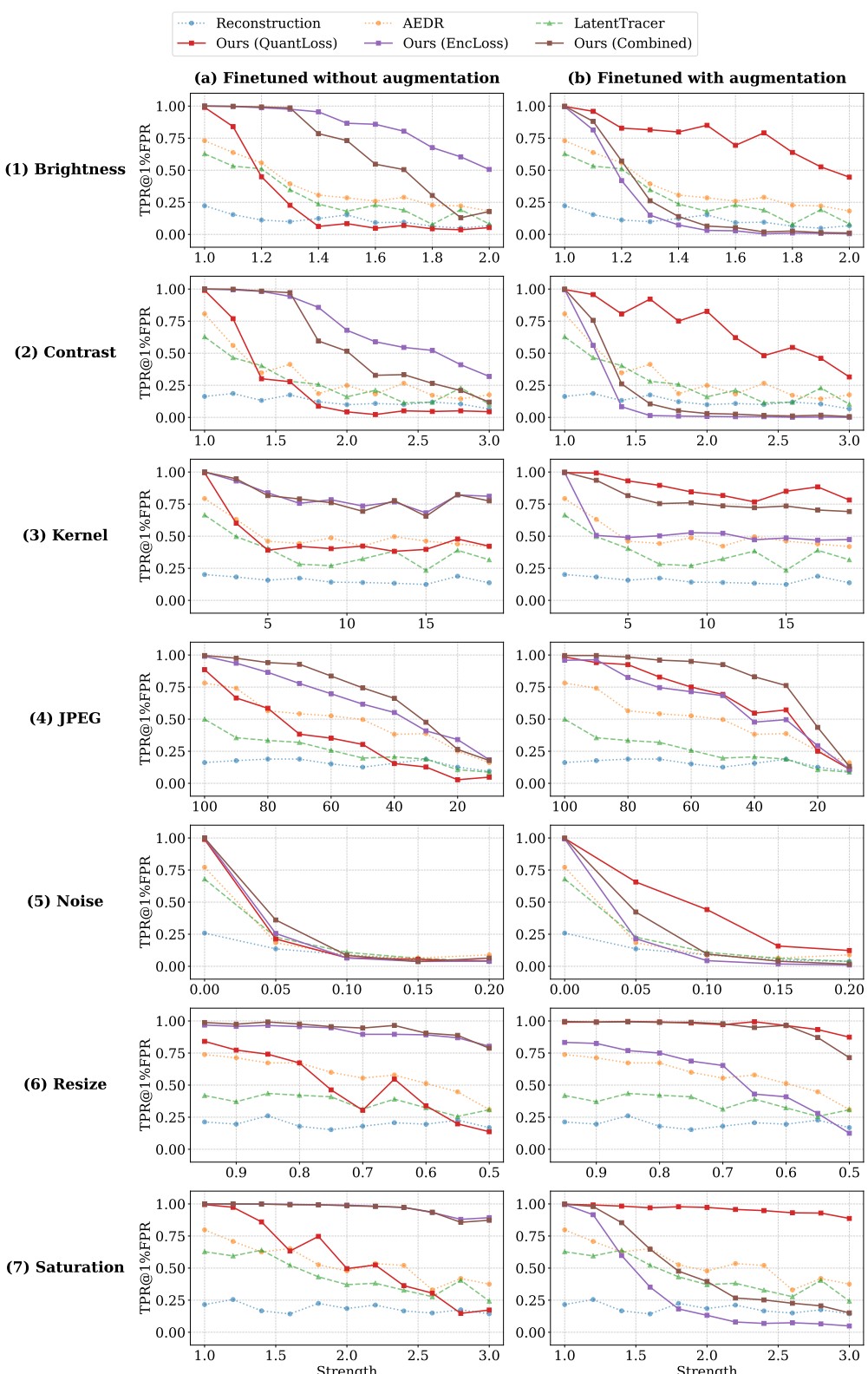

Figure A3: **Robustness Test for Taming on 7 common image post-processing techniques.**

achieves a high TPR@1%FPR for most attacks, outperforming the three baseline methods by a large margin, especially after finetuning with augmentations. Meanwhile, we also observe an interesting fact that our attribution by EncLoss performs worse after the augmentation. Here, we provide an intuition on why finetuning with augmentations works better for QuantLoss but worse for EncLoss.

The improved performance of QuantLoss after finetuning with augmentation can be attributed to the loss $\mathcal{L}_{\text{inv}}$ in Equation (6), where we optimize $D^{-1}$ to reconstruct the original feature map. On the finetuning setting **without** augmentations, the loss can be rewritten as:

$$\mathcal{L}_{\text{inv}} = \|f_Z - D^{-1}(img)\|_2, \tag{12}$$

where $img$ is the initial image reconstruction $D(f_Z)$. When training **with** augmentations, the augmentations are applied to $img$, which leads to an augmented version of our loss function:

$$\mathcal{L}_{\text{inv}} = \|f_Z - D^{-1}(Aug(img))\|_2. \tag{13}$$

Here, we want to invert an augmented, generated image to the feature map $f_Z$. Therefore, the tokens can still be well reconstructed for belonging images even after augmentation, which leads to the better performance of our QuantLoss.

However, the target feature map $f_Z$ is the original **un-augmented** feature map. When we optimize $D^{-1}$ to reconstruct $f_Z$, the $D(D^{-1}(Aug(img))$ becomes closer to the **un-augmented** image $img$. As a result, we actually train $D^{-1}$ to "remove" the augmentation and increase the loss for the EncLoss, as the loss for belonging images is now the loss of the augmentation:

$$\mathcal{L}_{\text{Enc}} = \|Aug(img) - img\|_2.$$

The EncLoss distributions of belonging and non-belonging images are now more overlapping, leading to lower TPR@1%FPR. Due to our construction of $\mathcal{L}_{\text{Comb}}$, the overlapping distributions of the EncLoss have a negative impact on the combined provenance signal. Therefore in settings, where robustness is critical, the QuantLoss provides a reliable provenance signal.

## I  AE ATTRIBUTION OR AR ATTRIBUTION

In this work, we choose to attribute images to the autoencoder (AE) instead of the autoregressive (AR) model. We think AE attribution is more important than AR attribution for IAR data provenance for the following reason: if different AR models are based on the same AE model and training data, they are essentially trained on the same token sequence. Those AE models are trained to fit the same token distribution, so they have similar probabilities of a generated image. Therefore, we find it more significant to detect that an image is from the autoencoder of a given IAR.

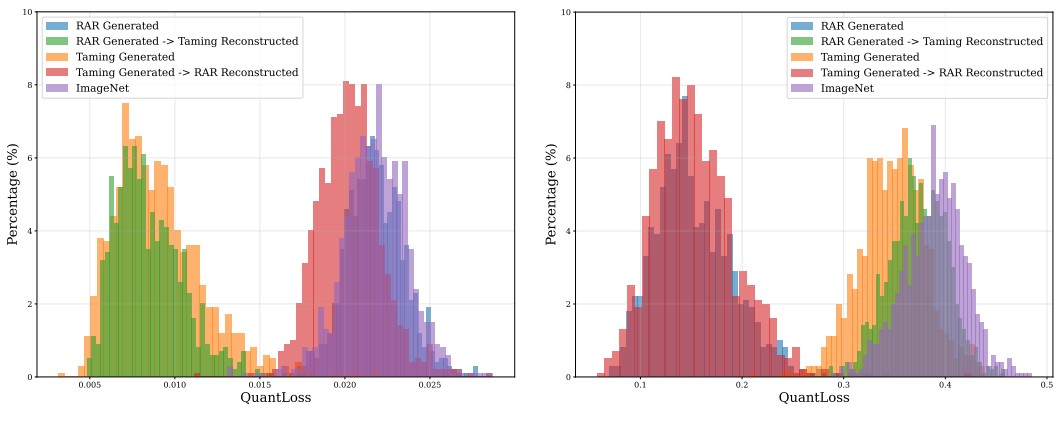

(a) Attributing images with the Taming AE.          (b) Attributing images with the RAR AE.

Figure A4: **Only the final AE decoding is significant for AE attribution.** We analyze the setting of 1. Taming Generated → 2. RAR Encoded + Generated and vice versa.

We analyze the AE attribution in Figure A4 and observe that only the final AE generation is significant for attribution. Specifically, we generate 1,000 token maps with the underlying AR model (*e.g.,*

RAR). These are decoded with the RAR decoder yielding the blue distribution in Figure A4b with a low QuantLoss. However, we can observe that the setting of Generated by RAR → then Encoded + Generated by the Taming AE, there is a clear distribution shift (green) such that the images are no longer attributed as belonging to RAR but to Taming.

This occurs due to the different codebook $Z$ and latent space of different AEs, where the original signal by the first AE (*e.g.,* RAR) is overwritten by the signal of the second AE (*e.g.,* Taming). The image originally constructed of the first codebook is now reconstructed by the second codebook removing the traces of the first.

## J EXTENDED RESULTS WITH MORE CONFIGURATIONS

We show an extended version of our main results in Table A5 for all single-scale models and in Table A4 for all multi-scale models. We use different colors for the baselines; Reconstruction, AEDR and LatentTracer , our EncLoss and QuantLoss and our Combined Loss .

Table A4: **TPR@1%FPR (%) for multi-scale IARs under different settings**. Here, the belonging images are generated by the model specified in the first column, and the non-belonging images are from 3 natural image datasets or generated by the other IARs. "Double Ratio" denotes the ratio between the losses of the first and second reconstruction. "FT" denotes using the finetuned decoder inversion.

| Model | Method | FT | Double Ratio | Natural | | | Generated | | | | | |
|---|---|---|---|---|---|---|---|---|---|---|---|---|
| | | | | ImageNet | LAION | MS-COCO | LlamaGen | RAR | Taming | VAR | Infinity | VQDiff |
| VAR | LatentTracer | - | - | 3.9 | 1.3 | 12.0 | 0.2 | 5.6 | 0.1 | - | 15.4 | 15.3 |
| | Reconstruction Loss | ✗ | ✗ | 1.4 | 1.4 | 3.6 | 0.1 | 1.6 | 0.0 | - | 5.9 | 5.9 |
| | | | ✓ | 29.1 | 15.7 | 50.6 | 14.7 | 28.3 | 14.0 | - | 37.5 | 50.8 |
| | | ✓ | ✗ | 2.3 | 2.9 | 6.6 | 0.3 | 2.7 | 0.0 | - | 10.5 | 10.4 |
| | | | ✓ | 32.7 | 12.2 | 47.5 | 3.8 | 15.0 | 3.8 | - | 36.5 | 33.5 |
| | EncLoss | ✗ | ✗ | 2.5 | 0.8 | 6.0 | 0.2 | 2.1 | 0.0 | - | 6.9 | 6.5 |
| | | | ✓ | 1.5 | 2.2 | 5.0 | 10.1 | 2.3 | 8.3 | - | 5.9 | 5.0 |
| | | ✓ | ✗ | 17.0 | 15.8 | 31.8 | 6.1 | 21.7 | 1.4 | - | 41.4 | 41.5 |
| | | | ✓ | 100.0 | 96.8 | 100.0 | 98.1 | 100.0 | 99.7 | - | 100.0 | 100.0 |
| | QuantLoss | ✗ | ✗ | 0.4 | 0.0 | 2.0 | 0.0 | 0.5 | 0.0 | - | 4.5 | 2.1 |
| | | | ✓ | 14.1 | 37.7 | 37.0 | 0.0 | 20.0 | 0.0 | - | 97.4 | 96.0 |
| | | ✓ | ✗ | 0.4 | 0.0 | 10.0 | 0.0 | 4.5 | 0.0 | - | 13.4 | 1.6 |
| | | | ✓ | 0.3 | 0.0 | 5.6 | 0.0 | 7.1 | 0.0 | - | 5.5 | 4.0 |
| | QuantLoss +Opt | ✗ | ✗ | 3.5 | 3.1 | 6.8 | 1.0 | 4.9 | 0.6 | - | 9.5 | 6.8 |
| | | | ✓ | 3.2 | 5.4 | 2.3 | 2.1 | 6.6 | 2.3 | - | 5.2 | 6.1 |
| | | ✓ | ✗ | 95.0 | 92.9 | 94.4 | 89.8 | 94.5 | 88.4 | - | 95.7 | 95.2 |
| | | | ✓ | 10.3 | 2.0 | 14.1 | 1.0 | 15.0 | 3.3 | - | 8.6 | 6.3 |
| | Combined Loss | ✗ | - | 2.7 | 3.5 | 5.4 | 8.4 | 3.2 | 6.3 | - | 8.4 | 7.8 |
| | | ✓ | - | 100.0 | 99.2 | 100.0 | 99.2 | 100.0 | 100.0 | - | 100.0 | 100.0 |
| Infinity | LatentTracer | - | - | 0.0 | 0.0 | 10.9 | 31.7 | 0.2 | 0.0 | 5.8 | - | 5.3 |
| | Reconstruction Loss | ✗ | ✗ | 0.0 | 0.0 | 16.6 | 0.0 | 1.3 | 0.0 | 2.8 | - | 0.0 |
| | | | ✓ | 0.4 | 0.1 | 0.1 | 0.1 | 51.6 | 0.4 | 17.5 | - | 4.5 |
| | | ✓ | ✗ | 0.2 | 0.3 | 1.6 | 0.0 | 0.1 | 0.0 | 0.2 | - | 0.5 |
| | | | ✓ | 0.3 | 97.2 | 99.2 | 2.2 | 1.9 | 1.1 | 42.5 | - | 27.0 |
| | EncLoss | ✗ | ✗ | 0.0 | 0.2 | 0.8 | 0.0 | 0.0 | 0.0 | 0.2 | - | 0.3 |
| | | | ✓ | 0.9 | 11.3 | 12.3 | 0.1 | 1.0 | 0.1 | 2.1 | - | 4.2 |
| | | ✓ | ✗ | 0.3 | 2.5 | 4.5 | 0.1 | 0.2 | 0.0 | 0.8 | - | 1.3 |
| | | | ✓ | 0.0 | 94.9 | 98.9 | 1.4 | 0.6 | 0.4 | 11.8 | - | 35.1 |
| | QuantLoss | ✗ | ✗ | 0.0 | 0.0 | 16.6 | 0.0 | 1.3 | 0.0 | 2.8 | - | 0.0 |
| | | | ✓ | 0.4 | 0.1 | 0.1 | 0.1 | 51.6 | 0.4 | 17.5 | - | 4.5 |
| | | ✓ | ✗ | 99.4 | 85.6 | 99.4 | 99.2 | 99.5 | 99.1 | 99.4 | - | 99.4 |
| | | | ✓ | 0.1 | 0.0 | 0.1 | 0.1 | 0.7 | 0.1 | 0.1 | - | 0.1 |
| | Combined Loss | ✗ | - | 0.1 | 0.0 | 29.6 | 0.0 | 2.5 | 0.0 | 5.4 | - | 0.9 |
| | | ✓ | - | 0.0 | 98.2 | 100.0 | 9.1 | 3.4 | 1.1 | 57.3 | - | 76.6 |

Table A5: **TPR@1%FPR (%) for single-scale models under different settings**. Here, the belonging images are generated by the model specified in the first column, and the non-belonging images are from 3 natural image datasets or generated by the other IARs. "Double Ratio" denotes the ratio between the losses of the first and second reconstruction. "FT" denotes using the finetuned encoder.

| Model | Method | FT | Double Ratio | Natural | | | Generated | | | | | |
|---|---|---|---|---|---|---|---|---|---|---|---|---|
| | | | | ImageNet | LAION | MS-COCO | LlamaGen | RAR | Taming | VAR | Infinity | VQDiff |
| LlamaGen | LatentTracer | - | - | 93.5 | 89.2 | 97.9 | - | 96.3 | 80.7 | 96.9 | 99.0 | 98.7 |
| | Reconstruction Loss | ✗ | ✗ | 33.6 | 34.0 | 44.3 | - | 39.7 | 4.3 | 45.7 | 70.0 | 63.0 |
| | | ✗ | ✓ | 50.9 | 55.3 | 50.5 | - | 59.5 | 57.7 | 67.0 | 70.7 | 68.1 |
| | | ✓ | ✗ | 98.0 | 98.3 | 98.3 | - | 98.3 | 90.0 | 98.5 | 99.3 | 99.2 |
| | | ✓ | ✓ | 0.0 | 0.0 | 0.0 | - | 0.0 | 0.0 | 0.0 | 0.0 | 0.0 |
| | EncLoss | ✗ | ✗ | 5.4 | 4.5 | 9.6 | - | 8.1 | 0.6 | 9.9 | 27.9 | 22.5 |
| | | ✗ | ✓ | 99.7 | 83.5 | 99.6 | - | 99.6 | 94.2 | 99.6 | 99.8 | 99.8 |
| | | ✓ | ✗ | 19.0 | 23.8 | 34.3 | - | 26.4 | 2.8 | 32.8 | 63.4 | 54.9 |
| | | ✓ | ✓ | 100.0 | 100.0 | 100.0 | - | 100.0 | 100.0 | 100.0 | 100.0 | 100.0 |
| | QuantLoss | ✗ | ✗ | 98.9 | 78.2 | 99.9 | - | 99.9 | 98.4 | 98.2 | 100.0 | 97.5 |
| | | ✗ | ✓ | 93.5 | 66.3 | 99.2 | - | 98.1 | 97.3 | 99.3 | 99.4 | 97.5 |
| | | ✓ | ✗ | 100.0 | 99.8 | 100.0 | - | 100.0 | 100.0 | 100.0 | 100.0 | 100.0 |
| | | ✓ | ✓ | 100.0 | 99.4 | 100.0 | - | 100.0 | 100.0 | 100.0 | 100.0 | 100.0 |
| | Combined Loss | ✗ | - | 99.9 | 99.6 | 99.9 | - | 99.9 | 99.6 | 99.9 | 99.9 | 100.0 |
| | | ✓ | - | 100.0 | 100.0 | 100.0 | - | 100.0 | 100.0 | 100.0 | 100.0 | 100.0 |
| RAR | LatentTracer | - | - | 6.0 | 6.1 | 15.2 | 0.4 | - | 0.0 | 9.3 | 24.6 | 26.9 |
| | Reconstruction Loss | ✗ | ✗ | 3.8 | 4.1 | 7.4 | 0.8 | - | 0.1 | 5.7 | 18.1 | 18.8 |
| | | ✗ | ✓ | 29.5 | 16.6 | 36.6 | 10.6 | - | 2.3 | 35.9 | 49.9 | 27.6 |
| | | ✓ | ✗ | 47.9 | 44.2 | 60.1 | 26.7 | - | 12.2 | 48.0 | 77.2 | 70.4 |
| | | ✓ | ✓ | 63.7 | 36.5 | 63.5 | 39.8 | - | 33.4 | 63.6 | 70.2 | 68.1 |
| | EncLoss | ✗ | ✗ | 2.0 | 3.5 | 4.4 | 0.4 | - | 0.2 | 2.8 | 11.1 | 19.6 |
| | | ✗ | ✓ | 1.9 | 6.4 | 6.4 | 1.0 | - | 0.7 | 1.1 | 7.4 | 12.2 |
| | | ✓ | ✗ | 22.6 | 21.2 | 27.3 | 5.1 | - | 2.5 | 26.0 | 47.9 | 44.0 |
| | | ✓ | ✓ | 98.2 | 98.0 | 98.9 | 93.5 | - | 91.9 | 96.6 | 99.5 | 99.7 |
| | QuantLoss | ✗ | ✗ | 12.8 | 13.0 | 14.4 | 2.7 | - | 1.7 | 10.2 | 14.1 | 22.1 |
| | | ✗ | ✓ | 30.4 | 26.4 | 67.1 | 30.4 | - | 12.1 | 52.7 | 72.3 | 76.1 |
| | | ✓ | ✗ | 99.9 | 99.8 | 99.9 | 99.8 | - | 99.2 | 100.0 | 100.0 | 99.8 |
| | | ✓ | ✓ | 99.7 | 85.9 | 100.0 | 95.6 | - | 96.2 | 100.0 | 100.0 | 100.0 |
| | Combined Loss | ✗ | - | 6.2 | 9.1 | 10.0 | 1.7 | - | 0.5 | 3.1 | 13.4 | 21.8 |
| | | ✓ | - | 100.0 | 100.0 | 100.0 | 99.9 | - | 99.9 | 100.0 | 100.0 | 100.0 |
| Taming | LatentTracer | - | - | 73.0 | 61.0 | 75.9 | 36.4 | 66.8 | - | 76.0 | 85.4 | 87.4 |
| | Reconstruction Loss | ✗ | ✗ | 27.5 | 21.5 | 27.6 | 10.1 | 18.9 | - | 27.7 | 39.0 | 46.1 |
| | | ✗ | ✓ | 80.4 | 82.5 | 81.9 | 70.7 | 80.7 | - | 78.1 | 91.9 | 87.5 |
| | | ✓ | ✗ | 77.3 | 70.7 | 74.9 | 63.9 | 76.0 | - | 77.3 | 86.0 | 88.0 |
| | | ✓ | ✓ | 87.6 | 84.1 | 89.9 | 81.9 | 90.7 | - | 89.9 | 90.4 | 89.3 |
| | EncLoss | ✗ | ✗ | 20.5 | 14.3 | 22.1 | 7.2 | 14.5 | - | 21.5 | 34.6 | 38.8 |
| | | ✗ | ✓ | 4.0 | 2.8 | 2.2 | 1.7 | 1.7 | - | 1.8 | 1.5 | 2.6 |
| | | ✓ | ✗ | 53.7 | 39.1 | 49.8 | 29.5 | 43.9 | - | 52.2 | 65.2 | 70.9 |
| | | ✓ | ✓ | 100.0 | 100.0 | 100.0 | 100.0 | 100.0 | - | 100.0 | 100.0 | 100.0 |
| | QuantLoss | ✗ | ✗ | 38.0 | 4.2 | 52.1 | 2.1 | 36.5 | - | 46.4 | 81.2 | 71.5 |
| | | ✗ | ✓ | 39.3 | 39.4 | 65.2 | 34.3 | 48.9 | - | 45.8 | 84.6 | 75.7 |
| | | ✓ | ✗ | 99.6 | 88.8 | 99.6 | 96.2 | 99.6 | - | 99.5 | 99.8 | 99.5 |
| | | ✓ | ✓ | 99.9 | 98.3 | 100.0 | 99.8 | 100.0 | - | 100.0 | 100.0 | 99.9 |
| | Combined Loss | ✗ | - | 15.3 | 15.7 | 15.3 | 7.8 | 8.8 | - | 12.5 | 13.7 | 19.2 |
| | | ✓ | - | 100.0 | 100.0 | 100.0 | 100.0 | 100.0 | - | 100.0 | 100.0 | 100.0 |
| VQ-Diffusion | LatentTracer | - | - | 97.7 | 93.8 | 98.4 | 97.3 | 97.9 | 93.6 | 98.5 | 98.6 | - |
| | Reconstruction Loss | ✗ | ✗ | 17.2 | 8.8 | 24.3 | 6.3 | 21.8 | 1.6 | 21.2 | 43.0 | - |
| | | ✗ | ✓ | 89.7 | 51.4 | 90.0 | 79.8 | 93.6 | 77.2 | 87.5 | 83.6 | - |
| | | ✓ | ✗ | 67.6 | 62.2 | 71.3 | 71.5 | 71.0 | 55.2 | 78.3 | 82.0 | - |
| | | ✓ | ✓ | 72.4 | 51.2 | 89.7 | 68.4 | 87.7 | 61.9 | 72.4 | 92.2 | - |
| | EncLoss | ✗ | ✗ | 0.6 | 0.3 | 5.3 | 0.6 | 1.0 | 0.1 | 2.6 | 13.0 | - |
| | | ✗ | ✓ | 87.8 | 36.6 | 50.9 | 90.8 | 95.6 | 93.7 | 83.6 | 63.6 | - |
| | | ✓ | ✗ | 15.7 | 5.4 | 24.6 | 15.5 | 14.8 | 3.5 | 21.9 | 34.3 | - |
| | | ✓ | ✓ | 100.0 | 100.0 | 100.0 | 99.7 | 100.0 | 100.0 | 100.0 | 100.0 | - |
| | QuantLoss | ✗ | ✗ | 17.4 | 1.6 | 40.7 | 17.0 | 30.5 | 7.1 | 19.9 | 63.1 | - |
| | | ✗ | ✓ | 99.9 | 100.0 | 100.0 | 99.6 | 100.0 | 93.3 | 100.0 | 100.0 | - |
| | | ✓ | ✗ | 92.1 | 43.3 | 99.1 | 96.8 | 97.6 | 85.8 | 95.7 | 99.1 | - |
| | | ✓ | ✓ | 100.0 | 99.8 | 100.0 | 100.0 | 100.0 | 100.0 | 100.0 | 100.0 | - |
| | Combined Loss | ✗ | - | 86.1 | 33.2 | 82.9 | 78.8 | 95.7 | 65.8 | 83.3 | 86.1 | - |
| | | ✓ | - | 100.0 | 99.4 | 100.0 | 99.9 | 100.0 | 99.9 | 100.0 | 100.0 | - |

## K  MAIN OBSERVATION

Our initial observation was that the token representations differ significantly between natural and IAR-generated images. Intuitively, the token representations of generated images are consistently closer to the codebook entries than those of natural images (shown in Figure 1). We compute the token representations for the natural and generated images and compare their distances to the closest token representations in the codebook. We present the results in the table below.

Table A6: **Distances between token representations and codebook entries for generated vs natural images.** We use the MS-COCO dataset as natural images (denoted *Natural*) and the images generated by a given model (represented as *Generated*). We compute the distances in the $\ell_2$ norm.

| Model | Natural | Generated |
|---|---|---|
| LlamaGen | 0.0108 ($\pm$0.000) | 0.0033 ($\pm$0.001) |
| RAR | 0.3942 ($\pm$0.030) | 0.1538 ($\pm$0.037) |
| Taming | 0.0225 ($\pm$0.002) | 0.0094 ($\pm$0.003) |
| VQ-Diffusion | 0.0216 ($\pm$0.003) | 0.0086 ($\pm$0.002) |
| Infinity | 0.0116 ($\pm$0.000) | 0.0109 ($\pm$0.000) |
| VAR | 0.1381 ($\pm$0.006) | 0.1075 ($\pm$0.011) |

## L  ROBUSTNESS ON MORE DATASETS

In addition to the robustness evaluation in Table 2, we show an extended version of robustness evaluation across more datasets in Table A7. We show that our method outperforms the baselines by a very large margin after image post-processing, validating the universal robustness of our approach.

## M  COMPREHENSIVE ANALYSIS ON MORE METRICS

Additionally to our TPR@1%FPR, we report the TPR for the baseline methods and our methods at stricter FPR values (0.5%FPR in Table A8 and 0.1%FPR in Table A9) as well as the AUC in Table A10. ROC plots for RAR compared to the baselines are illustrated in Figure A5. When evaluated under more strict settings in Table A8 and Table A9, baseline methods have a very limited performance in most cases, while our methods perform consistently well. The AUC and ROC results in Table A9 and Figure A5 show that our method strictly outperform all of the baselines for all models and non-belonging datasets.

## N  GENERALIZATION ACROSS HYPERPARAMETERS AND DATA SPLITS

To demonstrate the generalization of our method, we provided further experiments where the conditional guidance scales and sampling temperatures are different during generating fine-tuning and evaluation sets. We use CFG=4 and temperature=1.0 for generating the fine-tuning set. The results in Table A11 show that our method achieves high performance across different CFG (3,4,5) and temperatures (0.8, 1.0, 1.2). In addition, we performed an experiment for class split, where we separated the data used to fine-tune the inverse decoder according to the classes. Specifically, we use the first 500 classes for the model to generate the fine-tuning set and use the remaining 500 classes for evaluation, ensuring that the model can not overfit on the distribution. In Table A12, we denote this as Ours (class split) and our standard fine-tuning as Ours (random split). We report the TPR@1%FPR of belonging vs non-belonging data. The results show that our method performs consistently well across the two settings and outperforms the baseline methods significantly.

## O  EVALUATION OF DIFFERENT CODEBOOK DISTANCE METRICS

For QuantLoss, we use the L2 norm following the original quantization algorithms in IARs. We show in Table A13 that using cosine similarity yields similar results as the L2 norm. Our key

Table A7: **TPR@1%FPR (%) under different post-processing image transforms and on different datasets.** The first column indicates the evaluated transform and the strength of the transform. The second column indicates the evaluated method. The model is RAR, the belonging data is generated by RAR, and the non-belonging data is denoted in the table heading.

| Transform | Method | Natural | | | Generated | | | | |
|---|---|---|---|---|---|---|---|---|---|
| | | ImageNet | LAION | MS-COCO | LlamaGen | Taming | VAR | Infinity | VQDiff |
| Noise=0.05 | Reconstruction | 2.3 | 0.8 | 2.1 | 0.0 | 0.0 | 1.3 | 13.3 | 6.6 |
| | LatentTracer | 3.4 | 0.7 | 3.8 | 0.0 | 0.1 | 1.4 | 10.7 | 7.2 |
| | AEDR | 7.3 | 4.7 | 6.2 | 4.4 | 0.5 | 4.6 | 13.7 | 5.5 |
| | Ours | **87.8** | **82.3** | **94.6** | **75.2** | **65.4** | **90.3** | **95.9** | **93.1** |
| Kernel=9 | Reconstruction | 3.0 | 2.1 | 3.3 | 0.4 | 0.1 | 2.8 | 9.5 | 5.9 |
| | LatentTracer | 4.7 | 2.1 | 3.4 | 0.4 | 0.1 | 3.1 | 12.4 | 7.5 |
| | AEDR | 11.4 | 5.0 | 13.8 | 2.5 | 0.5 | 9.9 | 18.9 | 12.0 |
| | Ours | **80.5** | **74.1** | **82.3** | **69.7** | **63.9** | **78.3** | **83.4** | **82.6** |
| JPEG=60 | Reconstruction | 3.6 | 2.3 | 4.8 | 0.5 | 0.0 | 2.1 | 11.4 | 12.1 |
| | LatentTracer | 4.8 | 3.5 | 6.9 | 0.1 | 0.0 | 2.8 | 15.9 | 15.4 |
| | AEDR | 8.9 | 5.4 | 18.5 | 2.4 | 0.3 | 6.8 | 29.1 | 11.9 |
| | Ours | **96.1** | **94.1** | **98.8** | **90.3** | **83.3** | **98.3** | **98.9** | **98.5** |
| Brightness=1.6 | Reconstruction | 1.4 | 0.5 | 2.3 | 0.1 | 0.0 | 1.2 | 4.6 | 3.2 |
| | LatentTracer | 2.3 | 1.0 | 2.7 | 0.0 | 0.0 | 1.7 | 5.8 | 3.7 |
| | AEDR | 1.9 | 0.5 | 2.0 | 0.5 | 0.4 | 1.1 | 3.1 | 2.0 |
| | Ours | **92.3** | **75.6** | **95.1** | **78.6** | **60.4** | **94.0** | **97.3** | **96.1** |
| Contrast=2.0 | Reconstruction | 1.6 | 2.2 | 2.8 | 0.0 | 0.0 | 1.6 | 5.5 | 7.7 |
| | LatentTracer | 3.0 | 1.8 | 6.3 | 0.1 | 0.0 | 2.2 | 7.7 | 9.4 |
| | AEDR | 1.4 | 0.8 | 2.4 | 0.9 | 0.3 | 2.4 | 3.9 | 3.2 |
| | Ours | **91.1** | **83.7** | **95.1** | **74.3** | **65.7** | **92.3** | **95.1** | **94.6** |
| Saturation=2.0 | Reconstruction | 3.1 | 2.2 | 3.8 | 0.4 | 0.1 | 1.3 | 9.8 | 10.2 |
| | LatentTracer | 3.6 | 3.7 | 8.2 | 0.2 | 0.0 | 3.4 | 14.5 | 14.5 |
| | AEDR | 9.5 | 4.2 | 8.7 | 1.7 | 0.4 | 5.5 | 18.4 | 10.1 |
| | Ours | **99.2** | **99.7** | **99.8** | **99.5** | **98.8** | **99.8** | **99.9** | **99.8** |
| Resize=0.5 | Reconstruction | 1.0 | 1.9 | 4.5 | 0.9 | 0.0 | 2.4 | 9.9 | 10.0 |
| | LatentTracer | 2.2 | 2.2 | 4.8 | 0.5 | 0.0 | 2.6 | 12.8 | 8.6 |
| | AEDR | 0.2 | 1.5 | 9.7 | 0.8 | 0.3 | 9.1 | 10.8 | 11.6 |
| | Ours | **98.4** | **98.6** | **99.5** | **96.9** | **93.3** | **99.3** | **99.7** | **99.4** |

finding is that belonging images are closer to the codebook entries compared to non-belonging images, where two distance metrics can both capture the distance difference.

## P    COMBINING STRATEGIES FOR QUANTLOSS AND ENCLOSS

Since our EncLoss $L_{Enc}^{Cal}$ is a ratio, a multiplicative combination treats it as a scaling factor. We provide an ablation study comparing additive versus multiplicative combinations, as well as the use of learned weights in Table A14. For both addition and multiplication, we combine the two losses with the respective arithmetic operation. In the weighted scenarios, we determine optimal weights for EncLoss by keeping the weight for the QuantLoss fixed. For *Addition Weighted* we determine the optimal weight $w_{Enc}$ for EncLoss via grid search by leveraging ImageNet as a calibration set: we search 1,000 evenly spaced values between 0.001 and 1, and another 1,000 values between 1 and 1,000. For *Multiplication Power*, the weight is used as an exponent, and we apply a grid search over 1,000 values between 0.01 and 10.

## Q    ADDITIONAL BASELINE OF GENERAL AI DETECTION

To provide additional baseline methods, we evaluate a state-of-the-art AI-generated image detection methods, specifically AIDE (Yan et al., 2025) and a detection method carefully crafted for IARs called $D^3QE$ (Zhang et al., 2025). We leverage the provided pre-trained weights for each method and report the results of AIDE in Table A15 and $D^3QE$ in Table A16. We use 1,000 images as belonging and 1,000 images as non-belonging datasets. We note that AIDE has a very limited performance for detecting IAR-generated images, and both approaches have an even worse performance to distinguish data generated by different models.

Table A8: **TPR@0.5%FPR our method and the baselines**. The first column indicates the original model that has generated the belonging images, the heading of the other columns specifies the natural datasets or generators from which the non-belonging images are obtained. Our method is instantiated with the best-performing set of signals from Section 3.3 for each original model.

| Model | Method | Natural | | | Generated | | | | | |
|---|---|---|---|---|---|---|---|---|---|---|
| | | ImageNet | LAION | MS-COCO | LlamaGen | RAR | Taming | VAR | Infinity | VQDiff |
| LlamaGen | Reconstruction | 23.4 | 25.0 | 30.6 | - | 30.4 | 2.1 | 31.4 | 61.8 | 60.4 |
| | LatentTracer | 89.7 | 82.7 | 93.6 | - | 94.6 | 72.5 | 95.1 | 98.8 | 98.0 |
| | AEDR | 41.1 | 49.9 | 38.1 | - | 55.2 | 50.0 | 57.4 | 66.4 | 59.8 |
| | Ours | **100.0** | **100.0** | **100.0** | - | **100.0** | **100.0** | **100.0** | **100.0** | **100.0** |
| RAR | Reconstruction | 2.7 | 3.1 | 5.8 | 0.2 | - | 0.0 | 2.5 | 10.4 | 14.6 |
| | LatentTracer | 2.2 | 1.0 | 2.2 | 0.2 | - | 0.0 | 2.1 | 7.7 | 5.3 |
| | AEDR | 13.6 | 15.1 | 30.0 | 4.0 | - | 0.8 | 22.5 | 42.4 | 17.2 |
| | Ours | **99.9** | **100.0** | **100.0** | **99.9** | - | **98.9** | **99.9** | **100.0** | **100.0** |
| Taming | Reconstruction | 17.3 | 18.7 | 22.2 | 6.5 | 18.4 | - | 25.2 | 39.5 | 45.0 |
| | LatentTracer | 64.8 | 52.2 | 70.6 | 32.2 | 69.8 | - | 72.6 | 82.9 | **100.0** |
| | AEDR | 75.8 | 61.7 | 88.7 | 51.8 | 79.8 | - | 73.3 | 88.6 | 80.0 |
| | Ours | **100.0** | **100.0** | **100.0** | **100.0** | **100.0** | - | **100.0** | **100.0** | **100.0** |
| VAR | Reconstruction | 0.5 | 0.5 | 2.1 | 0.1 | 1.4 | 0.0 | - | 5.4 | 4.0 |
| | LatentTracer | 2.8 | 0.1 | 8.4 | 0.1 | 4.2 | 0.0 | - | 12.8 | 10.1 |
| | AEDR | 9.8 | 10.9 | 45.3 | 4.1 | 22.6 | 3.9 | - | 33.1 | 40.4 |
| | Ours | **100.0** | **97.1** | **100.0** | **96.8** | **100.0** | **99.6** | - | **100.0** | **100.0** |
| Infinity | Reconstruction | 0.0 | 0.2 | 0.3 | 0.0 | 0.0 | 0.0 | 0.2 | - | 0.2 |
| | LatentTracer | 0.0 | 6.5 | 25.8 | 0.0 | 0.0 | 0.0 | 1.6 | - | 3.6 |
| | AEDR | 1.1 | 7.1 | 36.3 | 0.6 | 2.7 | 0.3 | 8.2 | - | 6.1 |
| | Ours | **99.2** | **15.5** | **99.4** | **99.1** | **99.5** | **99.1** | **99.4** | - | **99.2** |
| VQ-Diffusion | Reconstruction | 4.5 | 6.0 | 12.9 | 4.5 | 16.6 | 0.6 | 16.7 | 33.6 | - |
| | LatentTracer | 95.0 | 90.9 | 97.4 | **96.1** | 97.7 | 88.9 | 98.2 | 98.4 | - |
| | AEDR | 82.7 | 43.1 | 87.3 | 71.0 | 91.7 | 60.0 | 80.2 | 79.8 | - |
| | Ours | **100.0** | **98.7** | **100.0** | 93.3 | **100.0** | **99.6** | **100.0** | **100.0** | - |

Table A9: **TPR@0.1%FPR our method and the baselines**. The first column indicates the original model that has generated the belonging images, the heading of the other columns specifies the natural datasets or generators from which the non-belonging images are obtained. Our method is instantiated with the best-performing set of signals from Section 3.3 for each original model.

| Model | Method | Natural | | | Generated | | | | | |
|---|---|---|---|---|---|---|---|---|---|---|
| | | ImageNet | LAION | MS-COCO | LlamaGen | RAR | Taming | VAR | Infinity | VQDiff |
| LlamaGen | Reconstruction | 13.2 | 15.4 | 18.5 | - | 17.2 | 0.3 | 21.9 | 47.3 | 17.9 |
| | LatentTracer | 79.7 | 75.1 | 85.9 | - | 90.0 | 63.5 | 89.3 | 95.4 | 91.8 |
| | AEDR | 22.4 | 23.7 | 17.7 | - | 31.1 | 38.0 | 45.4 | 57.7 | 46.3 |
| | Ours | **100.0** | **100.0** | **100.0** | - | **100.0** | **99.9** | **100.0** | **100.0** | **100.0** |
| RAR | Reconstruction | 1.8 | 1.8 | 1.4 | 0.1 | - | 0.0 | 1.6 | 7.4 | 3.9 |
| | LatentTracer | 0.4 | 0.2 | 0.9 | 0.0 | - | 0.0 | 0.7 | 4.1 | 0.4 |
| | AEDR | 1.4 | 1.4 | 9.1 | 0.0 | - | 0.0 | 16.6 | 6.7 | 2.6 |
| | Ours | **99.9** | **99.9** | **100.0** | **96.9** | - | **54.1** | **99.9** | **100.0** | **100.0** |
| Taming | Reconstruction | 11.8 | 11.0 | 12.9 | 3.0 | 12.7 | - | 19.1 | 30.7 | 16.4 |
| | LatentTracer | 36.8 | 33.1 | 54.7 | 24.4 | 51.3 | - | 66.3 | 71.1 | 100.0 |
| | AEDR | 58.3 | 35.6 | 68.1 | 30.9 | 75.5 | - | 55.2 | 77.7 | 76.3 |
| | Ours | **100.0** | **92.6** | **100.0** | **100.0** | **100.0** | - | **100.0** | **100.0** | **100.0** |
| VAR | Reconstruction | 0.3 | 0.2 | 0.4 | 0.0 | 0.5 | 0.0 | - | 1.9 | 0.3 |
| | LatentTracer | 0.2 | 0.0 | 3.4 | 0.0 | 1.8 | 0.0 | - | 3.4 | 2.4 |
| | AEDR | 0.0 | 1.6 | 29.5 | 0.6 | 3.8 | 1.2 | - | 16.8 | 25.4 |
| | Ours | **99.6** | **86.3** | **99.8** | **95.8** | **100.0** | **98.9** | - | **100.0** | **100.0** |
| Infinity | Reconstruction | 0.0 | 0.0 | 0.1 | **0.0** | 0.0 | 0.0 | 0.0 | - | 0.0 |
| | LatentTracer | 0.0 | **0.1** | 7.1 | **0.0** | 0.0 | 0.0 | 0.9 | - | 0.5 |
| | AEDR | 0.0 | 0.0 | 4.7 | **0.0** | 1.9 | 0.0 | 3.5 | - | 3.0 |
| | Ours | **98.3** | 0.0 | **99.4** | 0.0 | **99.4** | **31.2** | **99.1** | - | **99.1** |
| VQ-Diffusion | Reconstruction | 2.0 | 3.3 | 2.7 | 1.3 | 6.2 | 0.1 | 10.8 | 15.1 | - |
| | LatentTracer | 91.8 | **86.3** | 95.0 | **92.5** | 96.0 | 78.4 | 97.2 | 96.9 | - |
| | AEDR | 54.9 | 20.7 | 59.1 | 32.2 | 76.9 | 48.0 | 57.6 | 57.5 | - |
| | Ours | **99.9** | 55.6 | **100.0** | 84.8 | **100.0** | **99.0** | **99.5** | **100.0** | - |

Table A10: **AUC our method and the baselines**. The first column indicates the original model that has generated the belonging images, the heading of the other columns specifies the natural datasets or generators from which the non-belonging images are obtained. Our method is instantiated with the best-performing set of signals from Section 3.3 for each original model.

| Model | Method | Natural | | | Generated | | | | | |
|-------|--------|---------|-------|---------|-----------|-----|--------|-----|---------|--------|
| | | ImageNet | LAION | MS-COCO | LlamaGen | RAR | Taming | VAR | Infinity | VQDiff |
| LlamaGen | Reconstruction | 93.9 | 93.1 | 96.5 | - | 92.6 | 81.1 | 94.6 | 97.9 | 97.1 |
| | LatentTracer | 99.7 | 99.6 | 99.9 | - | 99.8 | 98.8 | 99.8 | 99.9 | 99.9 |
| | AEDR | 94.7 | 94.1 | 94.7 | - | 95.7 | 95.2 | 95.2 | 96.1 | 96.0 |
| | Ours | **100.0** | **100.0** | **100.0** | - | **100.0** | **100.0** | **100.0** | **100.0** | **100.0** |
| RAR | Reconstruction | 76.5 | 74.5 | 82.3 | 66.6 | - | 49.7 | 71.4 | 87.2 | 86.5 |
| | LatentTracer | 73.2 | 70.6 | 78.0 | 57.0 | - | 38.6 | 67.2 | 85.3 | 83.7 |
| | AEDR | 90.2 | 86.2 | 89.3 | 87.6 | - | 82.9 | 89.2 | 93.4 | 92.0 |
| | Ours | **100.0** | **100.0** | **100.0** | **100.0** | - | **99.8** | **100.0** | **100.0** | **100.0** |
| Taming | Reconstruction | 86.8 | 82.9 | 88.3 | 80.5 | 84.6 | - | 87.1 | 89.9 | 92.3 |
| | LatentTracer | 98.2 | 97.0 | 98.8 | 95.6 | 98.1 | - | 98.6 | 99.1 | **100.0** |
| | AEDR | 98.7 | 98.5 | 99.1 | 97.2 | 99.0 | - | 98.8 | 99.5 | 99.1 |
| | Ours | **100.0** | **100.0** | **100.0** | **100.0** | **100.0** | - | **100.0** | **100.0** | **100.0** |
| VAR | Reconstruction | 64.1 | 58.1 | 69.0 | 50.3 | 58.9 | 40.4 | - | 69.5 | 70.4 |
| | LatentTracer | 80.6 | 72.6 | 84.9 | 68.3 | 77.5 | 61.5 | - | 82.2 | 82.9 |
| | AEDR | 95.8 | 92.9 | 96.8 | 92.3 | 94.7 | 92.8 | - | 96.5 | 96.7 |
| | Ours | **100.0** | **99.9** | **100.0** | **100.0** | **100.0** | **100.0** | - | **100.0** | **100.0** |
| Infinity | Reconstruction | 30.4 | 61.4 | 64.9 | 20.2 | 19.1 | 23.0 | 24.8 | - | 27.1 |
| | LatentTracer | 62.7 | 91.6 | 94.6 | 53.0 | 50.6 | 54.2 | 62.5 | - | 61.6 |
| | AEDR | 86.2 | 97.2 | 98.9 | 81.5 | 82.7 | 85.5 | 91.4 | - | 86.7 |
| | Ours | **99.8** | **98.8** | **99.7** | **99.6** | **100.0** | **99.7** | **99.9** | - | **99.7** |
| VQ-Diffusion | Reconstruction | 90.7 | 84.3 | 91.9 | 88.9 | 88.5 | 79.2 | 91.4 | 92.4 | - |
| | LatentTracer | 99.9 | 99.8 | 99.9 | 99.9 | 99.9 | 99.6 | **100.0** | **100.0** | - |
| | AEDR | 99.5 | 98.5 | 99.6 | 99.2 | 99.8 | 99.2 | 99.4 | 99.5 | - |
| | Ours | **100.0** | **99.8** | **100.0** | **99.9** | **100.0** | **100.0** | **100.0** | **100.0** | - |

Table A11: **TPR@1%FPR (%) with different conditional guidance scales and sampling temperatures.** The evaluated model is RAR (Combined), and the inverse decoder finetuning data is generated with CFG=4 and temperature=1.0.

| CFG | Temperature | Natural | | | Generated | | | | |
|-----|-------------|---------|-------|---------|-----------|--------|-----|----------|--------|
| | | ImageNet | LAION | MS-COCO | LlamaGen | Taming | VAR | Infinity | VQDiff |
| 3 | 0.8 | 99.7 | 99.8 | 99.7 | 99.6 | 99.4 | 100.0 | 99.8 | 100.0 |
| | 1.0 | 100.0 | 100.0 | 100.0 | 99.5 | 99.4 | 100.0 | 100.0 | 100.0 |
| | 1.2 | 99.9 | 99.9 | 100.0 | 99.8 | 99.8 | 99.9 | 100.0 | 100.0 |
| **4** | 0.8 | 99.5 | 99.8 | 99.8 | 99.3 | 99.4 | 99.5 | 99.8 | 99.9 |
| | **1.0** | 100.0 | 100.0 | 100.0 | 99.9 | 99.9 | 100.0 | 100.0 | 100.0 |
| | 1.2 | 100.0 | 99.6 | 100.0 | 99.5 | 98.8 | 100.0 | 100.0 | 100.0 |
| 5 | 0.8 | 100.0 | 99.6 | 100.0 | 99.5 | 98.8 | 100.0 | 100.0 | 100.0 |
| | 1.0 | 99.9 | 99.0 | 100.0 | 98.4 | 98.9 | 100.0 | 100.0 | 100.0 |
| | 1.2 | 99.7 | 99.4 | 100.0 | 99.4 | 99.3 | 99.5 | 100.0 | 99.8 |

To further evaluate against AIDE, we re-train their model for 5 epochs on 50k images. Importantly, AIDE's training set includes both generated (belonging) and real images, giving it access to additional natural image data that our method does not use. Despite these advantages, the results shown in Table A17 demonstrate that our method still substantially outperforms AIDE. While AIDE achieves relatively strong performance in the natural vs. generated setting, it fails in the more critical setting of attributing a generated image to a specific model. For instance, for RAR, AIDE achieves only 25.9-73.2% TPR@1%FPR in distinguishing images from other IAR models, whereas our method achieves near-perfect 99.9%-100% TPR@1%FPR across all model pairs.

We note that general AI detection methods consider general distinctions between generated and real images, but do not leverage specific artifacts in different IARs and thus fail to attribute an image to a specific model family. However, we utilize the codebook of IARs as the inherent "fingerprint" of the model. Therefore, our method outperforms the general AI detection method significantly.

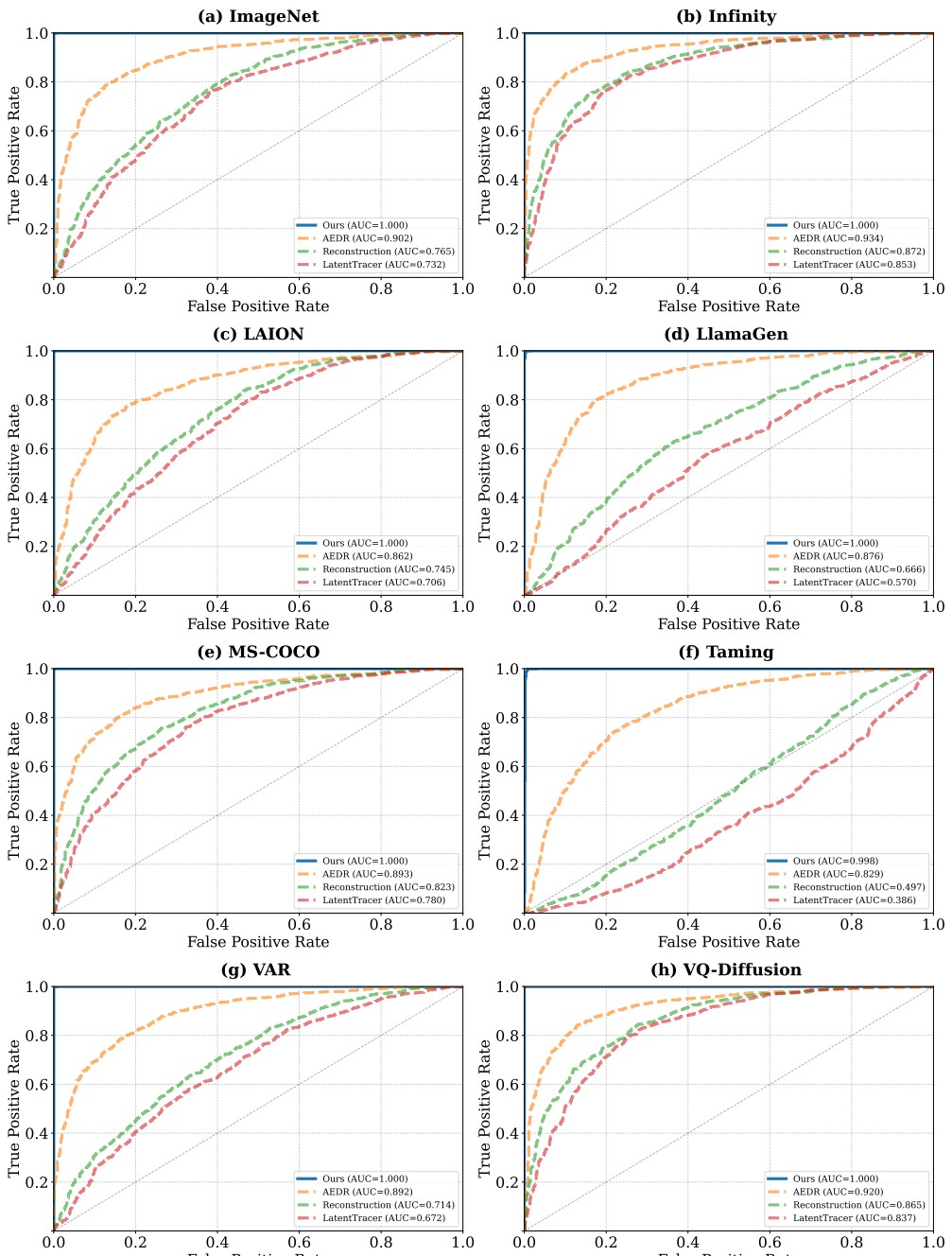

Figure A5: **ROC comparison** for RAR attribution of our method and the baselines.

# R ADAPTIVE ATTACK

Our method is primarily designed for the benign setting, where model owners leverage our framework to prevent model collapse and ensure responsible deployment of their trained models. However, to assess the robustness of our approach, we also consider a more challenging adversarial scenario where a malicious model owner intentionally attempts to evade our detection mechanism.

**Threat Model.** In this adaptive attack scenario, we assume the adversary has knowledge of our methodology. The adversary's goal is to craft adversarial perturbations that increase the distance

Table A12: **TPR@1%FPR (%) of our method evaluated with two types of data split**. The first column indicates the original model that has generated the belonging images, the heading of the other columns specifies the natural datasets or generators from which the non-belonging images are obtained. We denote two splits for Ours, *random split*, where we create training and validation data from the same classes and *class split*, where we create training data using the first 0-499 classes and validation data using the final 500-999 classes.

| Model | Method | Natural | | | Generated | | | | | |
|---|---|---|---|---|---|---|---|---|---|---|
| | | ImageNet | LAION | MS-COCO | LlamaGen | RAR | Taming | VAR | Infinity | VQDiff |
| RAR | Reconstruction | 3.8 | 4.1 | 7.4 | 0.8 | - | 0.1 | 5.7 | 18.1 | 18.8 |
| | LatentTracer | 6.0 | 6.1 | 15.2 | 0.4 | - | 0.0 | 9.3 | 24.6 | 26.9 |
| | AEDR | 29.5 | 16.6 | 36.6 | 10.6 | - | 2.3 | 35.9 | 49.9 | 27.6 |
| | Ours (random split) | 100.0 | 100.0 | 100.0 | 99.9 | - | 99.9 | 100.0 | 100.0 | 100.0 |
| | Ours (class split) | 99.8 | 99.9 | 100.0 | 99.4 | - | 99.7 | 99.7 | 100.0 | 100.0 |
| Taming | Reconstruction | 27.5 | 21.5 | 27.6 | 10.1 | 18.9 | - | 27.7 | 39.0 | 46.1 |
| | LatentTracer | 73.0 | 61.0 | 75.9 | 36.4 | 66.8 | - | 76.0 | 85.4 | 87.4 |
| | AEDR | 80.4 | 82.5 | 81.9 | 70.7 | 80.7 | - | 78.1 | 91.9 | 87.5 |
| | Ours (random split) | 100.0 | 100.0 | 100.0 | 100.0 | 100.0 | - | 100.0 | 100.0 | 100.0 |
| | Ours (class split) | 100.0 | 100.0 | 100.0 | 100.0 | 100.0 | - | 100.0 | 100.0 | 100.0 |
| VAR | Reconstruction | 1.4 | 1.4 | 3.6 | 0.1 | 1.6 | 0.0 | - | 5.9 | 5.9 |
| | LatentTracer | 3.9 | 1.3 | 12.0 | 0.2 | 5.6 | 0.1 | - | 15.4 | 15.3 |
| | AEDR | 29.1 | 15.7 | 50.6 | 14.7 | 28.3 | 14.0 | - | 37.5 | 50.8 |
| | Ours (random split) | 100.0 | 99.2 | 100.0 | 99.2 | 100.0 | 100.0 | - | 100.0 | 100.0 |
| | Ours (class split) | 99.9 | 98.9 | 100.0 | 99.4 | 100.0 | 99.1 | - | 100.0 | 99.9 |

Table A13: **TPR@1%FPR (%) comparison when using different distance metrics.** The evaluated model is RAR.

| Distance Metric | Natural | | | Generated | | | | |
|---|---|---|---|---|---|---|---|---|
| | ImageNet | LAION | MS-COCO | LlamaGen | Taming | VAR | Infinity | VQDiff |
| Cosine Distance | 100.0 | 100.0 | 100.0 | 100.0 | 99.9 | 100.0 | 100.0 | 100.0 |
| L2 Norm | 100.0 | 100.0 | 100.0 | 99.9 | 99.9 | 100.0 | 100.0 | 100.0 |

between the feature map of a generated image and its corresponding codebook entries, thereby causing belonging images to be misclassified as non-belonging images.

**Attack Formulation.** Specifically, the adversarial model owner finetunes an inverse decoder and performs an adversarial attack on a belonging image $x$ by minimizing the following adversarial loss:

$$\mathcal{L}_{\text{adv}}(x, \delta, D^{-1}) = -\|D^{-1}(x) - Q^{-1}(Q(D^{-1}(x)))\|_2 + \lambda\|\delta\|_2, \tag{14}$$

where $\delta$ denotes the adversarial perturbation and $\lambda$ controls the trade-off between attack effectiveness and perturbation magnitude. The adversarial sample is constructed as $x_{\text{adv}} = x + \delta$. This loss function aims to maximize the QuantLoss while constraining the perturbation to remain imperceptible.

**Results and Analysis.** The results are presented in Table A18. We evaluate our method under two attack strengths: $\epsilon = 1/255$ and $\epsilon = 2/255$. Several key observations emerge from these experiments: **First**, finetuning with augmentation significantly improves robustness against adaptive attacks. We attribute this to the fact that augmentation-based training enables the inverse decoder to recover the original tokens robustly even under image degradations, which also generalize to resilience against adversarial perturbations. **Second**, our method demonstrates strong robustness to relatively small adversarial perturbations ($\epsilon = 1/255$), maintaining high TPR@1%FPR across most datasets when finetuned with augmentations (e.g., 97.4% on ImageNet, 96.7% on LAION). **Third**, even under stronger attacks ($\epsilon = 2/255$), our augmentation-based approach retains considerable detection capability (e.g., 49.3% on ImageNet, 51.1% on MS-COCO), substantially outperforming all baseline methods. Notably, the baseline methods show very limited robustness even to weak attacks which are not even tailored to attack them. The TPR@1%FPR for baseline methods drops below 20% in most cases for $\epsilon = 2/255$.

These results demonstrate that while adaptive attacks can degrade detection performance, our framework maintains significantly better robustness compared to existing methods, particularly when

Table A14: **TPR@1%FPR for different combination methods**. The first column indicates the original model that generated the belonging images, the second column shows the combination method used. The heading of the other columns specifies the natural datasets or generators from which the non-belonging images are obtained. The last column shows the optimized weight $w_{Enc}$ for parameterized methods.

| Model | Method | Natural | | | Generated | | | | | | $w_{Enc}$ |
|---|---|---|---|---|---|---|---|---|---|---|---|
| | | ImageNet | LAION | MS-COCO | LlamaGen | RAR | Taming | VAR | Infinity | VQDiff | |
| LlamaGen | Addition | 100.0 | 100.0 | 100.0 | - | 100.0 | 100.0 | 100.0 | 100.0 | 100.0 | - |
| | Addition Weighted | 100.0 | 100.0 | 100.0 | - | 100.0 | 100.0 | 100.0 | 100.0 | 100.0 | 0.00 |
| | Multiplication | 100.0 | 100.0 | 100.0 | - | 100.0 | 100.0 | 100.0 | 100.0 | 100.0 | - |
| | Multiplication Power | 100.0 | 100.0 | 100.0 | - | 100.0 | 100.0 | 100.0 | 100.0 | 100.0 | 0.01 |
| RAR | Addition | 99.3 | 99.2 | 99.5 | 98.5 | - | 97.7 | 98.9 | 99.6 | 99.6 | - |
| | Addition Weighted | 100.0 | 99.8 | 100.0 | 99.8 | - | 99.3 | 99.8 | 100.0 | 100.0 | 0.00 |
| | Multiplication | 100.0 | 100.0 | 100.0 | 99.9 | - | 99.9 | 100.0 | 100.0 | 100.0 | - |
| | Multiplication Power | 100.0 | 99.8 | 100.0 | 99.8 | - | 99.3 | 99.9 | 100.0 | 100.0 | 0.01 |
| Taming | Addition | 100.0 | 100.0 | 100.0 | 100.0 | 100.0 | - | 100.0 | 100.0 | 100.0 | - |
| | Addition Weighted | 100.0 | 99.8 | 100.0 | 99.9 | 100.0 | - | 100.0 | 100.0 | 100.0 | 0.00 |
| | Multiplication | 100.0 | 100.0 | 100.0 | 100.0 | 100.0 | - | 100.0 | 100.0 | 100.0 | - |
| | Multiplication Power | 100.0 | 99.6 | 100.0 | 99.9 | 100.0 | - | 100.0 | 100.0 | 100.0 | 0.27 |
| VAR | Addition | 100.0 | 98.2 | 100.0 | 98.2 | 100.0 | 99.8 | - | 100.0 | 100.0 | - |
| | Addition Weighted | 100.0 | 99.3 | 100.0 | 98.9 | 100.0 | 99.5 | - | 100.0 | 100.0 | 0.02 |
| | Multiplication | 100.0 | 99.5 | 100.0 | 99.2 | 100.0 | 100.0 | - | 100.0 | 100.0 | - |
| | Multiplication Power | 100.0 | 99.3 | 100.0 | 99.2 | 100.0 | 99.5 | - | 100.0 | 100.0 | 0.30 |
| Infinity | Addition | 0.0 | 97.3 | 99.0 | 1.4 | 0.6 | 0.6 | 18.0 | - | 36.1 | - |
| | Addition Weighted | 98.8 | 98.9 | 99.2 | 98.8 | 99.1 | 98.8 | 99.1 | - | 99.1 | 0.00 |
| | Multiplication | 0.0 | 98.3 | 99.2 | 10.1 | 4.1 | 4.2 | 58.8 | - | 77.4 | - |
| | Multiplication Power | 99.3 | 97.8 | 99.4 | 99.1 | 99.4 | 99.1 | 99.3 | - | 99.2 | 0.01 |
| VQ-Diffusion | Addition | 100.0 | 100.0 | 100.0 | 98.0 | 100.0 | 100.0 | 100.0 | 100.0 | - | - |
| | Addition Weighted | 100.0 | 97.4 | 100.0 | 99.8 | 100.0 | 99.4 | 100.0 | 100.0 | - | 0.00 |
| | Multiplication | 100.0 | 99.5 | 100.0 | 99.9 | 100.0 | 99.9 | 100.0 | 100.0 | - | - |
| | Multiplication Power | 100.0 | 95.0 | 100.0 | 99.5 | 100.0 | 99.3 | 100.0 | 100.0 | - | 0.48 |

Table A15: **The performance of AIDE on data provenance**. We present TPR@1%FPR across all test datasets for each model. The first column indicates the original model that has generated the belonging images, the heading of the other columns specifies the natural datasets or generators from which the non-belonging images are obtained.

| Model | Natural | | | Generated | | | | | |
|---|---|---|---|---|---|---|---|---|---|
| | ImageNet | LAION | MS-COCO | LlamaGen | RAR | Taming | VAR | Infinity | VQDiff |
| LlamaGen | 18.8 | 16.8 | 23.2 | - | 0.5 | 0.9 | 3.3 | 6.0 | 6.5 |
| RAR | 27.9 | 26.8 | 30.6 | 5.2 | - | 4.5 | 9.6 | 15.2 | 15.9 |
| Taming | 29.4 | 25.8 | 34.3 | 1.4 | 0.2 | - | 4.3 | 8.8 | 9.5 |
| VAR | 14.6 | 12.5 | 18.4 | 0.2 | 0.1 | 0.2 | - | 3.4 | 3.7 |
| Infinity | 5.6 | 4.6 | 7.4 | 0.0 | 0.0 | 0.0 | 0.5 | - | 1.2 |
| VQ-Diffusion | 10.3 | 9.2 | 13.1 | 0.0 | 0.0 | 0.0 | 0.2 | 0.7 | - |

trained with augmentations. The robustness to adaptive attacks makes our method a practical solution even in the more challenging adversarial scenarios.

## S    EVALUATION ON MULTI-SOURCE DATASET

To simulate a real-world scenario where images come from different sources, we design a multi-source evaluation setting. In this setting, we mix and shuffle all the evaluated images in our experimental setting, including 3 different natural datasets (ImageNet, MS-COCO, LAION) and images generated by 6 different models (LlamaGen, RAR, Taming, VAR, Infinity, VQ-Diffusion). The results in Table A19 show that our method achieves near-perfect TPR@1%FPR on the multi-source dataset across all the evaluated models, demonstrating the applicability of our method.

## T    STATISTICAL TEST OF OUR METHOD

We test if a data point x significantly deviates from a given belonging distribution. Similar to RONAN (Wang et al., 2023) and LatentTracer (Wang et al., 2024) we leverage

Table A16: **TPR@1%FPR (%) of our method and D³QE**. The first column indicates the original model that has generated the belonging images, the heading of the other columns specifies the natural datasets or generators from which the non-belonging images are obtained.

| Model | Method | Natural | | | Generated | | | | | |
|---|---|---|---|---|---|---|---|---|---|---|
| | | ImageNet | LAION | MS-COCO | LlamaGen | RAR | Taming | VAR | Infinity | VQDiff |
| LlamaGen | D³QE | 86.9 | 67.7 | 86.6 | - | 6.8 | 2.0 | 2.0 | 60.1 | 3.7 |
| | Ours | **100.0** | **100.0** | **100.0** | - | **100.0** | **100.0** | **100.0** | **100.0** | **100.0** |
| RAR | D³QE | 78.0 | 49.7 | 77.5 | 0.0 | - | 0.2 | 0.2 | 42.2 | 0.4 |
| | Ours | **100.0** | **100.0** | **100.0** | **99.9** | - | **99.9** | **100.0** | **100.0** | **100.0** |
| Taming | D³QE | 78.0 | 49.7 | 77.5 | 0.0 | 0.2 | - | 0.2 | 42.2 | 0.4 |
| | Ours | **100.0** | **100.0** | **100.0** | **100.0** | **100.0** | - | **100.0** | **100.0** | **100.0** |
| VAR | D³QE | 73.5 | 52.2 | 72.3 | 0.0 | 3.5 | 1.4 | - | 46.7 | 2.3 |
| | Ours | **100.0** | **99.2** | **100.0** | **99.2** | **100.0** | **100.0** | - | **100.0** | **100.0** |
| Infinity | D³QE | 6.3 | 1.5 | 5.9 | 0.0 | 0.1 | 0.0 | 0.0 | - | 0.0 |
| | Ours | **99.4** | **85.6** | **99.4** | **99.2** | **99.5** | **99.1** | **99.4** | - | **99.4** |
| VQDiff | D³QE | 49.9 | 31.6 | 49.2 | 0.0 | 2.1 | 0.5 | 0.5 | 27.8 | - |
| | Ours | **100.0** | **99.4** | **100.0** | **99.9** | **100.0** | **99.9** | **100.0** | **100.0** | - |

Table A17: **TPR@1%FPR for AIDE (Yan et al., 2025) trained on different datasets**. The first column indicates the model, the second column shows the finetuning set used.

| Model | Method | Finetuning Set | Natural | | | Generated | | | | | |
|---|---|---|---|---|---|---|---|---|---|---|---|
| | | | ImageNet | LAION | MS-COCO | LlamaGen | RAR | Taming | VAR | Infinity | VQDiff |
| RAR | AIDE | RAR Generated + ImageNet | 99.7 | 99.4 | **100.0** | 73.2 | - | 53.7 | 48.8 | 99.5 | 54.9 |
| | | RAR Generated + MS-COCO | 97.7 | 98.6 | **100.0** | 41.7 | - | 25.9 | 30.0 | **100.0** | 68.7 |
| | Ours | RAR (Generated) | **100.0** | **100.0** | **100.0** | **99.9** | - | **99.9** | **100.0** | **100.0** | **100.0** |
| Llamagen | AIDE | Llamagen (Generated) + ImageNet | 99.2 | 99.8 | **100.0** | - | 70.4 | 15.5 | 6.2 | 99.8 | 36.1 |
| | | Llamagen Generated + MS-COCO | 86.5 | 97.1 | 99.9 | - | 43.2 | 13.1 | 3.1 | 99.9 | 49.1 |
| | Ours | Llamagen (Generated) | **100.0** | **100.0** | **100.0** | - | **100.0** | **100.0** | **100.0** | **100.0** | **100.0** |

Table A18: **Evaluation under adaptive adversarial attack**. The evaluated model is RAR and $\epsilon$ denotes the strength of the attack.

| $\epsilon$ | Method | Natural | | | Generated | | | | |
|---|---|---|---|---|---|---|---|---|---|
| | | ImageNet | LAION | MS-COCO | LlamaGen | Taming | VAR | Infinity | VQDiff |
| 1/255 | Reconstruction | 2.5 | 3.0 | 6.9 | 0.2 | 0.0 | 2.3 | 16.5 | 16.0 |
| | LatentTracer | 5.6 | 5.9 | 15.1 | 0.1 | 0.0 | 4.6 | 24.6 | 25.4 |
| | AEDR | 18.9 | 13.4 | 30.1 | 7.4 | 1.4 | 22.6 | 42.3 | 22.8 |
| | Ours (Finetuned w/o Aug) | 68.7 | 76.3 | 76.6 | 57.3 | 49.0 | 69.2 | 84.6 | 86.9 |
| | Ours (Finetuned w/ Aug) | **97.4** | **96.7** | **97.7** | **90.2** | **73.6** | **96.3** | **99.0** | **98.9** |
| 2/255 | Reconstruction | 1.4 | 1.8 | 4.5 | 0.4 | 0.0 | 1.3 | 13.6 | 12.5 |
| | LatentTracer | 3.4 | 3.4 | 11.1 | 0.0 | 0.0 | 2.4 | 18.4 | 19.5 |
| | AEDR | 10.7 | 6.9 | 18.2 | 3.2 | 0.2 | 14.1 | 28.7 | 14.4 |
| | Ours (Finetuned w/o Aug) | 0.3 | 0.6 | 0.6 | 0.0 | 0.0 | 0.3 | 1.9 | 2.8 |
| | Ours (Finetuned w/ Aug) | **49.3** | **43.6** | **51.1** | **15.9** | **2.7** | **40.0** | **64.7** | **60.7** |

Table A19: **Multi-source attribution performance across difference models.** We present TPR@1%FPR where all non-belong datasets are mixed for a given model.

| Method | LlamaGen | RAR | Taming | VAR | Infinity | VQ-Diffusion |
|---|---|---|---|---|---|---|
| Reconstruction | 28.0 | 2.3 | 24.7 | 0.4 | 0.0 | 10.5 |
| LatentTracer | 92.8 | 1.0 | 69.8 | 1.6 | 0.2 | 97.3 |
| AEDR | 59.4 | 17.2 | 80.0 | 26.9 | 3.5 | 84.2 |
| Ours | **100.0** | **100.0** | **100.0** | **100.0** | **99.2** | **100.0** |

Grubbs' hypothesis test (Grubbs, 1949). For this, we formulate the following hypothesis $\mathcal{H}_0$ : *The test sample does not belong to the given model.* leveraging Grubbs's hypothesis test which

Table A20: **Grubbs' Hypothesis Testing Results.** We report the TP, FP, TN, FN, TPR and FPR for model attribution. We test 1,000 belonging images and 1,000 non-belonging images randomly sampled across all datasets with $\alpha = 0.01$.

| Model | TP | FP | TN | FN | TPR (%) | FPR (%) |
|---|---|---|---|---|---|---|
| LlamaGen | 995 | 0 | 1000 | 5 | 99.5 | 0.0 |
| RAR | 999 | 0 | 1000 | 1 | 99.9 | 0.0 |
| Taming | 1000 | 1 | 999 | 0 | 100.0 | 0.1 |
| VAR | 1000 | 0 | 1000 | 0 | 100.0 | 0.0 |
| Infinity | 993 | 0 | 1000 | 7 | 99.3 | 0.0 |
| VQ-Diffusion | 999 | 0 | 1000 | 1 | 99.9 | 0.0 |

Table A21: **Acceleration options of the optimized quantization (Algorithm 3).** The non-belonging data is from ImageNet. We report the TPR@1%FPR and seconds required per image.

| Setting | Iterations | TPR@1%FPR (%) | Time |
|---|---|---|---|
| Default | 1200 | 95.0 | 8.24s |
| Less Iterations | 100 | 87.5 | 0.57s |
| Accelerated with Torch | 1200 | 94.8 | 7.79s |

rejects $\mathcal{H}_0$ if the following inequality holds:

$$\frac{x - \mu}{\sigma} < \frac{N-1}{\sqrt{N}} \sqrt{\frac{(t_{\alpha/N, N-2})^2}{N - 2 + (t_{\alpha/N, N-2})^2}} \tag{15}$$

Whereby $\mu$ and $\sigma$ are the mean and the standard deviation of a given belonging dataset, $x$ is the queried data sample and $N$ the number of samples of the belonging dataset. In Table A20 we report the result of applying Grubbs' hypothesis test on 1,000 belonging and 1,000 non-belonging samples across all datasets for each model. We find that, for all models, we achieve a TPR over 99% by only a single false positive for Taming.

## U    ACCELERATION FOR OPTIMIZED QUANTIZATION

We provide two acceleration options to reduce the latency of the optimized quantization has a relatively and report the results in Algorithm 3. First, our algorithm benefits from using quicker engineering implementations. By using the Einstein summation convention for calculating the codebook distance and using torch.compile to optimize the calculation of the feature map. These two techniques reduced the runtime of our method from 8.24s/image to 7.79s/image. The algorithm may be further accelerated with new developments in the deep learning toolkit. Second, Our method still maintains high detection performance and can be accelerated a lot with fewer iterations. We show that our method still achieves 87.5%TPR@1% FPR with only 100 iterations. This reduced the runtime from 8.24s/image to 0.57s/image. The results are shown in Table A21.

## V    EVALUATION FOR AR ATTRIBUTION

Although our approach is primarily designed for model family (autoencoder) attribution, we extend our evaluation to AR attribution settings, where multiple AR models share the same AE. We evaluate the following experimental settings.

### V.1    AR ATTRIBUTION WITH SHARED AE

We first evaluate scenarios where **two ARs share the same AE**. We consider the following settings within the LlamaGen model family:

Table A22: **AR attribution with shared AE.** We evaluate two settings where different AR models share the same autoencoder.

| Task | AE | AR | Belonging Data Generated by | Non-belonging Data Generated by | TPR@1%FPR (%) |
|---|---|---|---|---|---|
| Text-to-Image | LlamaGen-AE-T2I LlamaGen-AE-T2I | LlamaGen-T2I-COCO LlamaGen-T2I-Internal | LlamaGen-T2I-COCO LlamaGen-T2I-Internal | LlamaGen-T2I-Internal LlamaGen-T2I-COCO | 100.0 100.0 |
| Class-to-Image | LlamaGen-AE-C2I LlamaGen-AE-C2I | LlamaGen-L-256 LlamaGen-XL | LlamaGen-L-256 LlamaGen-XL | LlamaGen-XL LlamaGen-L-256 | 100.0 100.0 |

Table A23: **AR attribution with same AE architecture but different AE training data.** The AE fine-tuning data and evaluation belonging data are generated by different prompts (Text-to-Image) or different classes (Class-to-Image). Our method achieves 100% TPR@1%FPR in both settings.

| AE Architecture | AE Training Data | AR | Belonging Data Generated by | Non-belonging Data Generated by | TPR@1%FPR (%) |
|---|---|---|---|---|---|
| LlamaGen-AE LlamaGen-AE | COCO and Internal ImageNet | LlamaGen-T2I LlamaGen-C2I | LlamaGen-T2I LlamaGen-C2I | LlamaGen-C2I LlamaGen-T2I | 100.0 100.0 |

1. **LlamaGen Text-to-Image setting:** $AR_1$ is trained on LAION-COCO while $AR_2$ is trained on a 10M internal high-aesthetics quality dataset (Sun et al., 2024). Both models use LlamaGen-AE-T2I as the autoencoder.

2. **LlamaGen Class-to-Image setting:** $AR_1$ is LlamaGen-L-256 and $AR_2$ is LlamaGen-XL. Both models use LlamaGen-AE-C2I as the autoencoder.

The results in Table A22 demonstrate that our method achieves 100% TPR@1%FPR across all evaluated settings. This shows that fine-tuning the inverse decoder on images generated by one AR transformer can also help distinguish it from images generated by another AR using the same AE.

## V.2    AE WITH THE SAME ARCHITECTURE AND DIFFERENT TRAINING DATA

Although our method works very well in the above evaluations, we would like to point out that many ARs sharing exactly the same AE are less common in real-world scenarios. When model owners adapt an AE for their own task, it is more reasonable for the model owner to first train the existing AE on their own dataset, such that the AE performs better on their own dataset. For example, LlamaGen needs to train different AEs for their class-to-image image generation ("AE is trained on ImageNet") and text-to-image generation ("AE is trained on 50M LAION-COCO and 10M internal high aesthetic quality data"), as they use different datasets for the two tasks.

We provide the following case to show that our method can perfectly distinguish an AE with the same architecture trained on different datasets. As shown in Table A23, when the AE is trained on different data (COCO and Internal for Text-to-Image; ImageNet for Class-to-Image), our method maintains perfect attribution performance.

## V.3    AR ATTRIBUTION WITH SHARED AE AND CLASS-SPLIT EVALUATION

To evaluate whether our method generalizes beyond the specific classes used during fine-tuning, we design an experiment that combines the shared AE setting with a class-split evaluation setting. We use the LlamaGen Class-to-Image setting where the AE is LlamaGen-AE-C2I, AR model $A$ is LlamaGen-L-256, and AR model $B$ is LlamaGen-XL. We construct the following datasets:

- $\mathcal{D}_{A1}$: Generated by AR $A$ using classes 0–499
- $\mathcal{D}_{A2}$: Generated by AR $A$ using classes 500–999
- $\mathcal{D}_{B1}$: Generated by AR $B$ using classes 0–499
- $\mathcal{D}_{B2}$: Generated by AR $B$ using classes 500–999

Specifically, only $\mathcal{D}_{A1}$ is used to fine-tune the AE, while $\mathcal{D}_{A2}$, $\mathcal{D}_{B1}$, and $\mathcal{D}_{B2}$ are reserved only for evaluation. This setup tests whether our method can distinguish between AR models $A$ and $B$ on *unseen classes*. We evaluate three different settings with the above datasets and show the results in Table A24. The results reveal several important findings:

Table A24: **AR attribution with class-split evaluation.** Settings 1 and 2 evaluate cross-class generalization for AR attribution. The AE is fine-tuned only on $\mathcal{D}_{A1}$ (classes 0–499 from AR $A$), yet achieves 100% TPR@1%FPR when distinguishing $\mathcal{D}_{A2}$ from $\mathcal{D}_{B1}$ and $\mathcal{D}_{B2}$. Setting 3 confirms that the inverse decoder cannot distinguish images from different ARs when both are labeled as belonging, validating that our signal is AR-specific rather than class-specific.

| Setting | AE | AE Fine-tuning Data | Labeled as Belonging Data | Non-belonging Data | TPR@1%FPR (%) |
|---------|------|------|------|------|------|
| 1 | LlamaGen-AE-C2I | $\mathcal{D}_{A1}$ | $\mathcal{D}_{A2}$ | $\mathcal{D}_{B1}$ | 100.0 |
| 2 | LlamaGen-AE-C2I | $\mathcal{D}_{A1}$ | $\mathcal{D}_{A2}$ | $\mathcal{D}_{B2}$ | 100.0 |
| 3 | LlamaGen-AE-C2I | $\mathcal{D}_{A1}$ | $\mathcal{D}_{B1}$ | $\mathcal{D}_{B2}$ | 0.0 |

**Cross-class generalization (Settings 1 and 2).** Fine-tuning the AE on $\mathcal{D}_{A1}$ enables the inverse decoder to reliably distinguish $\mathcal{D}_{A2}$ (AR $A$, classes 500–999) from $\mathcal{D}_{B1}$ and $\mathcal{D}_{B2}$ (AR $B$, any classes), achieving 100% TPR@1%FPR. This demonstrates that an inverse decoder fine-tuned on *certain classes* of a given AR model can successfully invert images from *other classes* generated by the same model. Conversely, it cannot accurately invert images from a different AR model, regardless of the class.

**AR-specificity validation (Setting 3).** When we test whether $\mathcal{D}_{B1}$ and $\mathcal{D}_{B2}$ are distinguishable (both generated by AR $B$ but from different class ranges), the TPR@1%FPR drops to 0.0%. This confirms that training on $\mathcal{D}_{A1}$ does not improve inversion quality for either $\mathcal{D}_{B1}$ or $\mathcal{D}_{B2}$, as both originate from a different AR model. This result validates that our method captures AR-specific rather than class-specific patterns.

**Implications for model attribution.** These results support our design choice of focusing on model family (AE) attribution while demonstrating that finer-grained AR attribution is also achievable. The inverse decoder learns to recognize generation patterns specific to a particular AR transformer, which generalize across different input classes. This property is desirable for practical data provenance applications, where a model owner primarily seeks to determine whether an image was generated by their model family, independent of the specific content or class depicted.

## W    COMPARISON WITH MEMBERSHIP INFERENCE BASELINES

Table A25 We clarify the differences between our data provenance and membership inference attacks (MIAs) in Section 2, and explain that MIAs cannot be applied to our data provenance task because of the additional, over-strict requirements for the labels or prompts of a generated image. In this section, we would like to further explore what could be the **upper bound** of MIAs if given the additional information of labels for data provenance. Concretely, we provide the MIA-based methods with the ground truth labels for both generated and real images, which are usually absent in the real world. For the images generated by the class-to-image models, we use the conditional inputs of the models as the labels. For the real dataset, ImageNet, we directly use the ground truth label. Following the experimental setup in our work, we use the images generated by a given model as belonging images, and the other images, including the generated and natural datasets, as non-belonging images. We evaluate two MIA-based approaches that the reviewer mentioned: CFG-Diff (Kowalczuk et al., 2025) and ICAS (Yu et al., 2025). The TPR@1%FPR (%) for the two baselines and

Table A25: **TPR@1%FPR (%) of our method and two baseline methods based on membership inference attacks (MIAs).**. The two MIA-based approaches are CFG-Diff (Kowalczuk et al., 2025) and ICAS (Yu et al., 2025) The first column indicates the original model that has generated the belonging images, the heading of the other columns specifies the natural datasets or generators from which the non-belonging images are obtained.

| Model | Method | Natural | Generated | | | |
| --- | --- | --- | --- | --- | --- | --- |
| | | ImageNet | LlamaGen | RAR | Taming | VAR |
| RAR | CFG-Diff | 30.9 | 96.9 | - | **100.0** | 99.9 |
| | ICAS | 95.4 | 99.7 | - | 99.9 | 99.7 |
| | Ours | **100.0** | **99.9** | - | 99.9 | **100.0** |
| VAR | CFG-Diff | 2.5 | 6.2 | 16.4 | 54.9 | - |
| | ICAS | 7.1 | 24.4 | 44.7 | 66.0 | - |
| | Ours | **100.0** | **99.2** | **100.0** | **100.0** | - |

our method are shown as follows. The results demonstrate that our method outperforms the two MIA-based approaches in nearly every case, without the additional need for the ground truth labels. Notably, the two MIA-based approaches have a very low performance for VAR. They also have a lower TPR@1%FPR (%) when using the real images as non-belonging data than using generated images, which means that the MIA-based approaches tend to attribute many real images to one of the generative models. On the contrary, our method achieves low FPR, no matter what the non-belonging data is.

# X    LLM USAGE DECLARATION

Large language models (LLMs) were used solely to improve the clarity, grammar, spelling, and style of the manuscript. They were not employed to generate original research content, conduct data analysis, or modify the scientific meaning. All substantive ideas, interpretations, and conclusions are entirely those of the authors.

