# OpenReview forum: "Data Provenance for Image Auto-Regressive Generation"
_ICLR.cc/2026/Conference — ICLR 2026 Poster_

### Official Review · Reviewer_SetU · 2025-10-15

**Soundness:** 2
**Presentation:** 2
**Contribution:** 3
**Rating:** 4
**Confidence:** 5

**Summary:**

The paper propose a post-hoc data provenance method for image auto-regressive models (IARs), containing two stages. The first stage is training a inversion decoder to transform the image back to the feature map. The feature maps corresponding to the generated image can correspond more accurately to the codebook. than other images. The second stages  is calculating the metric for distinguishing whether the  image is generated by IARs. The experimental result shows the effectiveness of the proposed methods.

**Strengths:**

1. The experimental result shows the strong effectiveness to distinguish whether an image is generated by a given VAE decoder.
2. The EncLoss as a ratio of erros sounds interesting.

**Weaknesses:**

1. According to the provided code, the $D^{-1}$ is fine-tuned from the encoder. However, this process is not clearly described in the experimental setup.

2. The term “AR” (autoregressive) appears in the paper title and description, but no actual AR transformers are presented in the method. It seems the paper may intend to refer to *Data provenance for **Image VAE Generation*** rather than ***Image AR Generation.*** In the experiments, models with different VAEs are compared, but there are no results comparing AR models using the **same** VAE backbone—for example, VAR-d24 for belonging *vs.* VAR-d30 for non-belonging, or VAR *vs.* VAR-CCA [1]. Since the proposed method itself does not directly involve AR transformers, but only uses the images generated by the AR transformer when fine-tuning the inversion decoder, we cannot be sure whether this is effective in this case. Recent works on membership inference for IARs, such as [2] and [3], might provide relevant perspectives or complementary techniques. These methods calculate the metric using the loss of the AR transformers. It would further strengthen the paper if the authors could discuss their methods with in the Related Work section, and it is better to compare with them.

3. The reported value is too high. The evaluation protocol should therefore adopt stricter criteria, such as TPR@0.1%FPR, to provide a more reliable assessment of model robustness and discriminative power.

4. In line 812, the authors mention using the GPU with 48 GB of memory, while in line 856 it is stated as 40 GB. This inconsistency should be clarified.

[1] Chen, Huayu, et al. Toward guidance-free AR visual generation via condition contrastive alignment. ICLR 2025.

[2] Kowalczuk, Antoni, et al. Privacy attacks on image autoregressive models. ICML 2025.

[3] Yu H, et al. Icas: Detecting training data from autoregressive image generative models. ACM MM 2025.

**Questions:**

1. The difference between *Ours (EncLoss)* in Table 3 and Baseline *AEDR* is that the first one remove $Q$. Could the authors clarify why removing $Q$ leads to better performance?

2. According to the provided code, the random seed is fixed in the script to generate images. Is it the same when generating the evaluation dataset and fine-tuning dataset? Please check if there is a data leakage. Also, could you provide the result on ImageNet, using the first 500 classes for fine-tuning and last 500 classes for evaluation? This can help avoid the data leakage and verify the generalization of the proposed method.

3. Comparison of IAR for belonging and non-belonging with the same VAE. I provide several settings below, but you don’t need to run all of them. Just try one or two, or any other similar configurations you think make sense.

+ Models with difference sizes. For example, VARs have 4 difference sizes, including $d16, d20, d24, d30$. It is possible to compare $d16$ and $d24$
+ VAR and VAR-CCA [1]
+ LlamaGen, PAR [2] and LlamaGen-CCA [1]
+ ……

I would like to verify whether using only the images generated by the AR transformer for fine-tuning the inversion decoder is sufficient to distinguish belonging from non-belonging samples with the same VAE. If this approach proves effective, I would be inclined to raise my score.

For some other issues, please see "Weakness"

[1] Chen, Huayu, et al. Toward guidance-free AR visual generation via condition contrastive alignment. ICLR 2025.

[2] Wang, Yuqing, et al. "Parallelized autoregressive visual generation." CVPR 2025.

---

> ### Author Response · Authors · 2025-11-23
> **Responses to Reviewer SetU (1/5)**
>
> We thank the Reviewer for their detailed comments. We are happy that our method is recognized as interesting and highly effective.
>
> We provide detailed answers to all comments below. In summary, we performed additional experiments concerning AR attribution, evaluated our framework using TPR@0.1%FPR, and fine-tuned the inverse decoder on a class-split setup to show that it does not overfit to the training data.
>
> >**W1 The inverse decoder is fine-tuned from the encoder. However, this is not clearly described in the experimental setup.**
>
> We would like to point out that we stated this in lines 259-260 of the original submission: "*Concretely, we initialize this model’s weights with the original encoder weights and fine-tune this inverse decoder on images generated by the given IAR (see Equation (2)).*"
>
> >**W2.1 It seems the paper may intend to refer to Data provenance for Image VAE Generation rather than Image AR Generation.**
>
> >**Q3 Comparison of IAR for belonging and non-belonging with the same VAE.**
>
> We thank the Reviewer for the question and conduct the following experiments to address your concern about distinguishing ARs that share the same AE/VAE.
>
> We evaluate the following two settings where **two ARs share the same AE**:
>
> **1. LlamaGen Text-to-Image setting:** $AR_1$ trained on LAION-COCO vs $AR_2$ trained on 10M internal high-aesthetics quality dataset [C] (both using LlamaGen-AE-T2I)
>
> **2. LlamaGen Class-to-Image setting:** $AR_1$ is LlamaGen-L-256 vs $AR_2$ LlamaGen-XL (both using LlamaGen-AE-C2I)
>
> For both settings, the results show that our method distinguishes the images generated by different ARs with 100%TPR@1%FPR. For the evaluated settings, fine-tuning the inversion decoder on images from one AR transformer is indeed sufficient to distinguish it from another AR using the same AE.
>
> | Task | AE | AR | Belonging data is generated by | Non-belonging data is generated by | TPR@1%FPR (%) |
> |:---:|:---:|:---:|:---:|:---:|:---:|
> | Text-to-Image | LlamaGen-AE-T2I | LlamaGen-AR-T2I-COCO | LlamaGen-AR-T2I-COCO | LlamaGen-AR-T2I-Internal | 100.0 |
> |  |  LlamaGen-AE-T2I | LlamaGen-AR-T2I-Internal | LlamaGen-AR-T2I-Internal | LlamaGen-AR-T2I-COCO | 100.0 |
> | Class-to-Image | LlamaGen-AE-C2I | LlamaGen-L-256 | LlamaGen-L-256 | LlamaGen-XL | 100.0 |
> |  | LlamaGen-AE-C2I  | LlamaGen-XL | LlamaGen-XL | LlamaGen-L-256 | 100.0 |
>
>
> Although our method works very well in the above evaluations, we would like to point out that **the main focus of our work is to attribute an image to a IAR model family, not one specific AR transformer.** Notably, each model family designs a specific AE for their tasks. For example, a next-scale tokenizing paradigm is developed for the next-scale prediction in VAR, and a BSQ-based image autoencoder is designed for Infinity.
> Additionally, in our application to prevent model collapse, a model owner cares most about whether an image is generated by their model family to prevent re-training on these data, instead of by a specific transformer.
> Similar to the setup of our work, existing AI data provenance works, including RONAN, LatentTracer, and AEDR, focus on attributing an image to a model family instead of targeting a fine-grained model (like AR) attribution problem.
>
> Moreover, we would like to point out that **many ARs sharing exactly the same AE are less common in real-world scenarios.**
> When model owners adapt an AE for their own task, it is more reasonable for the model owner to first train the existing AE on their own dataset, such that the AE performs better on their own dataset.
> For example, LlamaGen needs to train different AEs for their **class-to-image** image generation (*”AE is trained on ImageNet”*) and **text-to-image** generation (*”AE is trained on 50M LAION-COCO and 10M internal high aesthetic quality data”*), as they use different datasets for the two tasks.
>
> We provide the following case to show that our method can perfectly distinguish an AE with the same architecture trained on different datasets.
>
> **Table. Same model family, AE finetuned on different datasets.** For the Text-to-Image setting, the AE fine-tuning data and evaluation belonging data are generated by different prompts. Similarly, for the Class-to-Image setting, the AE fine-tuning data and evaluation belonging data are generated from different classes.
>
> | AE Architecture | AE Training Data | AR | Belonging data is generated by | Non-belonging data is generated by | TPR@1%FPR (%) |
> |:---:|:---:|:---:|:---:|:---:|---:|
> | LlamaGen-AE | COCO and Internal | LlamaGen-T2I | LlamaGen-T2I | LlamaGen-C2I | 100.0 |
> | LlamaGen-AE | ImageNet | LlamaGen-C2I | LlamaGen-C2I | LlamaGen-T2I | 100.0 |
>
> In conclusion, we 1) provide experiments for two settings where our method successfully attributes between two ARs sharing the same AE, 2) clarify that we mainly focus on model family attribution, and 3) show that our method can distinguish two IARs within the same model family but with differently trained AEs.

---

> ### Author Response · Authors · 2025-11-23
> **Responses to Reviewer SetU (2/5)**
>
> >**W2.2 Recent works on membership inference for IARs, such as [2] and [3], might provide relevant perspectives or complementary techniques.**
>
> We would like to thank the Reviewer for the question. However, we would like to point out that (1) membership inference and dataset provenance are two distinct tasks and (2) membership inference signals are not applicable to data provenance.
>
> **1. Membership inference and dataset provenance are two distinct tasks.**
>
> *Membership inference is focused on the data used to train a given model, while the data provenance is to detect the data generated by the model.* In other words, membership inference attack addresses attribution in the training process, and the data provenance addresses attribution in the generation process.
>
> Below, we present a more specific and detailed setting as well as the further explanation.
>
> The general procedure of a generative model can be defined as:
>
> $$\text{Training Data } \xrightarrow{\text{train}} \text{Generative Model} \xrightarrow{\text{generate}} \text{Synthetic Data}$$
>
>
> * **Membership inference:** detect the **training data** of a model. Given a data point (usually a natural data point) and a model, we detect if this data point was used to **train** this model. The goal of the membership inference attack is to audit the privacy of the training procedure and infer whether a given model leaks information about its private training data.
>
> * **Data provenance:** detect the **generated data** of a model. Given a data point (we do not know if it is natural or synthetic) and a model, we detect if this model **generated** this data point. The goal of data provenance is to trace where the given data is from, especially to trace the source of synthetic images or prevent model collapse caused by training on synthetic data [A][B].
>
> **2. Membership inference signals are not applicable to data provenance.**
>
> In the case of membership inference methods for image generative models, it is necessary to have access to both the class labels or the prompts and the images themselves [1,2] in order to collect membership signals from the models. However, this is not the case for the data provenance task, since **generated images found online are usually not accompanied by their original class information or prompt**, thus, we have to find their origin by relying only on the image itself. On the contrary, our method only needs the image itself for attribution and does not require any additional information, making our method more applicable in real-world scenarios.
>
> We note that the membership inference is also not a solution for AR attribution. For example, the two LlamaGen AR variants for the C2I task are both trained on ImageNet, so the membership signals on both AR should reflect ImageNet as the member data. Therefore, these MIA metrics cannot serve as an AR attribution signal.
>
> We also update the above discussions in the Related Work of our revised manuscript.

---

> ### Author Response · Authors · 2025-11-23
> **Responses to Reviewer SetU (3/5)**
>
> >**W3 The reported value is too high. The evaluation protocol should therefore adopt stricter criteria, such as TPR\@0.1%FPR.**
>
> To extend our reported performance metric, we calculated TPR\@0.5%FPR (Table A8), TPR\@0.1%FPR (Table A9), as well as the AUC (Table A10) and plot the ROC (Figure A5). The results show that our methods perform consistently well, even when evaluated under stricter settings. For the Reviewer's convenience we also display the results for TPR\@0.1%FPR below.
>
> **Table. TPR\@0.1%FPR of our method and the baselines**
>
> | Model | Method | Natural | | | Generated| | | | | |
> |---|---|---|---|---|---|---|---|---|---|---|
> | | | **ImageNet** | **LAION** | **MS-COCO** | **LlamaGen** | **RAR** | **Taming** | **VAR** | **Infinity** | **VQDiff** |
> | **LlamaGen** | Reconstruction | 13.2 | 15.4 | 18.5 | - | 17.2 | 0.3 | 21.9 | 47.3 | 17.9 |
> | | LatentTracer | 79.7 | 75.1 | 85.9 | - | 90.0 | 63.5 | 89.3 | 95.4 | 91.8 |
> | | AEDR | 22.4 | 23.7 | 17.7 | - | 31.1 | 38.0 | 45.4 | 57.7 | 46.3 |
> | | **Ours** | **100.0** | **100.0** | **100.0** | - | **100.0** | **99.9** | **100.0** | **100.0** | **100.0** |
> | **RAR** | Reconstruction | 1.8 | 1.8 | 1.4 | 0.1 | - | 0.0 | 1.6 | 7.4 | 3.9 |
> | | LatentTracer | 0.4 | 0.2 | 0.9 | 0.0 | - | 0.0 | 0.7 | 4.1 | 0.4 |
> | | AEDR | 1.4 | 1.4 | 9.1 | 0.0 | - | 0.0 | 16.6 | 6.7 | 2.6 |
> | | **Ours** | **99.9** | **99.9** | **100.0** | **96.9** | - | **54.1** | **99.9** | **100.0** | **100.0** |
> | **Taming** | Reconstruction | 11.8 | 11.0 | 12.9 | 3.0 | 12.7 | - | 19.1 | 30.7 | 16.4 |
> | | LatentTracer | 36.8 | 33.1 | 54.7 | 24.4 | 51.3 | - | 66.3 | 71.1 | 100.0 |
> | | AEDR | 58.3 | 35.6 | 68.1 | 30.9 | 75.5 | - | 55.2 | 77.7 | 76.3 |
> | | **Ours** | **100.0** | **92.6** | **100.0** | **100.0** | **100.0** | - | **100.0** | **100.0** | **100.0** |
> | **VAR** | Reconstruction | 0.3 | 0.2 | 0.4 | 0.0 | 0.5 | 0.0 | - | 1.9 | 0.3 |
> | | LatentTracer | 0.2 | 0.0 | 3.4 | 0.0 | 1.8 | 0.0 | - | 3.4 | 2.4 |
> | | AEDR | 0.0 | 1.6 | 29.5 | 0.6 | 3.8 | 1.2 | - | 16.8 | 25.4 |
> | | **Ours** | **99.6** | **86.3** | **99.8** | **95.8** | **100.0** | **98.9** | - | **100.0** | **100.0** |
> | **Infinity** | Reconstruction | 0.0 | 0.0 | 0.1 | 0.0 | 0.0 | 0.0 | 0.0 | - | 0.0 |
> | | LatentTracer | 0.0 | 0.1 | 7.1 | 0.0 | 0.0 | 0.0 | 0.9 | - | 0.5 |
> | | AEDR | 0.0 | 0.0 | 4.7 | 0.0 | 1.9 | 0.0 | 3.5 | - | 3.0 |
> | | **Ours** | **98.3** | 0.0 | **99.4** | **0.0** | **99.4** | **31.2** | **99.1** | - | **99.1** |
> | **VQ-Diffusion** | Reconstruction | 2.0 | 3.3 | 2.7 | 1.3 | 6.2 | 0.1 | 10.8 | 15.1 | - |
> | | LatentTracer | 91.8 | **86.3** | 95.0 | 92.5 | 96.0 | 78.4 | 97.2 | 96.9 | - |
> | | AEDR | 54.9 | 20.7 | 59.1 | 32.2 | 76.9 | 48.0 | 57.6 | 57.5 | - |
> | | **Ours** | **99.9** | 55.6 | **100.0** | **84.8** | **100.0** | **99.0** | **99.5** | **100.0** | - |
>
> When evaluated under stricter settings with TPR\@0.1%FPR, baseline methods have a very limited performance in most cases, while our methods perform consistently well.

---

> ### Author Response · Authors · 2025-11-23
> **Responses to Reviewer SetU (4/5)**
>
> >**W4 In line 812, the authors mention using the GPU with 48 GB of memory, while in line 856 it is stated as 40 GB. This inconsistency should be clarified.**
>
> We corrected the typo in line 856 as “48GB”. Thank you for reading our submission so carefully, we really appreciate it!
>
>
> >**Q1 The difference between Ours (EncLoss) in Table 3 and Baseline AEDR is that the first one remove Q. Could the authors clarify why removing Q leads to better performance?**
>
> Thank you for this question. We would like to clarify that *the key difference is not simply "removing $Q$"*, but rather a conceptual difference with a similar format.
>
> Given the reconstruction process for a VQ-VAE: $x’ = D(Q(E(x)))$:
>
> The EncLoss of our method can be written as **$||x-D(D^{-1}(x))||_2$**, where we obtain an inverse decoder $D^{-1}$ via minimizing Equation.6. Our assumption is that, if a belonging image can be inverted to its original feature map by $D^{-1}$, it can be decoded into exactly the same image by $D$. Our aim is to **invert the decoding stage** and then decode the image again, so the original encoder $E$ and the quantization stage $Q$ are not involved in this process.
>
> In contrast, the reconstruction loss in AEDR should be written as **$||x-D(Q(E(x)))||_2$**. Their assumption is, a belonging image should be better reconstructed by the whole encoding, quantization, and decoding process of the autoencoder. This is only a naive forward pass for the reconstruction process.

---

> ### Author Response · Authors · 2025-11-23
> **Responses to Reviewer SetU (5/5)**
>
> >**Q2 According to the provided code, the random seed is fixed in the script to generate images.  Also, could you provide the result on ImageNet, using the first 500 classes for fine-tuning and last 500 classes for evaluation?**
>
> *The random seed is fixed:* We use different seeds for generating the evaluation dataset and fine-tuning dataset. We have checked the images carefully and made sure that there is no data leakage.
>
> *Using the first 500 classes for fine-tuning and the last 500 classes for evaluation:* We performed an experiment according to the Reviewer’s suggestion, where we use the first 500 classes to generate the fine-tuning set and use the remaining 500 classes for evaluation. We denote this specific split as Ours (class split) and our standard split as Ours (random split). We report the TPR@1%FPR of belonging vs non-belonging data. The results highlight that our method performs consistently well across the two settings and outperforms the baseline methods significantly.
>
> | Model | Method | Natural |  |  | Generated |  |  |  |  |  |
> |:---:|:---:|:---:|:---:|:---:|:---:|:---:|:---:|:---:|:---:|:---:|
> |  |  | ImageNet | LAION | MS-COCO | LlamaGen | RAR | Taming | VAR | Infinity | VQDiff |
> | RAR| Reconstruction | 3.8 | 4.1 | 7.4 | 0.8 | - | 0.1 | 5.7 | 18.1 | 18.8 |
> |  | LatentTracer | 6.0 | 6.1 | 15.2 | 0.4 | - | 0.0 | 9.3 | 24.6 | 26.9 |
> | | AEDR | 29.5 | 16.6 | 36.6 | 10.6 | - | 2.3 | 35.9 | 49.9 | 27.6 |
> |  | Ours (random split) | 100.0 | 100.0 | 100.0 | 99.9 | - | 99.9 | 100.0 | 100.0 | 100.0 |
> |  | Ours (class split) | 99.8 | 99.9 | 100.0 | 99.4 | - | 99.7 | 99.7 | 100.0 | 100.0 |
> |Taming | Reconstruction | 27.5 | 21.5 | 27.6 | 10.1 | 18.9 | - | 27.7 | 39.0 | 46.1 |
> |  | LatentTracer | 73.0 | 61.0 | 75.9 | 36.4 | 66.8 | - | 76.0 | 85.4 | 87.4 |
> |  | AEDR | 80.4 | 82.5 | 81.9 | 70.7 | 80.7 | - | 78.1 | 91.9 | 87.5 |
> |  | Ours (random split) | 100.0 | 100.0 | 100.0 | 100.0 | 100.0 | - | 100.0 | 100.0 | 100.0 |
> |  | Ours (class split) | 100.0 | 100.0 | 100.0 | 100.0 | 100.0 | - | 100.0 | 100.0 | 100.0 |
> |VAR | Reconstruction | 1.4 | 1.4 | 3.6 | 0.1 | 1.6 | 0.0 | - | 5.9 | 5.9 |
> |  | LatentTracer | 3.9 | 1.3 | 12.0 | 0.2 | 5.6 | 0.1 | - | 15.4 | 15.3 |
> | | AEDR | 29.1 | 15.7 | 50.6 | 14.7 | 28.3 | 14.0 | - | 37.5 | 50.8 |
> |  | Ours (random split) | 100.0 | 99.2 | 100.0 | 99.2 | 100.0 | 100.0 | - | 100.0 | 100.0 |
> |  | Ours (class split) | 99.9 | 98.9 | 100.0 | 99.4 | 100.0 | 99.1 | - | 100.0 | 99.9 |
>
>
> While we understand the Reviewer’s question about generalization, we also note that only using 500 classes to generate the fine-tuning set is unlikely to happen in the real world. As a model owner or given an open-source model, we can generate images covering all classes to fine-tune the inverse decoder.
>
> To further demonstrate the generalization of our method, we provided further experiments where the conditional guidance scales and sampling temperatures are also different during generating fine-tuning and evaluation sets. We use CFG=4 and temperature=1.0 for generating the fine-tuning set. We observe that our method achieves high performance across different CFG (3,4,5) and temperatures (0.8, 1.0, 1.2).
>
> | CFG | Temperature | Natural | | | Generated | | | | |
> |:---:|:---:|:---:|:---:|:---:|:---:|:---:|:---:|:---:|:---:|
> | | | ImageNet | LAION | MS-COCO | LlamaGen | Taming | VAR | Infinity | VQDiff |
> | 3 | 0.8 | 99.7 | 99.8 | 99.7 | 99.6 | 99.4 | 100.0 | 99.8 | 100.0 |
> |  | 1.0 | 100.0 | 100.0 | 100.0 | 99.5 | 99.4 | 100.0 | 100.0 | 100.0 |
> | | 1.2 | 99.9 | 99.9 | 100.0 | 99.8 | 99.8 | 99.9 | 100.0 | 100.0 |
> |**4**| 0.8 | 99.5 | 99.8 | 99.8 | 99.3 | 99.4 | 99.5 | 99.8 | 99.9 |
> |  | **1.0** | 100.0 | 100.0 | 100.0 | 99.9 | 99.9 | 100.0 | 100.0 | 100.0 |
> | | 1.2 | 100.0 | 99.6 | 100.0 | 99.5 | 98.8 | 100.0 | 100.0 | 100.0 |
> | 5| 0.8 | 100.0 | 99.6 | 100.0 | 99.5 | 98.8 | 100.0 | 100.0 | 100.0 |
> | | 1.0 | 99.9 | 99.0 | 100.0 | 98.4 | 98.9 | 100.0 | 100.0 | 100.0 |
> | | 1.2 | 99.7 | 99.4 | 100.0 | 99.4 | 99.3 | 99.5 | 100.0 | 99.8 |
>
>
> In addition, we would like to clarify that all the experiments about text-to-image models (Infinity and VQDiffusion) use different prompts for generating the fine-tuning and evaluation sets.
>
>
> **References**
>
> [A] Alemohammad, Sina, et al. "Self-consuming generative models go mad." ICLR. 2023.
>
> [B] Shumailov, Ilia, et al. "AI models collapse when trained on recursively generated data." Nature. 2024
>
> [C] Peize Sun, Yi Jiang, Shoufa Chen, Shilong Zhang, Bingyue Peng, Ping Luo, and Zehuan Yuan. “Autoregressive model beats diffusion: Llama for scalable image generation.” 2024.
>
>  ***
>
> We thank the Reviewer for their valuable questions, which have contributed to making our submission more robust and clear. If our rebuttal addresses the Reviewer's concerns, we would appreciate it very much if they would consider updating the score. We are also happy to address any remaining questions.

---

> > ### Comment · Reviewer_SetU · 2025-11-23
> >
> > Thanks for your rebuttal. Your response has addressed most of my concerns, and I have raised my score to 6.
> >
> > I have one additional question.
> >
> > Suppose AR models A and B share the same AE. The generated images for the first 500 classes are denoted as $D_{A1}$ and $D_{B1}$. The generated images for the last 500 classes are denoted as $D_{A2}$ and $D_{B2}$. Can the fine-tuning on $D_{A1}$ also distinguish well between $D_{B1}$ and $D_{B2}$? (Just combining the two experiment settings: the same AE and the split classes.) The outcome will not affect my score, whether the result is good or bad. I am simply curious about whether the result can help answer the following question: Distributions $D_{A1}$ and $D_{B1}$ differ, and distributions $D_{A1}$ and $D_{B2}$ also differ, will the nature of these two differences be substantially different from each other?
> >
> > A suggestion regarding the paper: your method applies to all discrete generative models, including autoregressive models and discrete diffusion models (both uniform and absorbing-state variants). Therefore, you might consider revising the description to “Data Provenance for Image Discrete Generation” instead of “AR Generation”.

---

> > > ### Author Response · Authors · 2025-11-27
> > > **Response to Reviewer SetU: Class Split Experiments and Scope Change**
> > >
> > > We thank the Reviewer for increasing the rating and the further interest in our work!
> > >
> > > >**Combining the two experiment settings: the same AE and the split classes.**
> > >
> > > Regarding the Reviewer’s question of splitting the classes, we perform the following experiments.
> > >
> > > We follow the LlamaGen Class-to-Image setting in our previous response (AE is LlamaGen-AE-C2I, AR model $A$ is LlamaGen-L, and AR model $B$ is LlamaGen-XL) and generate the following datasets for our experiments:
> > >
> > > $D_{A1}$: generated by AR $A$, 0-499 classes
> > >
> > > $D_{A2}$: generated by AR $A$, 500-999 classes
> > >
> > > $D_{B1}$: generated by AR $B$, 0-499 classes
> > >
> > > $D_{B2}$: generated by AR $B$, 500-999 classes
> > >
> > > Among these datasets, only $D_{A1}$ is used for fine-tuning the AE (after generating the above datasets), and $D_{A2}$, $D_{B1}$, and $D_{B2}$ are only used for evaluation. We show the results for **distinguishing AR $A$ and AR $B$ with split classes** as follows:
> > >
> > > | Setting | AE | AE Finetuning Data | Labeled as Belonging Data | Non-belonging Data | TPR@1%FPR (%) |
> > > |---------|---------|-------------------|----------------|-------------------|---------------|
> > > | 1       | LlamaGen-AE-C2I | $D_{A1}$  | $D_{A2}$      | $D_{B1}$         | 100.0         |
> > > | 2       | LlamaGen-AE-C2I |  $D_{A1}$ | $D_{A2}$      | $D_{B2}$         | 100.0         |
> > >
> > > Results from Setting 1 and 2 demonstrate that fine-tuning AE on $D_{A1}$ can make the inverse decoder reliably distinguish $D_{A2}$ vs $D_{B1}$ and $D_{A2}$ vs $D_{B2}$. This indicates that an inverse decoder finetuned on **certain classes** of a given model can also invert **the other classes** generated by the same model. On the contrary, this inverse decoder cannot invert images from another model very well, no matter what classes they are from.
> > >
> > > We also show the following results to **test if  $D_{B1}$ and $D_{B2}$ are distinguishable**. Regarding the following results on Setting 3, we **label $D\_{B1}$ as belonging data** and $D_{B2}$ as non-belonging data, only to test if the two sets are distinguishable. Note that real belonging data should always be generated by AR $A$, and we label $D\_{B1}$ as belonging data only for the convenience of description.
> > >
> > > | Setting | AE | AE Finetuning Data | **"Labeled"** as Belonging Data | Non-belonging Data | TPR@1%FPR (%) |
> > > |---------|---------|-------------------|----------------|-------------------|---------------|
> > > | 3  | LlamaGen-AE-C2I |  $D_{A1}$ | $D_{B1}$      | $D_{B2}$         | 0.0   |
> > >
> > >
> > > Results from Setting 3 show that training on $D_{A1}$ cannot make the inverse decoder distinguish $D_{B1}$ vs $D_{B2}$. Even though $D_{A1}$ and $D_{B1}$ are from the same classes, they are generated by **different models**. Therefore, training on $D_{A1}$ does not make the inverse decoder have better inversion for $D_{B1}$ (or $D_{B2}$), so there is no detectable difference between $D_{B1}$ and $D_{B2}$.
> > >
> > > In summary, we think that the inverse decoder is trained to better invert images generated **by a certain model**, but **not from certain classes**. We think this is a more desirable property for image attribution, as we conceptually want to attribute an image to its source model, regardless of which class the image is from. This observation is also validated by the results in our previous rebuttal (Q2).
> > >
> > >
> > > > **your method applies to all discrete generative models, including autoregressive models and discrete diffusion models (both uniform and absorbing-state variants). Therefore, you might consider revising the description to “Data Provenance for Image Discrete Generation” instead of “AR Generation”**
> > >
> > > We thank the Reviewer for the suggestion. Indeed, our method is applicable to both image autoregressive models and vector quantized diffusion, so changing the title to “discrete image generation” or “vector quantized image generation” have a larger scope and might potentially better place our work.
> > >
> > > ***
> > >
> > > We believe our discussion with the reviewer and our additional experiments really helped us improve our work. We would like to thank the reviewer again for the insightful discussion and suggestions.

---

> > > > ### Comment · Reviewer_SetU · 2025-11-28
> > > >
> > > > What a meaningful finding! This findings reveals an inherent connection between images generated by the same generative model. You can add this finding to the PDF.
> > > >
> > > > A minor error: In Table A20, 94.8s and 7.79 should be corrected to 94.8 and 7.79s.

---

> > > > > ### Author Response · Authors · 2025-12-03
> > > > > **Updated our submission**
> > > > >
> > > > > >**What a meaningful finding! This findings reveals an inherent connection between images generated by the same generative model. You can add this finding to the PDF.**
> > > > >
> > > > > We would like to thank the Reviewer for the insightful comments and for commenting on our observation as “a meaningful finding”! We have included the additional experiments in the updated manuscript as the Reviewer suggested.
> > > > >
> > > > > >**A minor error: In Table A20, 94.8s and 7.79 should be corrected to 94.8 and 7.79s.**
> > > > >
> > > > > Thank you for spotting it. We corrected the typo in the updated our submission.

---

> ### Author Response · Authors · 2025-12-03
> **Additional Baselines: MIA-based Methods**
>
> Regarding the Reviewer’s suggestion for adding evaluations on two MIA-based methods (W2), we clarified that membership inference attacks (MIAs) and our generated data provenance are two different tasks, and that MIAs cannot be applied to our task because of the additional, over-strict requirements for the labels or prompts of a generated image. In this response, we further explore what could be the **upper bound** of MIAs if given the additional information of labels for data provenance. Concretely, we provided the MIA-based methods with **the ground truth labels for both generated and real images**, which are usually **absent in the real world**. We evaluated two MIA-based approaches that the Reviewer mentioned: CFG-Diff [1] and ICAS [2]. The TPR\@1%FPR (\%) for the two baselines and our method are shown as follows.
>
>
> | Model | Method   | ImageNet | LlamaGen | RAR   | Taming | VAR   |
> |-------|----------|----------|----------|-------|--------|-------|
> | RAR   | CFG-Diff | 30.9     | 96.9     | -     | **100.0** | 99.9  |
> |       | ICAS     | 95.4     | 99.7     | -     | 99.9   | 99.7  |
> |       | Ours     | **100.0**| **99.9** | -     | 99.9   | **100.0** |
> | VAR   | CFG-Diff | 2.5      | 6.2      | 16.4  | 54.9   | -     |
> |       | ICAS     | 7.1      | 24.4     | 44.7  | 66.0   | -     |
> |       | Ours     | **100.0**| **99.2** | **100.0** | **100.0** | -     |
>
> The results demonstrate that **our method outperforms the two MIA-based approaches in nearly every case, even if we evaluate the upper bound performance of these methods by providing ground truth labels**. Notably, the two MIA-based approaches have a very low performance for VAR. They also have a lower TPR\@1%FPR (\%) when using the real images as non-belonging data than using generated images, which means that **the MIA-based approaches tend to attribute many real images to one of the generative models**. On the contrary, our method achieves low FPR, regardless of the types of non-belonging data.
>
> **References**
>
> [1] Kowalczuk, Antoni, et al. "Privacy Attacks on Image AutoRegressive Models." ICML 2025.
>
> [2] Yu, Hongyao, et al. "Icas: Detecting training data from autoregressive image generative models." ACM MM 2025.

---

### Official Review · Reviewer_891z · 2025-10-25

**Soundness:** 3
**Presentation:** 3
**Contribution:** 3
**Rating:** 6
**Confidence:** 2

**Summary:**

This paper proposes a post-hoc provenance framework for detecting and attributing images generated by Image Autoregressive Models (IARs). The key insight is that IAR-generated images leave characteristic patterns in their token representations due to the quantization process - generated images have feature representations closer to codebook entries than natural images. The authors design two provenance signals: QuantLoss (measuring distance to codebook entries) and EncLoss (measuring encoder-decoder reconstruction error), combined with a finetuned inverse decoder. Experiments on six IAR models (LlamaGen, RAR, Taming, VAR, Infinity, VQ-Diffusion) demonstrate near-perfect detection rates (≈100% TPR@1%FPR) and reasonable robustness to image post-processing.

While the empirical results are strong and the problem is timely, the work suffers from limited technical novelty, lacks theoretical depth, and makes strong assumptions (white-box access, per-model finetuning) that significantly limit practical applicability. The core contribution is primarily an engineering adaptation of reconstruction-based detection to the IAR setting.

**Strengths:**

1. The paper addresses an important and timely problem. As IAR models gain traction for image generation, reliable provenance methods become critical for combating misinformation and ensuring accountability.

2. The experimental evaluation is comprehensive, covering multiple state-of-the-art IAR architectures (next-token, next-scale, random-order prediction) and demonstrating consistently high detection rates across diverse test scenarios.

3. The post-hoc nature of the approach is valuable - no modification to the generation process is required, making it applicable to already-published content.

4. The paper includes thorough ablation studies (Table 3, Table 4) that validate the importance of different components, particularly the decoder inversion step.

5. Robustness evaluation against common image perturbations (JPEG compression, resizing, noise, etc.) is extensive, with augmentation-based training showing improved robustness.

6. The writing is generally clear and the paper is well-organized with comprehensive appendices.

**Weaknesses:**

1. Limited novelty. The core idea of using reconstruction error for detecting generated images is well-established. The main contribution is applying this to IARs by exploiting quantization artifacts. While the domain-specific adaptation is useful, the conceptual advance is incremental. The optimized quantization algorithm (Algorithm 3) for multi-scale IARs is the most novel technical component, but receives insufficient analysis.

2. Strong practical limitations undermine the claimed applicability:
   - Requires white-box access (encoder E, decoder D, codebook Z) to the target model, which may not be available for commercial or closed-source IARs.
   - Requires per-model finetuning (10-50 epochs, 10K-50K images) despite being called "post-hoc". This is computationally expensive and contradicts the claim of broad applicability to "already published content" without additional training data.
   - High computational cost for some models (VAR: 8.249 sec/image, Table A3) limits scalability.

3. Insufficient theoretical grounding:
   - Why does finetuning the encoder to invert the decoder improve attribution? The paper provides intuition (Eq. 6) but no theoretical analysis of what properties D^{-1} should satisfy.
   - Why are QuantLoss and EncLoss "orthogonal"? Only empirical observations are provided.
   - The choice of multiplicative combination in Eq. 10 lacks justification. Why not additive or learned weighting?
   - No analysis of failure modes or theoretical performance guarantees.

4. Experimental design concerns:
   - The setting is somewhat artificial: 1000 clean images at 256×256 resolution, binary classification against single sources. Real-world scenarios involve mixed resolutions, multiple processing steps, and multi-class attribution.
   - Finetuning uses generated images from the same distribution as test images, which may lead to overfitting. Cross-distribution generalization (e.g., different prompts, generation hyperparameters) is not evaluated.
   - Only single performance metric (TPR@1%FPR) is reported. Full ROC curves and AUC would provide better insight.
   - No statistical significance testing (error bars, confidence intervals) despite stochastic generation processes.

5. The augmentation-based robustness training has concerning properties. As acknowledged in Appendix H, augmentation training improves QuantLoss but degrades EncLoss performance. This suggests the approach lacks principled robustness and relies on dataset-specific overfitting. The explanation provided (training D^{-1} to "remove" augmentation) indicates a fundamental issue with the EncLoss design.

6. Limited baseline comparisons. The paper only compares against LatentTracer (diffusion-specific), AEDR (also diffusion-focused), and naive reconstruction. Comparisons with general deepfake detection methods or universal AI-generated image detectors are missing.

7. The AE attribution vs AR attribution discussion (Appendix I) reveals a conceptual issue: the method attributes to the autoencoder, not the autoregressive model. Multiple AR models sharing the same AE are indistinguishable, which is a significant limitation not adequately addressed in the main paper.

8. Presentation issues:
   - 24-page appendix is excessive; key results (Table A4, A5) should be in the main paper.
   - Some claims are overclaimed: "first post-hoc provenance framework" ignores reconstruction-based methods like RONAN.
   - Algorithm 3 lacks critical details (learning rate, convergence criteria, initialization strategy).

**Questions:**

1. How does the method perform when the test images come from different generation settings (different sampling temperatures, guidance scales, or prompts) than the finetuning data? This would test true generalization ability.

2. Can you provide theoretical analysis or empirical evidence on what properties the inverse decoder D^{-1} learns? For example, does it learn to undo specific decoder operations, or does it learn a direct mapping to the codebook?

3. Have you tested adversarial robustness? An adversary aware of your method could potentially add carefully crafted noise to increase QuantLoss for generated images or decrease it for natural images.

4. For the multi-scale VAR model, how sensitive is Algorithm 3 to hyperparameter choices (number of iterations N_{iters}, learning rate, initialization)? The current description is insufficient for reproduction.

5. Why not learn the combination weights between QuantLoss and EncLoss instead of using a fixed multiplicative combination? A learned weighted sum might be more principled.

6. How does the method perform on images that have undergone multiple sequential transformations (e.g., JPEG compression, then resize, then Gaussian blur)? Table 2 only shows single transformations.

7. Can you clarify the finetuning data generation process? Are images generated with the same prompts/settings as test data, or is there diversity? This is critical for understanding potential overfitting.

8. In Table 1, why does the method achieve 100% TPR@1%FPR for most settings but drops significantly for Infinity vs LAION (85.6%)? What is special about this case?

9. The overhead of finetuning is claimed to be "relatively small", but 50 epochs on 50K images is substantial. Can you quantify the total computational cost (GPU hours) and compare it to the cost of training the IAR itself?

10. Have you considered zero-shot or few-shot scenarios where only a small number of generated samples are available for finetuning? This would be more realistic for detecting newly-released or proprietary models.

---

> ### Author Response · Authors · 2025-11-23
> **Responses to Reviewer 891z (1/13)**
>
> We thank the Reviewer for the valuable feedback. We appreciate that the Reviewer finds our work timely and important, the post-hoc nature of our method valuable, and our evaluation comprehensive.
>
> In summary, we extended our evaluation to include AUC and ROC metrics, conducted hyperparameter ablations for Algorithm 3, analyzed the impact of distributional shift when using different data for belonging images or for fine-tuning the inverse decoder, added an additional baseline and examined adversarial robustness.
>
> We addressed all eight concerns and ten questions below and made corresponding modifications to the updated submission as well.
>
> >**W1.1 While the domain-specific adaptation is useful, the conceptual advance is incremental. The optimized quantization algorithm (Algorithm 3) for multi-scale IARs is the most novel technical component.**
>
> We thank the Reviewer for appreciating our optimized quantization algorithm. We would like to highlight **three novel technical contributions specific to IARs in our submission**:
>
> **1. Discovery and exploitation of quantization artifacts:** While reconstruction-based approaches exist for GANs and diffusion models, our work is the first to propose a **quantization-based** approach for data provenance. The quantization-based error is a **unique artifact in image autoregressive models (IARs)**, and our experiments prove it to be **significantly more effective** than the reconstruction-based methods (Table 1). In addition, since our proposed quantization loss detects artifacts in the feature space instead of pixel space, our method is also much more robust against the post-processing in the pixel space than the reconstruction-based methods. We demonstrate that our proposed latent-space signal stays consistent after transformations, while the pixel-space signals are completely destroyed (Section 4.3 and Table 2).
>
> **2. Decoder inversion methodology:** Unlike RONAN or LatentTracer which requires optimization per-image, we propose a **one-time fine-tuning approach** that creates an inverse decoder specifically optimized to detect generated images. Table 4 demonstrates that this is critical: the original encoder fails (6.4% TPR@1%FPR for RAR) while our inverted decoder succeeds (98.9% TPR@1%FPR).
>
> **3. Optimized quantization for multi-scale IARs** (Algorithm 3): This addresses a **fundamental challenge in scale-wise models**, where naive quantization completely fails (0.4%TPR@1%FPR, Table 3) due to scale interactions. Our algorithm achieves 93.3%TPR@1%FPR: this solves a novel technical problem for the state-of-the-art next-scale prediction paradigm in IARs. The Reviewer also acknowledges that Algorithm 3 is a "novel technical component".
>
> >**W1.2 The optimized quantization algorithm (Algorithm 3) receives insufficient analysis**
>
> We thank the Reviewer for their interest in our optimized quantization algorithm. We described the details of Algorithm 3 and compared it with the original quantization in VAR (Algorithm 2) in the Appendix A. We further provide a more detailed definition of the optimization problem and give further explanation in the Section 3.3.1 in the revised submission.
>
> We extended the analysis, by adding a hyperparameter analysis (answer to Q4 and Table A11), comparison to Algorithm 2 (answer to W2.3), as well as potential engineering possibilities to speed up performance (answer to W2.3). We have also include these tables and analysis in the appendix of the revised manuscript.
>
>
> >**Q4 For the multi-scale VAR model, how sensitive is Algorithm 3 to hyperparameter choices (number of iterations $N_{iters}$, learning rate, initialization)?**
>
> We also provide more detailed experiments to analyze the hyperparameter choices of the optimized quantization (Algorithm 3) as follows. We evaluate on the VAR model, VAR-generated images as the belonging images, and ImageNet as the non-belonging images. The setting we used in the submission is 1200 iterations, learning rate of 0.1, and using greedy search as initialization.
>
> | Configuration | Value | TPR@1%FPR for QuantLoss (%) |
> |---|---|---|
> | Baseline (Original Quantization in VAR) | - | 0.4 |
> | Number of Iterations | 100 | 87.5 (0.57 seconds) |
> | | 400 | 91.0 (2.34 seconds) |
> | | 1000 | 95.4 |
> | | 1200 | 95.0 (8.24 seconds) |
> | | 1400 | 93.8 |
> | | 1600 | 92.2 |
> | Learning Rate | 0.01 | 43.0 |
> | | 0.05 | 95.2 |
> | | 0.1 | 95.0 |
> | | 0.2 | 94.2 |
> | | 0.5 | 92.8 |
> | Initialization with Original Quant. | No | 94.3 |
> | | Yes | 95.0 |
>
> From the above table, we observe that the optimal performance (95.0-95.4% TPR@1%FPR) occurs with 1,000-1,400 iterations and learning rate of 0.1. Notably, our method still achieves 87.5%TPR @1% FPR with only 100 iterations, which can reduce the runtime from 8.24s/image to 0.57s/image. In addition, initializing with VAR's original quantization provides a modest boost (95.0% vs. 94.3%).

---

> ### Author Response · Authors · 2025-11-23
> **Responses to Reviewer 891z (2/13)**
>
> >**W2.1 Requires white-box access to the target model, which may not be available for commercial or closed-source IARs.**
>
> We appreciate this important question. We would like to point out that our framework addresses two critical real-world scenarios where model owners of closed-source IARs would have strong incentives to enable data provenance detection:
>
> *1. Prevent model collapse.*
> The success of image autoregressive models (IARs) relies on highly data-intensive training processes. Training data are typically scraped from web-scale internet collections. However, the growing amount of generated content online has led to an increasing proportion of generated rather than natural data in training corpora.
>
> Iterative training on model-generated data has been demonstrated to cause performance degradation [8][9], a phenomenon referred to as **model collapse**. This presents a critical challenge for model owners (especially major organizations such as OpenAI, Google, and ByteDance) who develop large-scale proprietary models deployed via public APIs. To mitigate potential quality degradation in subsequent model iterations, these organizations require effective methods to identify content generated by their own model families.
>
> Our method provides model owners with an automated and effective tool (which does not change the generation process) to identify and filter out synthetic images previously generated by their model family when curating training data for future model versions, thereby addressing this fundamental challenge in sustainable model development.
>
> *2. Responsible AI release.*
> Regulatory frameworks increasingly mandate transparency for synthetic content. For example, the EU AI Act Article 50 Recital 134 states: *”Further to the technical solutions employed by the providers of the AI system, deployers who use an AI system to generate or manipulate image, audio or video content that appreciably resembles existing persons, objects, places, entities or events and would falsely appear to a person to be authentic or truthful (deep fakes), should also clearly and distinguishably disclose that the content has been artificially created or manipulated by labelling the AI output accordingly and disclosing its artificial origin.”*
>
>
> Therefore, we expect that the model owners would be willing to provide an API to check if an image is generated even for a closed-source model. Many services already provide such methods, for example, SORA adds both visible and invisible watermarks on its generated videos as part of its responsible AI development goals [10]. Additionally, we only require white-box access to the autoencoder but black-box access to the transformer.
>
> Moreover, for users to identify generated images, the model owner does not need to provide full access to their model, but only needs to give users access to the inverse decoder and the codebook entries. This can also be done with a black-box API to answer if a given image was generated by a given model or not (or provide confidence scores).
>
>
> >**W2.2 Requires per-model finetuning (10-50 epochs, 10K-50K images) despite being called "post-hoc". This is computationally expensive and contradicts the claim of broad applicability to "already published content" without additional training data.**
>
> We thank the Reviewer for the question and would like to clarify the term “post-hoc”. Our framework is applied **after** the images were generated, making it a *post-hoc* provenance framework. Detection methods, such as watermarking [1, 2, 3], have to change the generation process of the model to embed a signal into the generated image. This impacts the generative process and usually leads to a degradation in image quality. Most importantly, these watermarking methods cannot identify already published content (which was not watermarked).
>
> While our framework requires fine-tuning for higher performance on the provenance task, it does not impact the generative process at all. Additionally, it can be used to trace the provenance of any already published content, making it more freely and easily applicable than watermarking. The fine-tuning itself only generates minor overhead for model-owners in comparison to the training of the IAR model, as we also show in response to Q9.

---

> ### Author Response · Authors · 2025-11-23
> **Responses to Reviewer 891z (3/13)**
>
> >**W2.3 High computational cost for some models (VAR: 8.249 sec/image, Table A3) limits scalability.**
>
> We would like to thank the Reviewer for the question. We note that the high computational cost for VAR is attributed to the token optimization that we propose in Algorithm 3. We investigate two options for accelerating the algorithm:
>
> 1. **Faster implementation:**
>
> Our algorithm benefits from using quicker engineering implementations, such as:
> Using the Einstein summation convention for calculating the codebook distance.
> Using torch.compile to optimize the calculation of the feature map.
>
> These two techniques reduced the runtime of our method from 8.24s/image to 7.79s/image. The algorithm may be further accelerated with new developments in the deep learning toolkit.
>
>
> 2. **Fewer iterations:**
>
> We show that our method still maintains high detection performance and can be accelerated a lot with fewer iterations. For example, our method still achieves 87.5%TPR @1% FPR with only 100 iterations, reducing the runtime from 8.24s/image to 0.57s/image.
>
> | Method | Iterations | TPR@1%FPR (%) | Time (seconds/image) |
> |:---|:---|:---:|:---:|
> | Default| 1200 | 95.0 | 8.24 |
> | Less Iterations | 100 | 87.5 | 0.57 |
> | Accelerated with Torch| 1200 | 94.8 | 7.79 |
>
> Moreover, in Table A3 the time cost is computed with a batch size of 1. In practice, the required time can be amortized when Algorithm 3 is executed on a mini-batch of images. For example, the per-image time cost can be reduced to 1.60s/image for 1200 iterations and 0.18s/image for 100 iterations when using a batch size of 8.
>
>
> >**W3.1&Q2 Why does finetuning the encoder to invert the decoder improve attribution? The paper provides intuition (Eq. 6) but no theoretical analysis of what properties $D^{-1}$ should satisfy.**
>
> We expand on the theoretical justification regarding the intuition provided by (Eq.6):
>
> The original encoder E is trained on natural images to minimize reconstruction loss on real data. For generated images $x_Z = D(f_Z)$, we need to recover $f_Z$ from $x_Z$. Since D is trained to map quantized features $f_Z \rightarrow x_Z$, the ideal inverse should map $ x_Z \rightarrow f_Z$. By fine-tuning on generated images with frozen $D$ and codebook $Z$, we optimize $D^{-1}$ to satisfy:
> $f_Z ≈ D^{-1}(D(f_Z)$)
> This ensures that for belonging images, $D^{-1}(x_Z)$ lands close to codebook entries, yielding low QuantLoss. For non-belonging images, no such guarantee exists.
>
> >**W3.2 Why are QuantLoss and EncLoss "orthogonal"? Only empirical observations are provided.**
>
> We would like to rephrase it as QuantLoss and EncLoss are “complementary” (also updated in Line 198). Specifically, **QuantLoss** measures proximity to discrete codebook entries (token-space method)
> **EncLoss** measures information preservation through encoding-decoding (latent+pixel-space method)
>
> These two methods capture different features in the token- and latent+pixel- spaces of the generation process. Table 3 shows combining them improves performance, validating their complementarity.

---

> ### Author Response · Authors · 2025-11-23
> **Responses to Reviewer 891z (4/13)**
>
> >**W3.3&Q5&Q8 The choice of multiplicative combination in Eq. 10 lacks justification. Why not additive or learned weighting?**
>
> Since our EncLoss $L^{Cal}\_{Enc}$ is a ratio (Eq. 9), multiplicative combination (Eq. 10) treats it as a scaling factor. We provide an ablation study comparing additive versus multiplicative combinations, as well as the use of learned weights, both in Table A14 and in this section below. For both addition and multiplication we simply combine the two losses with the respective arithmetic operation. In the weighted scenarios we determine optimal weights for EncLoss by keeping the weight for the QuantLoss fixed. Specifically, we determine the optimal weight  $w_{Enc}$  for EncLoss via grid search by leveraging ImageNet as a calibration set. For the weighted addition we iterate through 1,000 evenly spaced values between 0.001 and 1, and another 1,000 values between 1 and 1,000. For the weighted multiplicative case, the weight is used as an exponent, and we apply a grid search over 1,000 values between 0.01 and 10. The learned weights for multiplication power enables our method to achieve the best performance in most cases, even with a good performance with the more challenging case of Infinity vs LAION.
>
>
> | Model | Method | ImageNet | LAION | MS-COCO | LlamaGen | RAR  | Taming | VAR   | Infinity | VQDiff | $w_{Enc}$ |
> |:---------------:|:---------------------:|:--------:|:-----:|:-------:|:--------:|:-----:|:------:|:-----:|:--------:|:------:|:---------:|
> | **LlamaGen**    | Addition              | 100.0    | 100.0 | 100.0   | -        | 100.0 | 100.0  | 100.0 | 100.0    | 100.0  | -         |
> |                 | Addition Weighted     | 100.0    | 100.0 | 100.0   | -        | 100.0 | 100.0  | 100.0 | 100.0    | 100.0  | 0.00      |
> |                 | Multiplication        | 100.0    | 100.0 | 100.0   | -        | 100.0 | 100.0  | 100.0 | 100.0    | 100.0  | -         |
> |                 | Multiplication Power  | 100.0    | 100.0 | 100.0   | -        | 100.0 | 100.0  | 100.0 | 100.0    | 100.0  | 0.01      |
> | **RAR**         | Addition              | 99.3     | 99.2  | 99.5    | 98.5     | -     | 97.7   | 98.9  | 99.6     | 99.6   | -         |
> |                 | Addition Weighted     | 100.0    | 99.8  | 100.0   | 99.8     | -     | 99.3   | 99.8  | 100.0    | 100.0  | 0.00      |
> |                 | Multiplication        | 100.0    | 100.0 | 100.0   | 99.9     | -     | 99.9   | 100.0 | 100.0    | 100.0  | -         |
> |                 | Multiplication Power  | 100.0    | 99.8  | 100.0   | 99.8     | -     | 99.3   | 99.9  | 100.0    | 100.0  | 0.01      |
> | **Taming**      | Addition              | 100.0    | 100.0 | 100.0   | 100.0    | 100.0 | -      | 100.0 | 100.0    | 100.0  | -         |
> |                 | Addition Weighted     | 100.0    | 99.8  | 100.0   | 99.9     | 100.0 | -      | 100.0 | 100.0    | 100.0  | 0.00      |
> |                 | Multiplication        | 100.0    | 100.0 | 100.0   | 100.0    | 100.0 | -      | 100.0 | 100.0    | 100.0  | -         |
> |                 | Multiplication Power  | 100.0    | 99.6  | 100.0   | 99.9     | 100.0 | -      | 100.0 | 100.0    | 100.0  | 0.27      |
> | **VAR**         | Addition              | 100.0    | 98.2  | 100.0   | 98.2     | 100.0 | 99.8   | -     | 100.0    | 100.0  | -         |
> |                 | Addition Weighted     | 100.0    | 99.3  | 100.0   | 98.9     | 100.0 | 99.5   | -     | 100.0    | 100.0  | 0.02      |
> |                 | Multiplication        | 100.0    | 99.5  | 100.0   | 99.2     | 100.0 | 100.0  | -     | 100.0    | 100.0  | -         |
> |                 | Multiplication Power  | 100.0    | 99.3  | 100.0   | 99.2     | 100.0 | 99.5   | -     | 100.0    | 100.0  | 0.30      |
> | **Infinity**    | Addition              | 0.0      | 97.3  | 99.0    | 1.4      | 0.6   | 0.6    | 18.0  | -        | 36.1   | -         |
> |                 | Addition Weighted     | 98.8     | 98.9  | 99.2    | 98.8     | 99.1  | 98.8   | 99.1  | -        | 99.1   | 0.00      |
> |                 | Multiplication        | 0.0      | 98.3  | 99.2    | 10.1     | 4.1   | 4.2    | 58.8  | -        | 77.4   | -         |
> |                 | Multiplication Power  | 99.3     | 97.8  | 99.4    | 99.1     | 99.4  | 99.1   | 99.3  | -        | 99.2   | 0.01      |
> | **VQ-Diffusion**| Addition | 100.0  | 100.0 | 100.0 | 98.0 | 100.0 | 100.0  | 100.0 | 100.0    | - | - |
> | | Addition Weighted | 100.0    | 97.4  | 100.0   | 99.8     | 100.0 | 99.4   | 100.0 | 100.0    | -      | 0.00      |
> |                 | Multiplication        | 100.0    | 99.5  | 100.0   | 99.9     | 100.0 | 99.9   | 100.0 | 100.0    | -      | -         |
> |                 | Multiplication Power  | 100.0    | 95.0  | 100.0   | 99.5     | 100.0 | 99.3   | 100.0 | 100.0    | -      | 0.48      |

---

> ### Author Response · Authors · 2025-11-23
> **Responses to Reviewer 891z (5/13)**
>
> >**W3.4 No analysis of failure modes or theoretical performance guarantees**
>
> >**W4.4 No statistical significance testing (error bars, confidence intervals) despite stochastic generation processes.**
>
> We test if a data point x significantly deviates from a given belonging distribution. Similar to RONAN [4] we leverage Grubbs’ hypothesis test [5].
> For this, we formulate the following hypothesis
>
> $\mathcal{H}_0:\textit{The test sample does not belong to the given model.}$
>
> In Grubbs's hypothesis test, we reject $\mathcal{H}_0$ if the following inequality holds:
>
> $\frac{x-\mu}{\sigma}<\frac{N-1}{\sqrt{N}}\sqrt{\frac{(t_{\alpha/N,N-2})^2}{N-2+(t_{\alpha/N,N-2})^2}}$
>
> Where $\mu$ and $\sigma$ are the mean and the standard deviation of a given belonging dataset, $x$ is the queried data sample and $N$ the number of samples in the dataset.
>
> We report below and in Table A19 the results of applying Grubbs’ hypothesis test on 1,000 belonging and 1,000 non-belonging samples across all datasets for each model.
>
> | Model        | TP  | FP | TN  | FN | TPR (%) | FPR (%) |
> |--------------|-----|----|-----|----|---------|---------|
> | LlamaGen     | 995 | 0  | 1000| 5  | 99.5    | 0.0     |
> | RAR          | 999 | 0  | 1000| 1  | 99.9    | 0.0     |
> | Taming       |1000 | 1  | 999 | 0  | 100.0   | 0.1     |
> | VAR          |1000 | 0  |1000 | 0  | 100.0   | 0.0     |
> | Infinity     | 993 | 0  |1000 | 7  | 99.3    | 0.0     |
> | VQ-Diffusion | 999 | 0  |1000 | 1  | 99.9    | 0.0     |
>
> We find that we achieve a TPR over 99% for all models and 0% FPR for most models, demonstrating the effectiveness of our attribution.
>
>
> >**W4.1&Q6 The setting is somewhat artificial: 1000 clean images at 256×256 resolution, binary classification against single sources. Real-world scenarios involve mixed resolutions, multiple processing steps, and multi-class attribution.**
>
>
> *Mixed resolutions:* In our evaluation in Table 1 of the original manuscript, we consider images from ImageNet, MS-COCO and LAION as clean natural data. These datasets consist of many differing resolutions, yet our framework allows for near perfect provenance across all IARs.
>
> *Multiple processing steps:* Additionally to our robustness evaluation in Table 2, we evaluate our method against multiple post-processing steps. We use the following consistent settings for each attacks:
>
> *Attack strengths*: JPEG: 75%, Brightness: 1.2, Saturation: 1.2, Contrast: 1.2, Resize: Resizing to 80% of the original size, Kernel: Gaussian Blur with kernel size 3, Noise: Gaussian Noise with std of 0.025
>
> | Method | Attacks |  |  |  |  |
> |---|---|---|---|---|---|
> |  | Brightness+Contrast+Saturation | Brightness+JPEG | Contrast+Resize+JPEG | JPEG+Resize+Kernel | Noise+JPEG+Saturation+Resize |
> | LatentTracer | 4.0 | 6.0 | 3.0 | 3.2 | 2.9 |
> | Reconstruction | 1.6 | 3.1 | 1.8 | 1.8 | 1.2 |
> | AEDR | 3.5 | 4.3 | 5.6 | 3.6 | 9.2 |
> | Ours (w/o Aug) | 91.2 | 89.7 | 79.1 | 86.2 | 79.4 |
> | Ours (w/ Aug) | 98.3 | 95.3 | 86.8 | 93.1 | 89.5 |
>
> Our results highlight that our QuantLoss enables reliable provenance even after multiple post-processing steps, where other baselines break. While none of the baseline methods achieve at least 10%TPR@1%FPR, our framework allows detection with >86%TPR@1%FPR.
>
>
>
> *Multi-class attribution:* In the setting of data provenance against model collapse, model providers are concerned with reliably identifying their generated data. To fully evaluate this setting we perform a 1 vs. All comparison, detecting the data generated by a model against all possible data in a single dataset. The results, added to Table A18 and displayed here, show that a model provider can almost perfectly differentiate between their own data and data in the wild.
>
> | Method | LlamaGen | RAR | Taming | VAR | Infinity | VQ-Diffusion |
> |---|---|---|---|---|---|---|
> | Reconstruction | 28.0 | 2.3 | 24.7 | 0.4 | 0.0 | 10.5 |
> | LatentTracer | 92.8 | 1.0 | 69.8 | 1.6 | 0.2 | 97.3 |
> | AEDR | 59.4 | 17.2 | 80.0 | 26.9 | 3.5 | 84.2 |
> | Ours | **100.0** | **100.0** | **100.0** | **100.0** | **99.2** | **100.0** |

---

> ### Author Response · Authors · 2025-11-23
> **Responses to Reviewer 891z (6/13)**
>
> >**W4.2&Q1&Q7: Finetuning uses generated images from the same distribution as test images, which may lead to overfitting. Cross-distribution generalization (e.g., different prompts, generation hyperparameters) is not evaluated.**
>
> We ablate the cross-distribution generalization of our method, by the changing generation hyperparameters. Specifically we evaluated the RAR model, where the inverse decoder was fine-tuned using data generated with CFG=4 and temperature=1.0. We detected RAR generated data produced with different hyperparameters vs. natural data or data generated by other models. We report the TPR@1%FPR for our Combined Loss.
>
> The results, displayed below and added to Table A11, show that using different hyperparameters has limited impact on the performance of our method. We attribute this to the fact that the model generates the same tokens regardless of the generation hyperparameters and our method leverages the fact that the generated images consist of these tokens, independently of their generation order.
>
> | CFG | Temperature | Natural |  |  | Generated |  |  |  |  |
> |:---:|:---:|:---:|:---:|:---:|:---:|:---:|:---:|:---:|:---:|
> |  |  | ImageNet | LAION | MS-COCO | LlamaGen | Taming | VAR | Infinity | VQDiff |
> |3  | 0.8 | 99.7 | 99.8 | 99.7 | 99.6 | 99.4 | 100.0 | 99.8 | 100.0 |
> | 3 | 1.0 | 100.0 | 100.0 | 100.0 | 99.5 | 99.4 | 100.0 | 100.0 | 100.0 |
> | 3 | 1.2 | 99.9 | 99.9 | 100.0 | 99.8 | 99.8 | 99.9 | 100.0 | 100.0 |
> | 4 | 0.8 | 99.5 | 99.8 | 99.8 | 99.3 | 99.4 | 99.5 | 99.8 | 99.9 |
> | 4 | 1.0 | 100.0 | 100.0 | 100.0 | 99.9 | 99.9 | 100.0 | 100.0 | 100.0 |
> |  4| 1.2 | 100.0 | 99.6 | 100.0 | 99.5 | 98.8 | 100.0 | 100.0 | 100.0 |
> | 5 | 0.8 | 100.0 | 99.6 | 100.0 | 99.5 | 98.8 | 100.0 | 100.0 | 100.0 |
> | 5 | 1.0 | 99.9 | 99.0 | 100.0 | 98.4 | 98.9 | 100.0 | 100.0 | 100.0 |
> | 5 | 1.2 | 99.7 | 99.4 | 100.0 | 99.4 | 99.3 | 99.5 | 100.0 | 99.8 |
>
> Additionally we performed an experiment, where we separated the data used to fine-tune the inverse decoder between classes. Specifically we used the first 500 classes to fine-tune the inverse decoder and used the remaining 500 classes for evaluation, ensuring that the model can not overfit on the distribution. We denote this as Ours (class split) and our standard fine-tuning as Ours (random split). We report the TPR@1%FPR of belonging vs non-belonging data. We observe that our method maintains high performance consistently even in this setting.
>
> *P.S. Only using 500 classes to generate the fine-tuning set is unlikely to happen in the real world. As a model owner or given an open-source model, we can generate images covering all classes to fine-tune the inverse decoder. This is only an experiment performed under a more strict setting to demonstrate the generalization of our approach.*
>
> | Model | Method | Natural |  |  | Generated |  |  |  |  |  |
> |:---:|:---:|:---:|:---:|:---:|:---:|:---:|:---:|:---:|:---:|:---:|
> |  |  | ImageNet | LAION | MS-COCO | LlamaGen | RAR | Taming | VAR | Infinity | VQDiff |
> | **RAR** | Reconstruction | 3.8 | 4.1 | 7.4 | 0.8 | - | 0.1 | 5.7 | 18.1 | 18.8 |
> |  | LatentTracer | 6.0 | 6.1 | 15.2 | 0.4 | - | 0.0 | 9.3 | 24.6 | 26.9 |
> | | AEDR | 29.5 | 16.6 | 36.6 | 10.6 | - | 2.3 | 35.9 | 49.9 | 27.6 |
> |  | Ours (random split) | 100.0 | 100.0 | 100.0 | 99.9 | - | 99.9 | 100.0 | 100.0 | 100.0 |
> |  | Ours (class split) | 99.8 | 99.9 | 100.0 | 99.4 | - | 99.7 | 99.7 | 100.0 | 100.0 |
> | **Taming**| Reconstruction | 27.5 | 21.5 | 27.6 | 10.1 | 18.9 | - | 27.7 | 39.0 | 46.1 |
> |  | LatentTracer | 73.0 | 61.0 | 75.9 | 36.4 | 66.8 | - | 76.0 | 85.4 | 87.4 |
> |  | AEDR | 80.4 | 82.5 | 81.9 | 70.7 | 80.7 | - | 78.1 | 91.9 | 87.5 |
> |  | Ours (random split) | 100.0 | 100.0 | 100.0 | 100.0 | 100.0 | - | 100.0 | 100.0 | 100.0 |
> |  | Ours (class split) | 100.0 | 100.0 | 100.0 | 100.0 | 100.0 | - | 100.0 | 100.0 | 100.0 |
> |**VAR** | Reconstruction | 1.4 | 1.4 | 3.6 | 0.1 | 1.6 | 0.0 | - | 5.9 | 5.9 |
> |  | LatentTracer | 3.9 | 1.3 | 12.0 | 0.2 | 5.6 | 0.1 | - | 15.4 | 15.3 |
> | | AEDR | 29.1 | 15.7 | 50.6 | 14.7 | 28.3 | 14.0 | - | 37.5 | 50.8 |
> |  | Ours (random split) | 100.0 | 99.2 | 100.0 | 99.2 | 100.0 | 100.0 | - | 100.0 | 100.0 |
> |  | Ours (class split) | 99.9 | 98.9 | 100.0 | 99.4 | 100.0 | 99.1 | - | 100.0 | 99.9 |
>
> In addition, we would like to clarify that all the experiments about text-to-image models (Infinity and VQDiffusion) use different prompts for generating the fine-tuning and evaluation sets.
>
> Our results here and added to Table A12 show that the different fine-tuning and evaluation splits have near to zero impact on the performance of our method. We further attribute this to our method enabling the inverse decoder to recover the tokens used for generation, rather than focusing on the images themselves.

---

> ### Author Response · Authors · 2025-11-23
> **Responses to Reviewer 891z (7/13)**
>
> >**W4.3 ROC curves and AUC would provide better insight.**
>
> To extend our performance metric, we calculated TPR\@0.5%FPR (Table A8),  TPR\@0.1%FPR (Table A9), AUC (Table A10) and plotted the ROC curves (Figure A5). For the Reviewer's convenience, we also display the results for the TPR\@0.5%FPR and the AUC below.
>
> **Table. TPR\@0.5%FPR of our method and the baselines**
>
> | **Model** | **Method** | **Natural** | | | | **Generated** | | | | |
> |---|---|---|---|---|---|---|---|---|---|---|
> | | | ImageNet | LAION | MS-COCO | LlamaGen | RAR | Taming | VAR | Infinity | VQDiff |
> | **LlamaGen** | Reconstruction | 23.4 | 25.0 | 30.6 | - | 30.4 | 2.1 | 31.4 | 61.8 | 60.4 |
> | | LatentTracer | 89.7 | 82.7 | 93.6 | - | 94.6 | 72.5 | 95.1 | 98.8 | 98.0 |
> | | AEDR | 41.1 | 49.9 | 38.1 | - | 55.2 | 50.0 | 57.4 | 66.4 | 59.8 |
> | | **Ours** | **100.0** | **100.0** | **100.0** | - | **100.0** | **100.0** | **100.0** | **100.0** | **100.0** |
> | **RAR** | Reconstruction | 2.7 | 3.1 | 5.8 | 0.2 | - | 0.0 | 2.5 | 10.4 | 14.6 |
> | | LatentTracer | 2.2 | 1.0 | 2.2 | 0.2 | - | 0.0 | 2.1 | 7.7 | 5.3 |
> | | AEDR | 13.6 | 15.1 | 30.0 | 4.0 | - | 0.8 | 22.5 | 42.4 | 17.2 |
> | | **Ours** | **99.9** | **100.0** | **100.0** | **99.9** | - | **98.9** | **99.9** | **100.0** | **100.0** |
> | **Taming** | Reconstruction | 17.3 | 18.7 | 22.2 | 6.5 | 18.4 | - | 25.2 | 39.5 | 45.0 |
> | | LatentTracer | 64.8 | 52.2 | 70.6 | 32.2 | 69.8 | - | 72.6 | 82.9 | 100.0 |
> | | AEDR | 75.8 | 61.7 | 88.7 | 51.8 | 79.8 | - | 73.3 | 88.6 | 80.0 |
> | | **Ours** | **100.0** | **100.0** | **100.0** | **100.0** | **100.0** | - | **100.0** | **100.0** | **100.0** |
> | **VAR** | Reconstruction | 0.5 | 0.5 | 2.1 | 0.1 | 1.4 | 0.0 | - | 5.4 | 4.0 |
> | | LatentTracer | 2.8 | 0.1 | 8.4 | 0.1 | 4.2 | 0.0 | - | 12.8 | 10.1 |
> | | AEDR | 9.8 | 10.9 | 45.3 | 4.1 | 22.6 | 3.9 | - | 33.1 | 40.4 |
> | | **Ours** | **100.0** | **97.1** | **100.0** | **96.8** | **100.0** | **99.6** | - | **100.0** | **100.0** |
> | **Infinity** | Reconstruction | 0.0 | 0.2 | 0.3 | 0.0 | 0.0 | 0.0 | 0.2 | - | 0.2 |
> | | LatentTracer | 0.0 | 6.5 | 25.8 | 0.0 | 0.0 | 0.0 | 1.6 | - | 3.6 |
> | | AEDR | 1.1 | 7.1 | 36.3 | 0.6 | 2.7 | 0.3 | 8.2 | - | 6.1 |
> | | **Ours** | **99.2** | **15.5** | **99.4** | **99.1** | **99.5** | **99.1** | **99.4** | - | **99.2** |
> | **VQ-Diffusion** | Reconstruction | 4.5 | 6.0 | 12.9 | 4.5 | 16.6 | 0.6 | 16.7 | 33.6 | - |
> | | LatentTracer | 95.0 | 90.9 | 97.4 | 96.1 | 97.7 | 88.9 | 98.2 | 98.4 | - |
> | | AEDR | 82.7 | 43.1 | 87.3 | 71.0 | 91.7 | 60.0 | 80.2 | 79.8 | - |
> | | **Ours** | **100.0** | **98.7** | **100.0** | **93.3** | **100.0** | **99.6** | **100.0** | **100.0** | - |
>
> **Table. AUC of our method and the baselines**
>
> | **Model** | **Method** | **Natural** |  |  | **Generated** |  |  |  |  |  |
> |:---:|:---:|:---:|:---:|:---:|:---:|:---:|:---:|:---:|:---:|:---:|
> |  |  | ImageNet | LAION | MS-COCO | LlamaGen | RAR | Taming | VAR | Infinity | VQDiff |
> | **LlamaGen** | Reconstruction | 93.9 | 93.1 | 96.5 | - | 92.6 | 81.1 | 94.6 | 97.9 | 97.1 |
> | | LatentTracer | 99.7 | 99.6 | 99.9 | - | 99.8 | 98.8 | 99.8 | 99.9 | 99.9 |
> |  | AEDR | 94.7 | 94.1 | 94.7 | - | 95.7 | 95.2 | 95.2 | 96.1 | 96.0 |
> |  | Ours | 100.0 | 100.0 | 100.0 | - | 100.0 | 100.0 | 100.0 | 100.0 | 100.0 |
> | **RAR** | Reconstruction | 76.5 | 74.5 | 82.3 | 66.6 | - | 49.7 | 71.4 | 87.2 | 86.5 |
> | | LatentTracer | 73.2 | 70.6 | 78.0 | 57.0 | - | 38.6 | 67.2 | 85.3 | 83.7 |
> |  | AEDR | 90.2 | 86.2 | 89.3 | 87.6 | - | 82.9 | 89.2 | 93.4 | 92.0 |
> |  | Ours | 100.0 | 100.0 | 100.0 | 100.0 | - | 99.8 | 100.0 | 100.0 | 100.0 |
> | **Taming** | Reconstruction | 86.8 | 82.9 | 88.3 | 80.5 | 84.6 | - | 87.1 | 89.9 | 92.3 |
> |  | LatentTracer | 98.2 | 97.0 | 98.8 | 95.6 | 98.1 | - | 98.6 | 99.1 | 100.0 |
> |  | AEDR | 98.7 | 98.5 | 99.1 | 97.2 | 99.0 | - | 98.8 | 99.5 | 99.1 |
> |  | Ours | 100.0 | 100.0 | 100.0 | 100.0 | 100.0 | - | 100.0 | 100.0 | 100.0 |
> | **VAR** | Reconstruction | 64.1 | 58.1 | 69.0 | 50.3 | 58.9 | 40.4 | - | 69.5 | 70.4 |
> | | LatentTracer | 80.6 | 72.6 | 84.9 | 68.3 | 77.5 | 61.5 | - | 82.2 | 82.9 |
> |  | AEDR | 95.8 | 92.9 | 96.8 | 92.3 | 94.7 | 92.8 | - | 96.5 | 96.7 |
> |  | Ours | 100.0 | 99.9 | 100.0 | 100.0 | 100.0 | 100.0 | - | 100.0 | 100.0 |
> | **Infinity** | Reconstruction | 30.4 | 61.4 | 64.9 | 20.2 | 19.1 | 23.0 | 24.8 | - | 27.1 |
> |  | LatentTracer | 62.7 | 91.6 | 94.6 | 53.0 | 50.6 | 54.2 | 62.5 | - | 61.6 |
> |  | AEDR | 86.2 | 97.2 | 98.9 | 81.5 | 82.7 | 85.5 | 91.4 | - | 86.7 |
> |  | Ours | 99.8 | 98.8 | 99.7 | 99.6 | 100.0 | 99.7 | 99.9 | - | 99.7 |
> | **VQ-Diffusion** | Reconstruction | 90.7 | 84.3 | 91.9 | 88.9 | 88.5 | 79.2 | 91.4 | 92.4 | - |
> |  | LatentTracer | 99.9 | 99.8 | 99.9 | 99.9 | 99.9 | 99.6 | 100.0 | 100.0 | - |
> |  | AEDR | 99.5 | 98.5 | 99.6 | 99.2 | 99.8 | 99.2 | 99.4 | 99.5 | - |
> |  | Ours | 100.0 | 99.8 | 100.0 | 99.9 | 100.0 | 100.0 | 100.0 | 100.0 | - |
>
> The results show that our method strictly outperforms all of the baselines for all models and non-belonging datasets.

---

> ### Author Response · Authors · 2025-11-23
> **Responses to Reviewer 891z (8/13)**
>
> >**W5 As acknowledged in Appendix H, augmentation training improves QuantLoss but degrades EncLoss performance. This suggests the approach lacks principled robustness and relies on dataset-specific overfitting.**
>
> We appreciate Reviewer’s careful reading of our paper, including to Appendix H. Both methods work better than the baselines in the clean image setting. We stress tested both methods and, indeed, observed that in the case of additional image post-processing (noisy setting), the augmentations might degrade the performance of EncLoss. However, even in this hardest setup, our QuantLoss is still much more robust and performant than the baselines. Overall, in any setting, our QuantLoss significantly outperforms baselines.
> Furthermore, we performed experiments to confirm that our methods are not dataset-specific, as we show in our answer to W4.2, Q1 and Q7. Instead, our methods generalize very well across different datasets, outperforming the baselines by a large margin. Meanwhile, we show an extended table of robustness evaluation across more datasets. We show that our method outperforms the baselines by a very large margin after image post-processing, validating the universal robustness of our approach.
>
> | Transform | Method| **Natural** | | | **Generated** | | | | |
> |-----------|--------|----------|-------|---------|----------|--------|-----|----------|--------|
> | | | ImageNet | LAION | MS-COCO | LlamaGen | Taming | VAR | Infinity | VQDiff |
> | Noise=0.05 | Reconstruction | 2.3 | 0.8 | 2.1 | 0.0 | 0.0 | 1.3 | 13.3 | 6.6 |
> | | LatentTracer | 3.4 | 0.7 | 3.8 | 0.0 | 0.1 | 1.4 | 10.7 | 7.2 |
> | | AEDR | 7.3 | 4.7 | 6.2 | 4.4 | 0.5 | 4.6 | 13.7 | 5.5 |
> | | **Ours** | **87.8** | **82.3** | **94.6** | **75.2** | **65.4** | **90.3** | **95.9** | **93.1** |
> | Kernel=9 | Reconstruction | 3.0 | 2.1 | 3.3 | 0.4 | 0.1 | 2.8 | 9.5 | 5.9 |
> | | LatentTracer | 4.7 | 2.1 | 3.4 | 0.4 | 0.1 | 3.1 | 12.4 | 7.5 |
> | | AEDR | 11.4 | 5.0 | 13.8 | 2.5 | 0.5 | 9.9 | 18.9 | 12.0 |
> | | **Ours** | **80.5** | **74.1** | **82.3** | **69.7** | **63.9** | **78.3** | **83.4** | **82.6** |
> | JPEG=60 | Reconstruction | 3.6 | 2.3 | 4.8 | 0.5 | 0.0 | 2.1 | 11.4 | 12.1 |
> | | LatentTracer | 4.8 | 3.5 | 6.9 | 0.1 | 0.0 | 2.8 | 15.9 | 15.4 |
> | | AEDR | 8.9 | 5.4 | 18.5 | 2.4 | 0.3 | 6.8 | 29.1 | 11.9 |
> | | **Ours** | **96.1** | **94.1** | **98.8** | **90.3** | **83.3** | **98.3** | **98.9** | **98.5** |
> | Brightness=1.6 | Reconstruction | 1.4 | 0.5 | 2.3 | 0.1 | 0.0 | 1.2 | 4.6 | 3.2 |
> | | LatentTracer | 2.3 | 1.0 | 2.7 | 0.0 | 0.0 | 1.7 | 5.8 | 3.7 |
> | | AEDR | 1.9 | 0.5 | 2.0 | 0.5 | 0.4 | 1.1 | 3.1 | 2.0 |
> | | **Ours** | **92.3** | **75.6** | **95.1** | **78.6** | **60.4** | **94.0** | **97.3** | **96.1** |
> | Contrast=2.0 | Reconstruction | 1.6 | 2.2 | 2.8 | 0.0 | 0.0 | 1.6 | 5.5 | 7.7 |
> | | LatentTracer | 3.0 | 1.8 | 6.3 | 0.1 | 0.0 | 2.2 | 7.7 | 9.4 |
> | | AEDR | 1.4 | 0.8 | 2.4 | 0.9 | 0.3 | 2.4 | 3.9 | 3.2 |
> | | **Ours** | **91.1** | **83.7** | **95.1** | **74.3** | **65.7** | **92.3** | **95.1** | **94.6** |
> | Saturation=2.0 | Reconstruction | 3.1 | 2.2 | 3.8 | 0.4 | 0.1 | 1.3 | 9.8 | 10.2 |
> | | LatentTracer | 3.6 | 3.7 | 8.2 | 0.2 | 0.0 | 3.4 | 14.5 | 14.5 |
> | | AEDR | 9.5 | 4.2 | 8.7 | 1.7 | 0.4 | 5.5 | 18.4 | 10.1 |
> | | **Ours** | **99.2** | **99.7** | **99.8** | **99.5** | **98.8** | **99.8** | **99.9** | **99.8** |
> | Resize=0.5 | Reconstruction | 1.0 | 1.9 | 4.5 | 0.9 | 0.0 | 2.4 | 9.9 | 10.0 |
> | | LatentTracer | 2.2 | 2.2 | 4.8 | 0.5 | 0.0 | 2.6 | 12.8 | 8.6 |
> | | AEDR | 0.2 | 1.5 | 9.7 | 0.8 | 0.3 | 9.1 | 10.8 | 11.6 |
> | | **Ours** | **98.4** | **98.6** | **99.5** | **96.9** | **93.3** | **99.3** | **99.7** | **99.4** |

---

> ### Author Response · Authors · 2025-11-23
> **Responses to Reviewer 891z (9/13)**
>
> >**W6. Limited baseline comparisons. The paper only compares against LatentTracer (diffusion-specific), AEDR (also diffusion-focused), and naive reconstruction. Comparisons with general deepfake detection methods or universal AI-generated image detectors are missing.**
>
> We would like to highlight that we are the first to explore the area of data provenance in the context of image autoregressive models. Thus, **we adapted existing methods proposed for GAN and diffusion models, namely: Reconstruction, LatentTracer, and AEDR to image autoregressive models as baseline methods**.
>
> As for deepfake detection methods, we note that many such methods focus on special artifacts for facial feature editing and are therefore not applicable to IAR attribution. For example, LAA-Net needs to create a mask for the edited face region before training [6].
> Instead, we evaluate a general and very recent AI-generated image detection method, namely AIDE [7] from ICLR 2025.
>
>  We report the results of AIDE with its pre-trained weights in Table A15 and below. We use 1,000 images as belonging and 1,000 images as non-belonging datasets. We note that their approach has a very limited performance for detecting IAR-generated images, and has an even worse performance to distinguish data generated by different models.
>
> | Model | Natural | |  | Generated |  | |  |  |  |
> |:---:|:---:|:---:|:---:|:---:|:---:|:---:|:---:|:---:|:---:|
> |  | ImageNet | LAION | MS-COCO | LlamaGen | RAR | Taming | VAR | Infinity | VQDiff |
> | LlamaGen | 18.8 | 16.8 | 23.2 | - | 0.5 | 0.9 | 3.3 | 6.0 | 6.5 |
> | RAR | 27.9 | 26.8 | 30.6 | 5.2 | - | 4.5 | 9.6 | 15.2 | 15.9 |
> | Taming | 29.4 | 25.8 | 34.3 | 1.4 | 0.2 | - | 4.3 | 8.8 | 9.5 |
> | VAR | 14.6 | 12.5 | 18.4 | 0.2 | 0.1 | 0.2 | - | 3.4 | 3.7 |
> | Infinity | 5.6 | 4.6 | 7.4 | 0.0 | 0.0 | 0.0 | 0.5 | - | 1.2 |
> | VQ-Diffusion | 10.3 | 9.2 | 13.1 | 0.0 | 0.0 | 0.0 | 0.2 | 0.7 | - |
>
> To further evaluate against AIDE, we re-train their model following the same training procedure as [7] for 5 epochs on 50k images. Importantly, AIDE's training set includes both generated (belonging) and real images, **giving it access to additional natural image data that our method does not use**.
> Despite these advantages, the results shown below (also as Table A16) demonstrate that our method still substantially outperforms AIDE. While AIDE achieves relatively strong performance in the natural vs. generated setting, **it fails in the more critical setting of distinguishing between images generated by different models**, which is the primary focus of data provenance. For instance, for RAR, AIDE achieves only 25.9-73.2% TPR@1%FPR in distinguishing images from other IAR models, whereas our method achieves near-perfect 99.9-100% TPR@1%FPR across all model pairs.
>
> | Model | Method | Fine-tuning Set | Natural |  |  | Generated |  |  |  |  |  |
> |:---:|:---:|:---:|:---:|:---:|:---:|:---:|:---:|:---:|:---:|:---:|:---:|
> |  |  |  | ImageNet | LAION | MS-COCO | LlamaGen | RAR | Taming | VAR | Infinity | VQDiff |
> |RAR | AIDE | RAR Generated + ImageNet | 99.7 | 99.4 | 100.0 | 73.2 | - | 53.7 | 48.8 | 99.5 | 54.9 |
> | | AIDE | RAR Generated + MS-COCO | 97.7 | 98.6 | 100.0 | 41.7 | - | 25.9 | 30.0 | 100.0 | 68.7 |
> |  | Ours | RAR (Generated) | 100.0 | 100.0 | 100.0 | 99.9 | - | 99.9 | 100.0 | 100.0 | 100.0 |
> |Llamagen | AIDE | Llamagen (Generated) + ImageNet | 99.2 | 99.8 | 100.0 | - | 70.4 | 15.5 | 6.2 | 99.8 | 36.1 |
> |  | AIDE | Llamagen Generated + MS-COCO | 86.5 | 97.1 | 99.9 | - | 43.2 | 13.1 | 3.1 | 99.9 | 49.1 |
> |  | Ours | Llamagen (Generated) | 100.0 | 100.0 | 100.0 | - | 100.0 | 100.0 | 100.0 | 100.0 | 100.0 |
>
> We note that **general AI detection methods usually consider general distinctions between generated and real images, but do not leverage specific artifacts in different models and thus fail to attribute an image to a specific model**. Instead, we utilize the codebook of IARs as the inherent *“fingerprint”* of the models. Therefore, our method outperforms the general AI detection method significantly.

---

> ### Author Response · Authors · 2025-11-23
> **Responses to Reviewer 891z (10/13)**
>
> >**W7 The method attributes to the autoencoder, not the autoregressive model. Multiple AR models sharing the same AE are indistinguishable**
>
>
> We thank the Reviewer for the question and conduct the following experiments to address your concern about distinguishing ARs that share the same AE/VAE.
>
> **Evaluating our method on ARs sharing the same AE:**
>
> We evaluate the following two settings where two ARs share the same AE:
>
> **1. LlamaGen Text-to-Image setting:** $AR_1$ trained on LAION-COCO vs $AR_2$ trained on 10M internal high-asthetics quality dataset [11] (both using LlamaGen-AE-T2I)
>
> **2. LlamaGen Class-to-Image setting:** $AR_1$ is LlamaGen-L-256 vs $AR_2$ LlamaGen-XL (both using LlamaGen-AE-C2I)
>
> For both settings, the results show that our method distinguishes the images generated by different ARs with 100%TPR@1%FPR. For the evaluated settings, fine-tuning the inversion decoder on images from one AR transformer is indeed sufficient to distinguish it from another AR using the same AE.
>
> | Task | AE | AR | Belonging data is generated by | Non-belonging data is generated by | TPR@1%FPR (%) |
> |:---:|:---:|:---:|:---:|:---:|:---:|
> | Text-to-Image | LlamaGen-AE-T2I | LlamaGen-AR-T2I-COCO | LlamaGen-AR-T2I-COCO | LlamaGen-AR-T2I-Internal | 100.0 |
> |  |  LlamaGen-AE-T2I | LlamaGen-AR-T2I-Internal | LlamaGen-AR-T2I-Internal | LlamaGen-AR-T2I-COCO | 100.0 |
> | Class-to-Image | LlamaGen-AE-C2I | LlamaGen-L-256 | LlamaGen-L-256 | LlamaGen-XL | 100.0 |
> |  | LlamaGen-AE-C2I  | LlamaGen-XL | LlamaGen-XL | LlamaGen-L-256 | 100.0 |
>
>
> Although our method works very well in the above evaluations, we would like to point out that **the main focus of our work is to attribute an image to a IAR model family, not one specific AR transformer.** Notably, each model family usually designs a specific AE for their tasks. For example, a next-scale image paradigm is developed for the next-scale prediction in VAR and a BSQ-based image autoencoder is designed for Infinity.
> Additionally, in our application to prevent model collapse, a model owner cares most about whether an image is generated by their model family to prevent re-training on these data, instead of by a specific transformer.
> Similar to the setup of our work, existing AI data provenance works, including RONAN, LatentTracer, and AEDR, are focused on attributing an image to a model family/series instead of targeting a fine-grained model (like AR) attribution problem.
>
>
> Moreover, we would like to point out that **many ARs sharing exactly the same AE are less common in real-world scenarios.**
> When model owners adapt an AE for their own task, it is more reasonable for the model owner to first train the existing AE on their own image dataset, such that the AE performs better on their own dataset.
> For example, LlamaGen mentioned that they need to train different AEs for their **class-to-image** image generation (*”AE is trained on ImageNet”*) and **text-to-image** generation (*”AE is trained on 50M LAION-COCO and 10M internal high aesthetic quality data”*), as they use different datasets for the two tasks.
>
> We provide the following case to show that our method can perfectly distinguish an AE with the same architecture trained on different datasets.
>
> **Table. Same model family, AE finetuned on different datasets.** For the Text-to-Image setting, the AE fine-tuning data and evaluation belonging data are generated by different prompts. Similarly, for the Class-to-Image setting, the AE fine-tuning data and evaluation belonging data are generated from different classes.
>
> | AE Architecture | AE Training Data | AR | Belonging data is generated by | Non-belonging data is generated by | TPR@1%FPR (%) |
> |:---:|:---:|:---:|:---:|:---:|---:|
> | LlamaGen-AE | COCO and Internal | LlamaGen-T2I | LlamaGen-T2I | LlamaGen-C2I | 100.0 |
> | LlamaGen-AE | ImageNet | LlamaGen-C2I | LlamaGen-C2I | LlamaGen-T2I | 100.0 |
>
> In conclusion, we 1) provide experiments for two settings where our method successfully attributes between two ARs sharing the same AE, 2) clarify that we mainly focus on model family attribution, and 3) show that our method can distinguish two IARs within the same model family but with differently trained AEs.

---

> ### Author Response · Authors · 2025-11-23
> **Responses to Reviewer 891z (11/13)**
>
> >**W8 Presentation issues**
>
> *Table A4, A5 should be in the main paper*: While we fully agree that the results from Table A4 and A5 are an important part of the paper, these two full tables are simply too large for the main content. However, we included three key observations of these two tables in the main content:
>
> 1) Our method largely outperforms the baseline methods (Table 1).
>
> 2) Using the optimized quantization and combining our two loss metrics give us a performance boost (Table 3).
>
> 3) Our inverse decoder is critical for most models (Table 4)
>
> According to the suggestion of the Reviewer, we further moved more results about observation 3 into Table 4. In addition, we also include the information for EncLoss calibration in Table A4 and A5 in the main content (provide Table 5 in the revised manuscript).  We also added more descriptions of the results and referred the audience to Table A4 and A5 for a comprehensive view of the full results.
>
> *Overclaimed as the first provenance framework*: We would like to point out that we explicitly say that we are the “first post-hoc provenance framework *for IAR-generated images*”. While reconstruction-based methods such as RONAN, allow for provenance in GANs and diffusion models, we are the first to propose a provenance framework explicitly targeted for IARs leveraging their specific artifacts.
>
> *Algorithm 3 lacks details*:  We use 1200 iterations, a learning rate of 0.1, a batch size of 8 and the Adam optimizer. As initialization we use the original quantization in VAR. We added this additional description of our hyperparameters for Algorithm 3 to Appendix C.

---

> ### Author Response · Authors · 2025-11-23
> **Responses to Reviewer 891z (12/13)**
>
> >**Q3 Have you tested adversarial robustness? An adversary aware of your method could potentially add carefully crafted noise to increase QuantLoss for generated images or decrease it for natural images.**
>
> We would like to point out that our method is primarily designed for the benign setting, where model owners leverage our framework to prevent model collapse and ensure responsible deployment of their trained models. However, to assess the robustness of our approach, we also consider the most challenging adversarial scenario, where a malicious model owner intentionally attempts to evade our detection mechanism.
>
> **Threat Model.** In this adaptive attack scenario, we assume the adversary has knowledge of our methodology. The adversary's goal is to craft adversarial perturbations that increase the distance between the feature map of a generated image and its corresponding codebook entries, thereby causing belonging images to be misclassified as non-belonging images.
>
> **Attack Formulation.** Specifically, the adversarial model owner fine-tunes an inverse decoder and performs an adversarial attack on a belonging image $x$ by minimizing the following adversarial loss:
>
> $$\mathcal{L}_{\text{adv}}(x, \delta, D^{-1}) = -||D^{-1}(x) - \text{sg}(Q^{-1}(Q(D^{-1}(x))))||_2 + \lambda||\delta||_2,$$
>
> where $\delta$ denotes the adversarial perturbation, $\text{sg}(\cdot)$ denotes stop gradient operation, and $\lambda$ controls the trade-off between attack effectiveness and perturbation magnitude. The adversarial sample is constructed as $x_\text{adv}=x+\delta$. This loss function aims to maximize the QuantLoss while constraining the perturbation to remain imperceptible.
>
> **Results and Analysis.** The results are presented here and in Table A16. We evaluate our method under two attack strengths: $\epsilon=1/255$ and $\epsilon=2/255$. Several key observations emerge from these experiments: **First**, fine-tuning with augmentation significantly improves robustness against adaptive attacks. We attribute this to the fact that augmentation-based training enables the inverse decoder to recover the original tokens even under image degradations, which also generalizes to resilience against adversarial perturbations. **Second**, our method demonstrates strong robustness to relatively small adversarial perturbations ($\epsilon=1/255$), maintaining high TPR@1%FPR across most datasets when fine-tuned with augmentations (e.g., 97.4% on ImageNet). Even under stronger attacks ($\epsilon=2/255$), our augmentation-based approach retains considerable detection capability (e.g., 51.1% on MS-COCO), substantially outperforming all baselines. Notably, the baseline methods show very limited robustness even to weak attacks which are not even tailored to them. The TPR@1%FPR for baseline methods drops below 20% in most cases for $\epsilon=2/255$.
>
> **Table: Evaluation under adaptive adversarial attack.** The evaluated model is RAR. This is also table A17 in the Appendix.
>
> |  ε | Method |  **Natural** | | | **Generated** | | | | |
> |---|--------|----------|-------|---------|----------|--------|-----|----------|--------|
> | | | ImageNet | LAION | MS-COCO | LlamaGen | Taming | VAR | Infinity | VQDiff |
> | 1/255 | Reconstruction | 2.5 | 3.0 | 6.9 | 0.2 | 0.0 | 2.3 | 16.5 | 16.0 |
> | | LatentTracer | 5.6 | 5.9 | 15.1 | 0.1 | 0.0 | 4.6 | 24.6 | 25.4 |
> | | AEDR | 18.9 | 13.4 | 30.1 | 7.4 | 1.4 | 22.6 | 42.3 | 22.8 |
> | | Ours (Fine-tuned w/o Aug) | 68.7 | 76.3 | 76.6 | 57.3 | 49.0 | 69.2 | 84.6 | 86.9 |
> | | Ours (Fine-tuned w/ Aug) | **97.4** | **96.7** | **97.7** | **90.2** | **73.6** | **96.3** | **99.0** | **98.9** |
> | 2/255 | Reconstruction | 1.4 | 1.8 | 4.5 | 0.4 | 0.0 | 1.3 | 13.6 | 12.5 |
> | | LatentTracer | 3.4 | 3.4 | 11.1 | 0.0 | 0.0 | 2.4 | 18.4 | 19.5 |
> | | AEDR | 10.7 | 6.9 | 18.2 | 3.2 | 0.2 | 14.1 | 28.7 | 14.4 |
> | | Ours (Fine-tuned w/o Aug) | 0.3 | 0.6 | 0.6 | 0.0 | 0.0 | 0.3 | 1.9 | 2.8 |
> | | Ours (Fine-tuned w/ Aug) | **49.3** | **43.6** | **51.1** | **15.9** | **2.7** | **40.0** | **64.7** | **60.7** |
>
> These results demonstrate that while adaptive attacks can degrade detection performance, our framework maintains significantly better robustness compared to existing methods, particularly when trained with augmentations. The robustness to adaptive attacks makes our method a practical solution even in the more challenging adversarial scenarios.

---

> ### Author Response · Authors · 2025-11-23
> **Responses to Reviewer 891z (13/13)**
>
> >**Q9 The overhead of finetuning is claimed to be "relatively small", but 50 epochs on 50K images is substantial. Can you quantify the total computational cost (GPU hours) and compare it to the cost of training the IAR itself?**
>
> We estimate the pre-training time of different models as follows and compare it with our inverse decoder fine-tuning time.
>
> | Stage | LlamaGen | RAR | Taming | VQDiff | Infinity | VAR |
> |-------|----------|-----|--------|--------|----------|-----|
> | Model Pre-training (hours) | 18000 | 20000 | 20000 | 10000 | 50000 | 20000 |
> | Inverse decoder fine-tuning (hours) | 9 | 32 | 42 | 5 | 15 | 11 |
>
> Notably, our inverse decoder fine-tuning is a relatively small overhead compared to the model pre-training stage. For example, the fine-tuning time is less than 0.05\% compared to the pre-training time for LlamaGen. We also include this information and descriptions in Appendix G and Table A3.
>
>
> >**Q10 Have you considered zero-shot or few-shot scenarios where only a small number of generated samples are available for fine-tuning? This would be more realistic for detecting newly-released or proprietary models.**
>
>
> **Zero-shot scenario:** We would like to highlight that our EncLoss, QuantLoss and Combined Loss perform well even in a zero-shot setting on LlamaGen. The results are shown below and in Table A5 of the submission.
>
> **Table. Zero-shot setting for LlamaGen, where the original encoder is directly used as the inverse decoder.**
> | Method | ImageNet | LAION | MS-COCO | LlamaGen | RAR | Taming | VAR | Infinity | VQDiff |
> |:---:|:---:|:---:|:---:|:---:|:---:|:---:|:---:|:---:|:---:|
> | EncLoss | 99.7 | 83.5 | 99.6 | - | 99.6 | 94.2 | 99.6 | 99.8 | 99.8 |
> | QuantLoss | 98.9 | 78.2 | 99.9 | - | 99.9 | 98.4 | 98.2 | 100.0 | 97.5 |
> | CombLoss | 99.9 | 99.6 | 99.9 | - | 99.9 | 99.6 | 99.9 | 99.9 | 100.0 |
>
> **Application to newly-released and proprietary models:** We note that a newly-released IAR, as long as the model is *open-source*, can be used to generate many samples to finetune an inverse decoder. Under the setting of *proprietary* models, our method can mainly be used by the model owner to **prevent model collapse** and **ensure responsible AI release**, as explained in W2.1.
>
>
> **References**
>
> [1] Louis Kerner, Michel Meintz, Bihe Zhao, Franziska Boenisch, and Adam Dziedzic. Bitmark
> for Infinity: Watermarking bitwise autoregressive image generative models. NeurIPS2025
>
> [2] Nikola Jovanović, Ismail Labiad, Tomáš Souček, Martin Vechev, Pierre Fernandez. Watermarking Autoregressive Image Generation. NeurIPS2025
>
> [3] Yu Tong, Zihao Pan, Shuai Yang, and Kaiyang Zhou. Training-free watermarking for autoregressive image generation.
>
> [4] Wang, Zhenting, et al. "Where Did I Come From? Origin Attribution of AI-Generated Images." NeurIPS 2023.
>
> [5] Frank E Grubbs. Sample criteria for testing outlying observations. The Annals of Mathematical Statistics, pages 27–58, 1950.
>
> [6] Nguyen, Dat, et al. "Laa-net: Localized artifact attention network for quality-agnostic and generalizable deepfake detection." CVPR 2024.
>
> [7] Yan, Shilin, et al. "A Sanity Check for AI-generated Image Detection." ICLR 2025.
>
> [8] Alemohammad, Sina, et al. "Self-consuming generative models go mad." ICLR. 2023.
>
> [9]Shumailov, Ilia, et al. "AI models collapse when trained on recursively generated data." Nature. 2024
>
> [10] https://openai.com/index/launching-sora-responsibly/
> “Distinguishing AI content: Every video generated with Sora includes both visible and invisible provenance signals. At launch, all outputs carry a visible watermark. All Sora videos also embed C2PA metadata—an industry-standard signature—and we maintain internal reverse-image and audio search tools that can trace videos back to Sora with high accuracy, building on successful systems from ChatGPT image generation and Sora 1.”
>
> [11] Peize Sun, Yi Jiang, Shoufa Chen, Shilong Zhang, Bingyue Peng, Ping Luo, and Zehuan Yuan. “Autoregressive model beats diffusion: Llama for scalable image generation.” 2024.
>
>
> ***
>
> We thank the Reviewers for their valuable and extensive questions, which help make our submission much more solid and clear. If our rebuttal addresses the Reviewer's concerns, we would appreciate it very much if they consider updating the score. We are also happy to address any remaining questions.

---

> > ### Comment · Reviewer_891z · 2025-11-27
> > **Reviewer's response**
> >
> > Thank authors for the detailed response, most of my concerns are solved. Thus I increased my score to 8.

---

> > > ### Author Response · Authors · 2025-11-27
> > > **Thank you for the comprehensive review!**
> > >
> > > We would like to thank the reviewer for the comprehensive review and the score raise. We really appreciate the reviewer's insightful suggestions, which we believe improve our work a lot. If there are any further questions, we are more than happy to clarify them.

---

> ### Author Response · Authors · 2025-12-03
> **Additional Baselines: MIA-based and Classifier-based Methods**
>
> Regarding the Reviewer’s suggestion for adding comparisons to more baseline methods (W6), we provided the evaluation of an additional baseline (AIDE) in the previous responses. Apart from AIDE, below we provide 3 additional baseline methods: two MIA-based (CFG-Diff & ICAS) and classifier-based (D³QE).
> ## 1. Two MIA-based Methods (CFG-Diff & ICAS)
> We clarified that membership inference attacks (MIAs) and our generated data provenance are two different tasks, and that MIAs cannot be applied to our task because of the additional, over-strict requirements for the labels or prompts of a generated image. In this response, we further explore what could be the **upper bound** of MIAs if given the additional information of labels for data provenance. Concretely, we provided the MIA-based methods with **the ground truth labels for both generated and real images**, which are usually **absent in the real world**. We evaluated two MIA-based approaches that the Reviewer mentioned: CFG-Diff [1] and ICAS [2]. The TPR\@1%FPR (\%) for the two baselines and our method are shown as follows.
>
>
> | Model | Method   | ImageNet | LlamaGen | RAR   | Taming | VAR   |
> |-------|----------|----------|----------|-------|--------|-------|
> | RAR   | CFG-Diff | 30.9     | 96.9     | -     | **100.0** | 99.9  |
> |       | ICAS     | 95.4     | 99.7     | -     | 99.9   | 99.7  |
> |       | Ours     | **100.0**| **99.9** | -     | 99.9   | **100.0** |
> | VAR   | CFG-Diff | 2.5      | 6.2      | 16.4  | 54.9   | -     |
> |       | ICAS     | 7.1      | 24.4     | 44.7  | 66.0   | -     |
> |       | Ours     | **100.0**| **99.2** | **100.0** | **100.0** | -     |
>
> The results demonstrate that **our method outperforms the two MIA-based approaches in nearly every case, even if we evaluate the upper bound performance of these methods by providing ground truth labels**. Notably, the two MIA-based approaches have a very low performance for VAR. They also have a lower TPR\@1%FPR (\%) when using the real images as non-belonging data than using generated images, which means that **the MIA-based approaches tend to attribute many real images to one of the generative models**. On the contrary, our method achieves low FPR, regardless of the types of non-belonging data.
> ## 2. Additional Classifier-based Method (D³QE)
> We additionally report the results for another newly released classifier-based method for IAR-generated image detection, namely D³QE [3]. The method is designed to distinguish IAR-generated images from natural images, while our approach is designed for a more extensive scope: attributing an image to a IAR model, instead of only detecting if it’s real or IAR-generated. Note that D³QE was only published on October 7, which is after the ICL submission deadline (on September 24th). We compare the TPR@1%FPR of our method and D³QE as follows.
>
>
> | Model | Method | **Natural** | | | **Generated** | | | | | |
> |:---:|:---:|:---:|:---:|:---:|:---:|:---:|:---:|:---:|:---:|:---:|
> | | | ImageNet | LAION | MS-COCO | LlamaGen | RAR | Taming | VAR | Infinity | VQDiff |
> | LlamaGen | D³QE | 86.9 | 67.7 | 86.6 | - | 6.8 | 2.0 | 2.0 | 60.1 | 3.7 |
> | | Ours | **100.0** | **100.0** | **100.0** | - | **100.0** | **100.0** | **100.0** | **100.0** | **100.0** |
> | RAR | D³QE | 78.0 | 49.7 | 77.5 | 0.0 | - | 0.2 | 0.2 | 42.2 | 0.4 |
> | | Ours | **100.0** | **100.0** | **100.0** | **99.9** | - | **99.9** | **100.0** | **100.0** | **100.0** |
> | Taming | D³QE | 78.0 | 49.7 | 77.5 | 0.0 | - | 0.2 | 0.2 | 42.2 | 0.4 |
> | | Ours | **100.0** | **100.0** | **100.0** | **100.0** | **100.0** | - | **100.0** | **100.0** | **100.0** |
> | VAR | D³QE | 73.5 | 52.2 | 72.3 | 0.0 | 3.5 | 1.4 | - | 46.7 | 2.3 |
> | | Ours | **100.0** | **99.2** | **100.0** | **99.2** | **100.0** | **100.0** | - | **100.0** | **100.0** |
> | Infinity | D³QE | 6.3 | 1.5 | 5.9 | 0.0 | 0.1 | 0.0 | 0.0 | - | 0.0 |
> | | Ours | **99.4** | **85.6** | **99.4** | **99.2** | **99.5** | **99.1** | **99.4** | - | **99.4** |
> | VQDiff | D³QE | 49.9 | 31.6 | 49.2 | 0.0 | 2.1 | 0.5 | 0.5 | 27.8 | - |
> | | Ours | **100.0** | **99.4** | **100.0** | **99.9** | **100.0** | **99.9** | **100.0** | **100.0** | - |
>
>
> The results show that our approach has a significantly better performance than D³QE across all the evaluated models and datasets. Notably, D³QE has very limited performance when the belonging and non-belonging images are from different IAR model families, but our method can successfully attribute an image to the correct model, achieving nearly 100%TPR@1%FPR for all settings.
>
> **References**
>
> [1] Kowalczuk, Antoni, et al. "Privacy Attacks on Image AutoRegressive Models." ICML 2025.
>
> [2] Yu, Hongyao, et al. "Icas: Detecting training data from autoregressive image generative models." ACM MM 2025.
>
> [3] Zhang, Yanran, et al. "D3QE: Learning Discrete Distribution Discrepancy-aware Quantization Error for Autoregressive-Generated Image Detection." ICCV 2025.

---

### Official Review · Reviewer_XMV2 · 2025-10-31

**Soundness:** 3
**Presentation:** 4
**Contribution:** 3
**Rating:** 8
**Confidence:** 2

**Summary:**

This paper proposes a data provenance method for image auto-regressive generation models, based on two carefully designed losses.

**Strengths:**

1) Their method does not require modifications to the models or images, and is applicable to already released ones.
1) Their insight of QuantLoss and EncLoss is intuitive.
1) They propose an optimized algorithm to address multi-scale VAR.
1) Their experimental results are excellent, achieving 100% success in most cases.
1) Their writing is clear and fluent.

**Weaknesses:**

1) They state that the idea of the two signals is based on their observations, but there is no quantitative presentation of their observations in the main text.
1) Lack of the formalization of greedy search, as well as the comparisons with Algorithm 3 in terms of performance and cost.
1) According to Table A3, the cost of Algorithm 3 is relatively high. Perhaps it should be accelerated.

Minor revisions:
1) The $D^{-1}$ in Line 253 and the $Q$ in Line 270 should be italic.
1) What are the $R$ in Line 731 and the $t$ in Line 749?
1) In Table 1, it is difficult to find the boundary between "Natural" and "Generated".

**Questions:**

1) Consider a scenario of adaptive attack, where the model owner is aware of your detector and intends to mislead you. How might he do, and how would you handle it? Could you give some discussion?
1) Why is the 2-norm used for all distances? Will there be any difference if using other distances?

---

> ### Author Response · Authors · 2025-11-22
> **Responses to Reviewer XMV2 (1/3)**
>
> We thank the Reviewer for their comments and questions. We appreciate that the Reviewer recognizes the effective applicability of our framework and our excellent experimental results.
>
> In summary, we have clarified the intuition behind the observation of the two signals and show it quantitatively, improved the efficiency of Algorithm 3, and conducted an analysis of an adaptive attack together with the defense strategy.
>
> We would like to clarify the concerns and answer the questions with additional experiments as follows.
>
> >**W1: No quantitative presentation of the two observed signals in the main text.**
>
> Our initial observation is that the token representations differ significantly between natural and IAR-generated images. Intuitively, the representations of generated images are consistently closer to the codebook entries than those of natural images (shown in Figure 1).
> We use our inverse decoder to obtain the token representations for natural and generated images and compare their distance to the closest token representations in the codebook.
>
> | Model | Natural | Generated |
> | :--- | :--- | :--- |
> | LlamaGen | 0.0108 ($\pm$0.000) | 0.0033 ($\pm$0.001) |
> | RAR | 0.3942 ($\pm$0.030) | 0.1538 ($\pm$0.037) |
> | Taming | 0.0225 ($\pm$0.002) | 0.0094 ($\pm$0.003) |
> | VQ-Diffusion | 0.0216 ($\pm$0.003) | 0.0086 ($\pm$0.002) |
> | Infinity | 0.0116 ($\pm$0.000) | 0.0109 ($\pm$0.000) |
> | VAR | 0.1381 ($\pm$0.006) | 0.1075 ($\pm$0.011) |
>
> The results, shown here and added to Table A6 in  Appendix K, highlight that the distances between token representations and codebook entries are much smaller for the *generated* than for *natural* images.
>
> >**W2: Lack of the formalization of greedy search, as well as the comparisons with Algorithm 3.**
>
> We would like to clarify that the greedy search refers to the original quantization algorithm in VAR, which is formalized in Algorithm 2 of the original submission. We extended the description in Section 3.3.1 of our paper to make the connection more clear.
>
> Additionally we show the direct comparison of performance and cost between the original quantization based on greedy search (Algorithm 2) and our optimized quantization (Algorithm 3) in the table below.
>
> | Method | Algorithm | Performance (TPR@1%FPR) | Time (seconds/image) |
> |:---:|:---:|:---:|:---:|
> | Original VAR quantization (greedy search) | Algorithm 2 | 0.4 | 0.004 |
> | Optimized quantization+1200 iterations (ours) | Algorithm 3 | 95.0 | 8.24 |
>
> The results show that the original quantization does not allow data provenance, whereas our method achieves 95.0%TPR@1%FPR. Our algorithm trades computation time with improved data provenance. Additionally, we accelerate our optimized quantization algorithm in response to the next question.
>
> >**W3: Perhaps Algorithm 3 should be accelerated.**
>
> We would like to thank the Reviewer for the suggestion and investigate two different options how Algorithm 3 could be accelerated.
>
> **1. Faster implementation**
>
> Our algorithm benefits from using quicker engineering implementations, such as:
> Using the Einstein summation convention for calculating the codebook distance.
> Using torch.compile to optimize the calculation of the feature map.
>
> These two techniques reduced the runtime of our method from 8.24s/image to 7.79s/image. The algorithm may be further accelerated with new developments in the deep learning toolkit.
>
>  **2. Fewer iterations**
>
> Our method still maintains high detection performance and can be accelerated a lot with fewer iterations. In Table 1 of the original submission, we used 1200 iterations to achieve a detection performance of 95.0%TPR@1%FPR. However, we performed additional experiments, added to Table A20, and found that our method still achieves 87.5%TPR @1% FPR with only 100 iterations. This reduced the runtime from 8.24s/image to 0.57s/image.
>
> | Method | Iterations | TPR@1%FPR (%) | Time (seconds/image) |
> |:---|:---|:---:|:---:|
> | Default| 1200 | 95.0 | 8.24 |
> | Less Iterations | 100 | 87.5 | 0.57 |
> | Accelerated with Torch| 1200 | 94.8 | 7.79 |
>
> Moreover, in Table A3 the time cost is computed with a batch size of 1. In practice, the required time can be amortized when Algorithm 3 is executed on a mini-batch of images. For example, the per-image time cost can be reduced to 0.18s/image for 100 iterations when using a batch size of 8.
>
> >**W4: Formatting issues**
>
> We updated the notations in line 253 and 270 and would like to thank the Reviewer for the suggestion and carefully reading our submission. We updated the notations and comments in Algorithm 1 and 2. In both algorithms, the token maps from all scales are now denoted as $t$ instead of $R$. Additionally we added a line break between “Natural” and “Generated” to Table 1, Table 3 and all related tables in the Appendix.

---

> ### Author Response · Authors · 2025-11-22
> **Responses to Reviewer XMV2 (2/3)**
>
> >**Q1: Consider a scenario of adaptive attack, where the model owner is aware of your detector and intends to mislead you. How might he do it, and how would you handle it? Could you give some discussion?**
>
> We would like to point out that our method is primarily designed for a benign setting, where model owners leverage our framework to prevent model collapse and ensure responsible deployment of their trained models. However, to assess the robustness of our approach, we also consider the most challenging adversarial scenario where a malicious model owner intentionally attempts to evade our detection mechanism. We design the following attack for this setting:
>
> **Attack Formulation.** The adversarial model owner fine-tunes an inverse decoder and performs an adversarial attack on a belonging image $x$ by minimizing the following adversarial loss:
>
> $$\mathcal{L}_{\text{adv}}(x, \delta, D^{-1}) = -||D^{-1}(x) - \text{sg}(Q^{-1}(Q(D^{-1}(x))))||_2 + \lambda||\delta||_2,$$
>
> where $\delta$ denotes the adversarial perturbation,  $\text{sg}(\cdot)$ denotes stop gradient operation, and $\lambda$ controls the trade-off between attack effectiveness and perturbation magnitude. The adversarial sample is constructed as $x_\text{adv}=x+\delta$. This loss function aims to maximize the QuantLoss while constraining the perturbation to remain imperceptible.
>
> **Results and Analysis.** The results are presented here and in Table A16. We evaluate our method under two attack strengths: $\epsilon=1/255$ and $\epsilon=2/255$. Several key observations emerge from these experiments: **First**, fine-tuning with augmentation significantly improves robustness against adaptive attacks. We attribute this to the fact that augmentation-based training enables the inverse decoder to recover the original tokens even under image degradations, which also generalizes to resilience against adversarial perturbations. **Second**, our method demonstrates strong robustness to relatively small adversarial perturbations ($\epsilon=1/255$), maintaining high TPR@1%FPR across most datasets when fine-tuned with augmentations (e.g., 97.4% on ImageNet). Even under stronger attacks ($\epsilon=2/255$), our augmentation-based approach retains considerable detection capability (e.g., 51.1% on MS-COCO), substantially outperforming all baselines. Notably, the baseline methods show very limited robustness even to weak attacks which are not even tailored to them. The TPR@1%FPR for baseline methods drops below 20% in most cases for $\epsilon=2/255$.
>
> **Table: Evaluation under adaptive adversarial attack.** The evaluated model is RAR.
>
> |  ε | Method |  **Natural** | | | **Generated** | | | | |
> |---|--------|----------|-------|---------|----------|--------|-----|----------|--------|
> | | | ImageNet | LAION | MS-COCO | LlamaGen | Taming | VAR | Infinity | VQDiff |
> | 1/255 | Reconstruction | 2.5 | 3.0 | 6.9 | 0.2 | 0.0 | 2.3 | 16.5 | 16.0 |
> | | LatentTracer | 5.6 | 5.9 | 15.1 | 0.1 | 0.0 | 4.6 | 24.6 | 25.4 |
> | | AEDR | 18.9 | 13.4 | 30.1 | 7.4 | 1.4 | 22.6 | 42.3 | 22.8 |
> | | Ours (Fine-tuned w/o Aug) | 68.7 | 76.3 | 76.6 | 57.3 | 49.0 | 69.2 | 84.6 | 86.9 |
> | | Ours (Fine-tuned w/ Aug) | **97.4** | **96.7** | **97.7** | **90.2** | **73.6** | **96.3** | **99.0** | **98.9** |
> | 2/255 | Reconstruction | 1.4 | 1.8 | 4.5 | 0.4 | 0.0 | 1.3 | 13.6 | 12.5 |
> | | LatentTracer | 3.4 | 3.4 | 11.1 | 0.0 | 0.0 | 2.4 | 18.4 | 19.5 |
> | | AEDR | 10.7 | 6.9 | 18.2 | 3.2 | 0.2 | 14.1 | 28.7 | 14.4 |
> | | Ours (Fine-tuned w/o Aug) | 0.3 | 0.6 | 0.6 | 0.0 | 0.0 | 0.3 | 1.9 | 2.8 |
> | | Ours (Fine-tuned w/ Aug) | **49.3** | **43.6** | **51.1** | **15.9** | **2.7** | **40.0** | **64.7** | **60.7** |
>
>
> While our method remains considerable robustness under adaptive attack, we would like to point out that our method is mostly designed for two scenarios where the model owner is a benign actor, including **preventing model collapse** and **ensuring responsible AI release**:
>
> *1. Prevent model collapse.*
> The success of generative models relies on highly data intensive training. This data is typically scraped from web scale internet collection, with growing amounts of generated rather than natural data. Iterative training on model generated data has been demonstrated to cause performance degradation [A][B], a phenomenon referred to as **model collapse**. Our method provides model owners with an automated and effective tool to identify and filter generated images when training for future model versions.
>
>
> *2. Responsible AI release.*
> Regulatory frameworks increasingly mandate transparency for synthetic content, for example, the EU AI Act Article 50 Recital 134 [C].Therefore, we expect that the model owners would be willing to provide an API to check if an image is generated even for a closed-source model. Many services already provide such methods, for example, SORA adds both visible and invisible watermarks on its generated videos as part of its responsible AI development goals [D].

---

> ### Author Response · Authors · 2025-11-22
> **Responses to Reviewer XMV2 (3/3)**
>
> >**Q2: Why is the 2-norm used for all distances? Will there be any difference if using other distances?**
>
> For EncLoss, we use the L2 norm since it’s a standard metric to measure pixel-level difference between two images.
> For QuantLoss, we use the L2 norm following the original quantization algorithms in IARs. We show here and in Table A13 that using cosine similarity yields similar results as the L2 norm. Our key finding is that belonging images are closer to the codebook entries compared to non-belonging images, where two distance metrics can both capture the distance difference.
>
> **Table. TPR@1%FPR (%) comparison when using different distance metrics.** The evaluated model is RAR.
> | Distance Metric | Natural | | | Generated | | | | |
> |-----------------|---------|-------|---------|----------|--------|-----|---------|-------|
> |                | ImageNet | LAION | MS-COCO | LlamaGen | Taming | VAR | Infinity | VQDiff |
> | Cosine Distance | 100.0 | 100.0 | 100.0 | 100.0 | 99.9 | 100.0 | 100.0 | 100.0 |
> | L2 Norm | 100.0 | 100.0 | 100.0 | 99.9 | 99.9 | 100.0 | 100.0 | 100.0 |
>
> **References**
>
> [A] Alemohammad, Sina, et al. "Self-consuming generative models go mad." ICLR. 2023.
>
> [B] Shumailov, Ilia, et al. "AI models collapse when trained on recursively generated data." Nature. 2024.
>
> [C] The EU AI Act Article 50 Recital 134.*”Further to the technical solutions employed by the providers of the AI system, deployers who use an AI system to generate or manipulate image, audio or video content that appreciably resembles existing persons, objects, places, entities or events and would falsely appear to a person to be authentic or truthful (deep fakes), should also clearly and distinguishably disclose that the content has been artificially created or manipulated by labelling the AI output accordingly and disclosing its artificial origin.”*
>
> [D] https://openai.com/index/launching-sora-responsibly/
> “Distinguishing AI content: Every video generated with Sora includes both visible and invisible provenance signals. At launch, all outputs carry a visible watermark. All Sora videos also embed C2PA metadata—an industry-standard signature—and we maintain internal reverse-image and audio search tools that can trace videos back to Sora with high accuracy, building on successful systems from ChatGPT image generation and Sora 1.”
>
> ***
>
> We thank the Reviewers for their valuable questions, which help make our submission more solid and clear! We are also happy to address any further questions if needed.

---

> > ### Comment · Reviewer_XMV2 · 2025-11-26
> >
> > Thank you for your reply. I will keep my high score.

---

> > > ### Author Response · Authors · 2025-11-27
> > > **Thank you for your review!**
> > >
> > > We would like to thank the reviewer for recognizing the value of our work with a high score! We also appreciate the insightful feedback from the reviewer, which makes our work more solid and clear.

---

### Official Review · Reviewer_Rqvo · 2025-11-01

**Soundness:** 3
**Presentation:** 3
**Contribution:** 2
**Rating:** 4
**Confidence:** 4

**Summary:**

This paper addresses the critical need for data provenance in Image Autoregressive Models (IARs). The authors propose the first post-hoc, model-agnostic provenance framework that requires no modifications to IAR training/generation processes. Specifically, the paper proposes two nove signals for data provenance, which are QuantLoss-based and EncLoss-based, respectively. The QuantLoss-based signal compares the original feature map and re-quantized one of the input image, while the EncLoss-based siganl measures the input image and the re-encoded image. The experiments have demonstrated the excellent effectiveness of the proposed method.

**Strengths:**

- Effectiveness: the proposed method requires no modification to the model’s training or generation process and exhibits distinguished performance conpared to current baselines.
- Writing: the paper is well-written and easy to follow.
- Holistic Evaluation: the authors evaluate not only detection accuracy but also robustness.

**Weaknesses:**

- This paper seems to transfer the technique of membership inference to the scenario of data provenance, and the authors need to further clarify the differences between the settings of this paper and those of membership inference.
- The number of baseline methods used for comparison in the experimental section is relatively few. It is recommended to compare with more baselines, such as [1] and [2].
- The framework assumes white-box access to the target models. For closed-source IARs, this access may not be available—limiting real-world applicability. The paper does not discuss potential workarounds.

[1] Kowalczuk A, Dubiński J, Boenisch F, et al. Privacy Attacks on Image AutoRegressive Models[C]//Forty-second International Conference on Machine Learning.

[2] Yu H, Qiu Y, Yang Y, et al. ICAS: Detecting Training Data from Autoregressive Image Generative Models[C]//Proceedings of the 33rd ACM International Conference on Multimedia. 2025: 11209-11217.

**Questions:**

Refer to weaknesses.

---

> ### Author Response · Authors · 2025-11-22
> **Responses to Reviewer Rqvo (1/3)**
>
> We thank the Reviewer for the insightful comments. We appreciate that the Reviewer recognizes our holistic evaluation and the effectiveness of our proposed framework.
>
> In summary, we clarified the distinction between membership inference and data provenance, added an additional baseline to our empirical evaluation, and introduced the real-world use cases of our method.
>
> We provide the following clarifications and additional results regarding the comments and questions:
>
> >**W1: The authors need to further clarify the differences between the settings of this paper and those of membership inference.**
>
> We thank the Reviewer for the question. However, we would like to point out that (1) membership inference and dataset provenance are two distinct tasks and (2) membership inference signals are not applicable to data provenance.
>
> **1. Membership inference and dataset provenance are two distinct tasks.**
>
> *Membership inference is focused on the data used to train a given model, while the data provenance is to detect the data generated by the model.* In other words, membership inference attack addresses attribution in the training process, and the data provenance addresses attribution in the generation process.
>
> Below, we present a more specific and detailed setting as well as the further explanation.
>
> The general procedure of a generative model can be defined as:
>
> $$\text{Training Data } \xrightarrow{\text{train}} \text{Generative Model} \xrightarrow{\text{generate}} \text{Synthetic Data}$$
>
>
> * **Membership inference:** detect the **training data** of a model. Given a data point (usually a natural data point) and a model, we detect if this data point was used to **train** this model. The goal of the membership inference attack is to audit the privacy of the training procedure and infer whether a given model leaks information about its private training data.
>
> * **Data provenance:** detect the **generated data** of a model. Given a data point (we do not know if it is natural or synthetic) and a model, we detect if this model **generated** this data point. The goal of data provenance is to trace where the given data is from, especially to trace the source of synthetic images or prevent model collapse caused by training on synthetic data [A][B].
>
> **2. Membership inference signals are not applicable to data provenance.**
>
> In the case of membership inference methods for image generative models, it is necessary to have access to both the class labels or the prompts and the images themselves [1,2] in order to collect membership signals from the models. However, this is not the case for the data provenance task, since **generated images found online are usually not accompanied by their original class information or prompt**, thus, we have to find their origin by relying only on the image itself. On the contrary, our method only needs the image itself for attribution and does not require any additional information, making our method more applicable in real-world scenarios.
>
> We also included the above discussions about the two MIA papers in the **Related Work** of our revised submission.

---

> ### Author Response · Authors · 2025-11-22
> **Responses to Reviewer Rqvo (2/3)**
>
> >**W2: The number of baseline methods used for comparison in the experimental section is relatively few. It is recommended to compare with more baselines, such as [1] and [2].**
>
> We would like to highlight that we are the first to explore the area of data provenance in the context of image autoregressive models. Thus, we were only able to adapt existing methods proposed for GAN and diffusion models (namely: Reconstruction, LatentTracer, and AEDR) to IARs.
>
> In addition, as mentioned in the previous answer, our data provenance setting is inherently different from the membership inference setting, so the MIA baselines do not apply here.
>
> To provide an additional baseline method, we evaluate a state-of-the-art AI-generated image detection method, namely AIDE [C] (from ICLR 2025).  We report the results of AIDE with its pre-trained weights in Table A15 and below. We use 1,000 images as belonging and 1,000 images as non-belonging datasets. We note that their approach has a very limited performance for detecting IAR-generated images, and has an even worse performance to distinguish data generated by different models.
>
> | Model |  | Natural |  |  |  | Generated |  |  |  |
> |:---:|:---:|:---:|:---:|:---:|:---:|:---:|:---:|:---:|:---:|
> |  | ImageNet | LAION | MS-COCO | LlamaGen | RAR | Taming | VAR | Infinity | VQDiff |
> | LlamaGen | 18.8 | 16.8 | 23.2 | - | 0.5 | 0.9 | 3.3 | 6.0 | 6.5 |
> | RAR | 27.9 | 26.8 | 30.6 | 5.2 | - | 4.5 | 9.6 | 15.2 | 15.9 |
> | Taming | 29.4 | 25.8 | 34.3 | 1.4 | 0.2 | - | 4.3 | 8.8 | 9.5 |
> | VAR | 14.6 | 12.5 | 18.4 | 0.2 | 0.1 | 0.2 | - | 3.4 | 3.7 |
> | Infinity | 5.6 | 4.6 | 7.4 | 0.0 | 0.0 | 0.0 | 0.5 | - | 1.2 |
> | VQ-Diffusion | 10.3 | 9.2 | 13.1 | 0.0 | 0.0 | 0.0 | 0.2 | 0.7 | - |
>
> We further adapt AIDE to make it usable for the data provenance task. We re-train their model following the same training procedure as [C] for 5 epochs on 50k images. Importantly, AIDE's training set includes both generated (belonging) and real images, **giving it access to additional natural image data that our method does not use**.
> Despite these advantages, the results shown below (also as Table A17) demonstrate that our method still substantially outperforms AIDE. While AIDE achieves relatively strong performance in the natural vs. generated setting, it fails in the more critical setting of attributing a generated image to a specific model, which is the primary focus of data provenance. For instance, for RAR, AIDE achieves only 25.9-73.2% TPR@1%FPR in distinguishing images from other IAR models, whereas our method achieves near-perfect 99.9-100% TPR@1%FPR across all model pairs.
>
> | Model | Method | Fine-tuning Set | Natural |  |  | Generated |  |  |  |  |  |
> |:---:|:---:|:---:|:---:|:---:|:---:|:---:|:---:|:---:|:---:|:---:|:---:|
> |  |  |  | ImageNet | LAION | MS-COCO | LlamaGen | RAR | Taming | VAR | Infinity | VQDiff |
> | **RAR**| AIDE | RAR Generated + ImageNet | 99.7 | 99.4 | 100.0 | 73.2 | - | 53.7 | 48.8 | 99.5 | 54.9 |
> | | AIDE | RAR Generated + MS-COCO | 97.7 | 98.6 | 100.0 | 41.7 | - | 25.9 | 30.0 | 100.0 | 68.7 |
> |  | Ours | RAR (Generated) | 100.0 | 100.0 | 100.0 | 99.9 | - | 99.9 | 100.0 | 100.0 | 100.0 |
> | **Llamagen** | AIDE | Llamagen (Generated) + ImageNet | 99.2 | 99.8 | 100.0 | - | 70.4 | 15.5 | 6.2 | 99.8 | 36.1 |
> |  | AIDE | Llamagen Generated + MS-COCO | 86.5 | 97.1 | 99.9 | - | 43.2 | 13.1 | 3.1 | 99.9 | 49.1 |
> |  | Ours | Llamagen (Generated) | 100.0 | 100.0 | 100.0 | - | 100.0 | 100.0 | 100.0 | 100.0 | 100.0 |
>
> We note that general AI detection methods usually consider **general distinctions** between generated and real images, but do not leverage specific artifacts in different IAR model families and thus fail to attribute a generated image to a specific model. However, we utilize the codebook of IARs as the **inherent fingerprint** of the model. Therefore, our method outperforms the general AI detection method significantly.

---

> ### Author Response · Authors · 2025-11-22
> **Responses to Reviewer Rqvo (3/3)**
>
> >**W3: The framework assumes white-box access to the target models. For closed-source IARs, this access may not be available—limiting real-world applicability. The paper does not discuss potential workarounds.**
>
> We appreciate this important question. We would like to point out that our framework addresses two critical real-world scenarios where model owners of closed-source IARs would have strong incentives to enable data provenance detection:
>
> **1. Prevent model collapse.**
> The success of image autoregressive models (IARs) relies on highly data-intensive training processes. Training data are typically scraped from web-scale internet collections. However, the growing amount of generated content online has led to an increasing proportion of generated rather than natural data in training corpora.
>
> Iterative training on model-generated data has been demonstrated to cause performance degradation [A][B], a phenomenon referred to as **model collapse**. This presents a critical challenge for model owners (especially major organizations such as OpenAI, Google, and ByteDance) who develop large-scale proprietary models deployed via public APIs. To mitigate potential quality degradation in subsequent model iterations, these organizations require effective methods to identify content generated by their own model families.
>
> Our method provides model owners with an automated and effective tool to identify and filter out synthetic images previously generated by their model family (without changing the generation process) when curating training data for future model versions, thereby addressing this fundamental challenge in sustainable model development.
>
>
> **2. Responsible AI release.**
> Regulatory frameworks increasingly mandate transparency for synthetic content. For example, the EU AI Act Article 50 Recital 134 states: *Further to the technical solutions employed by the providers of the AI system, deployers who use an AI system to generate or manipulate image, audio or video content that appreciably resembles existing persons, objects, places, entities or events and would falsely appear to a person to be authentic or truthful (deep fakes), should also clearly and distinguishably disclose that the content has been artificially created or manipulated by labelling the AI output accordingly and disclosing its artificial origin.*
>
>
> Therefore, we expect that the model owners would be willing to provide an API to check if an image is generated even for a closed-source model. Many services already provide such methods, for example, SORA adds both visible and invisible watermarks on its generated videos as part of its responsible AI development goals [D]. Additionally, we only require white-box access to the autoencoder but black-box access to the transformer.
>
> Moreover, for users to identify generated images, the model owner does not need to provide full access to their model, but only needs to give users access to the inverse decoder and the codebook entries. This can also be done with a black-box API which returns QuantLoss and EncLoss, or simply the answer if a given image was generated by a given model or not (or some confidence score).
>
>
> **References:**
>
> [A] Alemohammad, Sina, et al. "Self-consuming generative models go mad." ICLR. 2023.
>
> [B] Shumailov, Ilia, et al. "AI models collapse when trained on recursively generated data." Nature. 2024.
>
> [C] Yan, Shilin, et al. "A Sanity Check for AI-generated Image Detection." ICLR 2025.
>
> [D] https://openai.com/index/launching-sora-responsibly/
> *“Distinguishing AI content: Every video generated with Sora includes both visible and invisible provenance signals. At launch, all outputs carry a visible watermark. All Sora videos also embed C2PA metadata—an industry-standard signature—and we maintain internal reverse-image and audio search tools that can trace videos back to Sora with high accuracy, building on successful systems from ChatGPT image generation and Sora 1.”*
>
> ---
>
> We thank the Reviewer for the valuable questions, which help make our submission more solid and clear. If our rebuttal addresses the Reviewer's concerns, we would appreciate it very much if  they consider updating the score. We are also happy to address any remaining questions.

---

> ### Author Response · Authors · 2025-12-03
> **Additional Baselines: MIA-based and Classifier-based Methods**
>
> Regarding the Reviewer’s suggestion for adding comparisons to more baseline methods (W2), we provided the evaluation of an additional baseline (AIDE) in the previous responses. Apart from AIDE, below we provide 3 additional baseline methods: two MIA-based (CFG-Diff & ICAS) and classifier-based (D³QE).
> ## 1. Two MIA-based Methods (CFG-Diff & ICAS)
> We clarified that membership inference attacks (MIAs) and our generated data provenance are two different tasks, and that MIAs cannot be applied to our task because of the additional, over-strict requirements for the labels or prompts of a generated image. In this response, we further explore what could be the **upper bound** of MIAs if given the additional information of labels for data provenance. Concretely, we provided the MIA-based methods with **the ground truth labels for both generated and real images**, which are usually **absent in the real world**. We evaluated two MIA-based approaches that the Reviewer mentioned: CFG-Diff [1] and ICAS [2]. The TPR\@1%FPR (\%) for the two baselines and our method are shown as follows.
>
>
> | Model | Method   | ImageNet | LlamaGen | RAR   | Taming | VAR   |
> |-------|----------|----------|----------|-------|--------|-------|
> | RAR   | CFG-Diff | 30.9     | 96.9     | -     | **100.0** | 99.9  |
> |       | ICAS     | 95.4     | 99.7     | -     | 99.9   | 99.7  |
> |       | Ours     | **100.0**| **99.9** | -     | 99.9   | **100.0** |
> | VAR   | CFG-Diff | 2.5      | 6.2      | 16.4  | 54.9   | -     |
> |       | ICAS     | 7.1      | 24.4     | 44.7  | 66.0   | -     |
> |       | Ours     | **100.0**| **99.2** | **100.0** | **100.0** | -     |
>
> The results demonstrate that **our method outperforms the two MIA-based approaches in nearly every case, even if we evaluate the upper bound performance of these methods by providing ground truth labels**. Notably, the two MIA-based approaches have a very low performance for VAR. They also have a lower TPR\@1%FPR (\%) when using the real images as non-belonging data than using generated images, which means that **the MIA-based approaches tend to attribute many real images to one of the generative models**. On the contrary, our method achieves low FPR, regardless of the types of non-belonging data.
> ## 2. Additional Classifier-based Method (D³QE)
> We additionally report the results for another newly released classifier-based method for IAR-generated image detection, namely D³QE [3]. The method is designed to distinguish IAR-generated images from natural images, while our approach is designed for a more extensive scope: attributing an image to a IAR model, instead of only detecting if it’s real or IAR-generated. Note that D³QE was only published on October 7, which is after the ICL submission deadline (on September 24th). We compare the TPR@1%FPR of our method and D³QE as follows.
>
>
> | Model | Method | **Natural** | | | **Generated** | | | | | |
> |:---:|:---:|:---:|:---:|:---:|:---:|:---:|:---:|:---:|:---:|:---:|
> | | | ImageNet | LAION | MS-COCO | LlamaGen | RAR | Taming | VAR | Infinity | VQDiff |
> | LlamaGen | D³QE | 86.9 | 67.7 | 86.6 | - | 6.8 | 2.0 | 2.0 | 60.1 | 3.7 |
> | | Ours | **100.0** | **100.0** | **100.0** | - | **100.0** | **100.0** | **100.0** | **100.0** | **100.0** |
> | RAR | D³QE | 78.0 | 49.7 | 77.5 | 0.0 | - | 0.2 | 0.2 | 42.2 | 0.4 |
> | | Ours | **100.0** | **100.0** | **100.0** | **99.9** | - | **99.9** | **100.0** | **100.0** | **100.0** |
> | Taming | D³QE | 78.0 | 49.7 | 77.5 | 0.0 | - | 0.2 | 0.2 | 42.2 | 0.4 |
> | | Ours | **100.0** | **100.0** | **100.0** | **100.0** | **100.0** | - | **100.0** | **100.0** | **100.0** |
> | VAR | D³QE | 73.5 | 52.2 | 72.3 | 0.0 | 3.5 | 1.4 | - | 46.7 | 2.3 |
> | | Ours | **100.0** | **99.2** | **100.0** | **99.2** | **100.0** | **100.0** | - | **100.0** | **100.0** |
> | Infinity | D³QE | 6.3 | 1.5 | 5.9 | 0.0 | 0.1 | 0.0 | 0.0 | - | 0.0 |
> | | Ours | **99.4** | **85.6** | **99.4** | **99.2** | **99.5** | **99.1** | **99.4** | - | **99.4** |
> | VQDiff | D³QE | 49.9 | 31.6 | 49.2 | 0.0 | 2.1 | 0.5 | 0.5 | 27.8 | - |
> | | Ours | **100.0** | **99.4** | **100.0** | **99.9** | **100.0** | **99.9** | **100.0** | **100.0** | - |
>
>
> The results show that our approach has a significantly better performance than D³QE across all the evaluated models and datasets. Notably, D³QE has very limited performance when the belonging and non-belonging images are from different IAR model families, but our method can successfully attribute an image to the correct model, achieving nearly 100%TPR@1%FPR for all settings.
>
> **References**
>
> [1] Kowalczuk, Antoni, et al. "Privacy Attacks on Image AutoRegressive Models." ICML 2025.
>
> [2] Yu, Hongyao, et al. "Icas: Detecting training data from autoregressive image generative models." ACM MM 2025.
>
> [3] Zhang, Yanran, et al. "D3QE: Learning Discrete Distribution Discrepancy-aware Quantization Error for Autoregressive-Generated Image Detection." ICCV 2025.

---

### Author Response · Authors · 2025-11-22

We would like to thank all the Reviewers for their valuable feedback and insightful comments, which greatly helped us further improve our submission. We appreciate that the Reviewers find our work "addressing an important and timely problem" (Reviewer 891z) by proposing "the first post-hoc, model-agnostic provenance framework" for Image Autoregressive Models (Reviewer Rqvo). Our work is described as novel (Reviewer Rqvo), interesting, and demonstrating excellent effectiveness (Reviewer Rqvo, XMV2, SetU) while covering "multiple state-of-the-art IAR architectures" (Reviewer 891z).

In summary, we address the following questions during our rebuttal:

1. Compare our approach to an additional state-of-the-art baseline method for AI-Detection (AIDE from ICLR 2025), and show that our approach outperforms it even after adapting AIDE to our use-case.
2. Clarify that our data provenance setting differs from membership inference attack (MIA) and why MIA methods are not applicable to our setting.
3. Clarify two important use cases of our method: preventing model collapse and ensuring responsible AI release.
4. Design a statistical test based on Grubbs’ outlier test to provide more theoretical guarantees for our approach.
5. Design a more challenging adaptive adversarial attack and evaluate a defense strategy.
6. More explanation and extensive analysis for the optimized token search (Algorithm 3), including hyperparameter test and acceleration options.
7. Extensive experiments for robustness evaluation, including evaluations on more datasets and combining different post-processing methods.
8. More metrics for our evaluations, including TPR\@0.5%FPR, TPR\@0.1%FPR, AUC, and ROC.
9. Quantify our main observation (based on Figure 1 in the initial submission).
10. Evaluate two distance metrics, L2 norm and cosine similarity, and show our method has similarly good results with both metrics..
11. Provide two experiments where our approach achieves good AR attribution, and clarify our method’s focus on model family attribution.
12. Design and evaluate different methods to combine our QuantLoss and EncLoss, including addition, multiplication, and learned weights.
13. Demonstrate the generalization of our method on different sampling conditional guidance scales, different sampling temperatures, and different data splits.

We also updated our manuscript according to the Reviewer’s questions, and included our new results in the appendix.

Finally, we sincerely thank the Reviewers for their careful reading and constructive feedback, which has helped us improve our work. We have done our best to address the raised concerns through detailed explanations and additional experiments. If our rebuttal addresses the Reviewer's concerns, we would appreciate it very much if  they consider updating the score. We are also happy to address any remaining questions.

---

### Author Response · Authors · 2025-12-03
**Discussion Summary (3/3)**

## 3. Discussion with Reviewer SetU

After we provided the clarifications and experiments that the Reviewer requested, the Reviewer SetU commented, “*Your response has addressed most of my concerns, and I have* **raised my score to 6.**” After we further showed the effectiveness of our approach in the class-split setting, the Reviewer commented, “What a **meaningful finding!** This finding **reveals an inherent connection between images generated by the same generative model**.” We addressed their questions with the following responses:

* Provided additional experiments where our approach achieves good results for AR attribution with different dataset splits, and clarified that our method mainly focuses on model family attribution (Appendix V, Table A22, Table A23, Table A24).

* Clarified that our data provenance setting differs from membership inference attack (MIA) and explained that the MIA-based methods are not applicable to our setting because of the additional requirements for ground truth labels or prompts. Even if the ground truth labels are provided, our method still outperforms the two MIA-based methods (CFG-Diff and ICAS) by a large margin (Appendix W, Table A25).

* Evaluated our method and the baseline methods with more metrics, including TPR@0.5%FPR, TPR@0.1%FPR, AUC, and ROC. Our method demonstrates superior performance across all metrics, even achieving near-perfect TPR at 0.1%FPR for many models (Appendix M, Table A8, Table A9, Table A10, and Figure A5).

* Demonstrated that our method generalizes well on different sampling conditional guidance scales, different sampling temperatures, and different data splits (Appendix N, Table A11, Table A12).


## 4. Discussion with Reviewer Rqvo
Although the Reviewer Rqvo has not had the chance to reply to our response, **all of their questions were also asked by the other Reviewers, and the other Reviewers have confirmed that these questions are properly addressed and gave us a high rating**. We summarize the questions, our responses, and the Reviewers with similar questions as follows.

| Question of Reviewer Rqvo | Our Response and Additional Experiments | Reviewer with Similar Concerns |
|---------------------------|----------------------------------------|--------------------------------------------------------------------------|
| **Question 1**: Explain the difference between membership inference attacks (MIAs) and data provenance. | Clarified that our data provenance setting differs from membership inference attack (MIA) and explained that the MIA-based methods are not applicable to our setting because of the **additional requirements for ground truth labels or prompts**. | **Reviewer SetU** (W2), who confirmed that their concerns have been addressed and raised the score from 4 to 6. |
| **Question 2**: Compare with more baselines, e.g. the two MIA methods (CFG-Diff and ICAS). | 1. We showed that, even if the ground truth labels are provided, our method still outperforms two MIA-based methods (CFG-Diff and ICAS) by a large margin (Appendix W, Table A25). | **Reviewer 891z** (W6), who confirmed that their concerns were addressed and raised the score from 6 to 8. **Reviewer SetU** (W2), who confirmed that their concerns were addressed and raised the score from 4 to 6. |
| | 2. Compared our approach to an additional state-of-the-art baseline method for AI detection (AIDE from ICLR 2025), and showed that our approach outperforms it significantly (Appendix Q, Table A15), even after adapting the baseline to our use case (Appendix Q, Table A17). | |
| | 3. Compared our approach to the newly released baseline method for IAR-generated image detection (D³QE from ICCV 2025, published after our submission), and showed that our approach significantly outperforms it (Appendix Q, Table A16). | |
| **Question 3**: Explain how the method can be applied to closed-source models. | Clarified two important use cases of our method for closed-source models: preventing model collapse and ensuring responsible AI release. | **Reviewer 891z** (Q10), who confirmed that their concerns have been addressed and raised from 6 to 8. |

***

In conclusion, we believe that all the comments and questions by all four Reviewers have been properly addressed. We really appreciate the positive feedback from the Reviewers and the dedicated work of the AC!

---

### Author Response · Authors · 2025-12-03
**Discussin Summary (2/3)**

## 2. Discussion with Reviewer 891z

Review 891z made a very comprehensive review and gave us a 6 before the discussion. After we addressed every comment they had with very detailed explanations and additional experiments, the Reviewer commented, “Thank authors for the detailed response, most of my concerns are solved. Thus I **increased my score to 8.**” We addressed their questions with the following responses:

* Compared our approach to an additional state-of-the-art baseline method for AI detection (AIDE from ICLR 2025), and showed that our approach outperforms it significantly (Appendix Q, Table A15), even after adapting the baseline to our use case (Appendix Q, Table A17)

* Compared our approach to a newly released baseline method for IAR-generated image detection (D³QE from ICCV 2025, published after our submission), and showed that our approach significantly outperforms it (Appendix Q, Table A16).

* Clarified that our data provenance setting differs from membership inference attack (MIA) and explained that the MIA-based methods are not applicable to our setting because of the additional requirements for ground truth labels or prompts. Even if the ground truth labels are provided, our method still outperforms the two MIA-based methods (CFG-Diff and ICAS) by a large margin (Appendix W, Table A25).

* Clarified two important use cases of our method: preventing model collapse and ensuring responsible AI release.

* Designed a statistical test based on Grubbs’ outlier test to provide more rigorous theoretical guarantees for our approach (Appendix T, Table A20).

* Designed a more challenging adaptive adversarial attack and designed a defense strategy that largely improves the robustness of our method against the attack, outperforming the baseline methods (Appendix R, Table A18).

* Provided more explanation and extensive analysis for the optimized token search (Algorithm 3), including hyperparameter test (Table 6) and acceleration options (Appendix U, Table A21).

* Extended our robustness evaluation to more datasets and on different post-processing combinations, where our method demonstrates much stronger robustness than the baseline methods across all evaluated datasets (Appendix L, Table A7).

* Evaluated our method and the baseline methods with more metrics, including TPR@0.5%FPR, TPR@0.1%FPR, AUC, and ROC. Our method demonstrates superior performance across all metrics, even achieving near-perfect TPR at 0.1%FPR for many models (Appendix M, Table A8, Table A9, Table A10, and Figure A5).

* Provided additional experiments where our approach achieves high performance for AR attribution with different dataset splits, and clarifed that our method mainly focuses on model family attribution (Appendix V, Table A22, Table A23, Table A24).

* Designed and evaluated different methods to combine our QuantLoss and EncLoss, including addition, multiplication, and learned weights (Appendix P, Table A14).

* Demonstrated that our method generalizes well on different sampling conditional guidance scales, different sampling temperatures, and different data splits (Appendix N, Table A11, Table A12).

* Demonstrated our method still achieves near-perfect performance when the images come from mixed sources, including 3 natural datasets and images generated by 6 different models (Appendix S, Table A19).

---

### Author Response · Authors · 2025-12-03
**Discussion Summary (1/3)**

We would like to thank all the Reviewers and the Area Chairs for the great efforts that they devoted to the discussion process. We believe that we addressed all the Reviewers’ comments and questions with extensive experiments and detailed clarifications, which helped us further improve our work.

Notably, three of four Reviewers have replied to us and show recognition of our work’s value, where **Reviewer 891z raised the rating from 6 to 8, Reviewer SetU from 4 to 6, and Reviewer XMV2 kept the initial high score of 8**. Although Reviewer Rqvo did not have the chance to reply to our response yet, all of their questions were addressed also in responses to other Reviewers. The other Reviewers confirmed that our answers fully addressed their questions.

On top of these positive feedback from the Reviewers, we further (after the first part of the rebuttal) included evaluation of **3 additional baseline methods** and showed that our method **significantly outperforms all of the baselines**. We would like to summarize our discussion with each Reviewer as follows:

## 1. Discussion with Reviewer XMV2

Reviewer XMV2 initially gave us rating 8 and decided to **“keep the high score” of 8** after the discussion. We addressed their questions with the following responses:

* Quantified our main observation: we showed that the feature map of a generated image has a smaller distance to the codebook entries than the feature map of a natural image (Appendix K, Table A6).

* Provided more explanation and extensive analysis for the optimized token search (Algorithm 3), directly compared the algorithm with the greedy search (Algorithm 2), and included hyper-parameter test (Table 6) and acceleration options (Appendix U, Table A21).

* Designed a more challenging adaptive adversarial attack and designed a defense strategy that largely improves the robustness of our method against such attacks, outperforming the baseline methods (Appendix R, Table A18).

* Evaluated two distance metrics, L2 norm and cosine similarity, and showed that our method has nearly 100% TPR at 1% FPR with both metrics (Appendix O, Table A13).

---

### Meta-Review · Area_Chair_JMUD · 2026-01-07

**Summary:**

The reviewers agree the submission tackles an important problem (provenance/attribution for images from discrete autoregressive generators) and presents consistently strong empirical results across multiple model families. The main pre-rebuttal concerns centered on (i) limited baseline coverage and unclear relationship to membership inference, (ii) practical constraints (white-box access, per-model fine-tuning, and runtime—especially for the multi-scale setting), and (iii) insufficient analysis/justification (quantifying the core observation, Algorithm 3 details/sensitivity, choice/combination of signals, broader metrics/robustness, and generalization settings).

**Reviewer Concerns:**

Addressed concerns:
- Authors added/ran additional detector baselines and also explored MIA-based baselines under extra information (labels/prompts) plus a classifier-based detector; reviewers who raised this concern (notably 891z) explicitly stated it was largely resolved.
- Authors clarified task mismatch and applicability constraints; both SetU and 891z indicated their conceptual concerns were addressed.
- Authors added quantitative evidence supporting the “closer-to-codebook” phenomenon, satisfying XMV2’s main missing-piece critique.
- Authors provided direct comparisons against the greedy/standard quantization, hyperparameter sweeps, and acceleration options; XMV2 remained satisfied and 891z raised the score.
- Authors reported stricter operating points (e.g., TPR at lower FPR), AUC/ROC, class-split experiments to mitigate leakage concerns, and tests across sampling settings; SetU and 891z treated these as resolving key questions.
- Authors added an adaptive-attack evaluation and a defense via augmentation-based fine-tuning; reviewers did not raise further objections afterward.

Remaining concerns:
- While the rebuttal adds analyses and tests (e.g., statistical testing and ablations), the core critique that the conceptual advance is incremental and theory is limited appears only partially mitigated. This is unlikely to fully shift a novelty-focused reviewer’s view, even if empirical strength is clear.
- The rebuttal provides plausible deployment narratives (e.g., model-owner APIs) and cost comparisons, but the fundamental dependency on privileged access and additional fine-tuning remains a real limitation for third-party attribution in closed-source settings. This may keep at least one reviewer cautious.

**Reviewer Scores:**

- Reviewer XMV2: stays 8 → 8 (explicit follow-up keeping the score).
- Reviewer 891z: 6 → 8 (explicit follow-up raising score).
- Reviewer SetU: 4 → 6 (explicit follow-up raising score; subsequent positive comment).
- Reviewer Rqvo: likely 4 → 5–6 (no follow-up, but the rebuttal directly targets their listed weaknesses/questions: MIA distinction, more baselines, and closed-source discussion).

---

### Decision · Program_Chairs · 2026-01-26

Accept (Poster)